# Fast label-free live imaging with FlowVision reveals key principles of cancer cell arrest on endothelial monolayers

Gautier Follain [1,2,3], Sujan Ghimire [2,15], Joanna W Pylvänäinen [2,4,15], Monika Vaitkevičiūtė [1,2,4], Iván Hidalgo-Cenalmor[2], Diana Wurzinger [2], Camilo Guzmán [5], James R W Conway [6], Michal Dibus[1], Jouni Härkönen [1,7], Sanna Oikari[8], Kirsi Rilla[8], Marko Salmi[4,9,10], Johanna Ivaska [1,3,11,12,13 ✉] & Guillaume Jacquemet [1,2,3,14 ✉]

## Abstract

The rapid, transient, and unpredictable nature of interactions between circulating cells and the endothelium challenges the investigation of these events under flow conditions. Here, we developed an imaging and image-analysis framework called Flow-Vision, which integrates fast, bright-field live-cell imaging with deep-learning-based image analysis to quantitatively track cell landing and arrest on an endothelial monolayer under physiological flow conditions. Using FlowVision, we find that pancreatic ductal adenocarcinoma (PDAC) cells exhibit variable adhesion strength and flow sensitivity. Remarkably, some PDAC cells demonstrate comparable endothelial engagement to leukocytes, preferentially arresting at endothelial junctions, providing them access to the underlying basal extracellular matrix. PDAC cells attach and form clusters in areas with high expression of the endothelial CD44 receptor. Targeting CD44 using siRNA, function-blocking antibodies, or degrading its ligand, hyaluronic acid (HA), strongly reduces PDAC cell attachment. Overall, our label-free live-imaging approach demonstrates that cancer and immune cells share both common and unique features in endothelial adhesion under flow, and allows identification of CD44 and HA as key mediators of PDAC cell arrest.

**Keywords** Circulating Tumor Cells; Circulating Immune Cells; Adhesion; Image Analysis; Deep Learning
**Subject Categories** Cancer; Cell Adhesion, Polarity & Cytoskeleton; Vascular Biology & Angiogenesis

## Introduction

Pancreatic ductal adenocarcinoma (PDAC) is the most common and lethal form of pancreatic cancer, notorious for its aggressive nature and poor prognosis. Despite advancements in surgical and chemotherapeutic approaches, the five-year survival rate for PDAC patients remains low, at just 10% (Park et al, 2021). The prognosis marginally improves to around 20% for those who undergo surgical resection, but the majority of patients present with unresectable advanced or metastatic disease (Strobel et al, 2022). The poor outcomes associated with PDAC underscore the need for a deeper understanding of the mechanisms driving its progression to metastasis.

Metastasis is a complex, multi-step process that enables cancer cells to escape the primary tumor, enter the circulatory system, and colonize distant organs (Gupta and Massagué, 2006). A critical but poorly understood step in this cascade involves circulating tumor cells (CTCs) adhering to the vascular endothelium, crossing the endothelial barrier (extravasation), and establishing metastatic lesions. While leukocyte extravasation follows a well-characterized cascade (Butcher, 1991; Wen et al, 2022), the specific molecular events involved in CTC arrest, adhesion, and extravasation, especially in PDAC, remain less defined.

Within the bloodstream, multiple biomechanical factors influence the arrest of CTCs on the endothelium. First, blood flow exerts shear forces that favor continued circulation (Follain et al, 2018a). Second, the diameter and complexity of the vascular network can facilitate passive partial or complete vascular occlusion, leading to CTC arrest (Kienast et al, 2010; Follain et al, 2018a). Finally, CTCs can also actively engage endothelial cells through specific adhesion receptors, promoting stable arrest and subsequent extravasation, a process that mirrors the features of leukocyte transmigration

[1]Turku Bioscience Centre, University of Turku and Åbo Akademi University, Turku 20520, Finland. [2]Faculty of Science and Engineering, Cell Biology, Åbo Akademi University, Turku 20520, Finland. [3]Turku Collegium for Science, Medicine and Technology (TCSMT), Turku 20520, Finland. [4]InFLAMES Research Flagship Center, University of Turku and Åbo Akademi University, Turku 20520, Finland. [5]Euro-BioImaging ERIC, Turku 20520, Finland. [6]Department of Biochemistry and Developmental Biology, Faculty of Medicine, University of Helsinki, Helsinki 00014, Finland. [7]Department of Pathology, Hospital Nova of Central Finland, Well Being Services County of Central Finland, Jyväskylä 40620, Finland. [8]Institute of Biomedicine, University of Eastern Finland, Kuopio 70211, Finland. [9]Institute of Biomedicine, University of Turku, Turku 20520, Finland. [10]MediCity Research Laboratory, University of Turku, Turku 20520, Finland. [11]Department of Life Technologies, University of Turku, Turku 20520, Finland. [12]Western Finnish Cancer Center (FICAN West), University of Turku, Turku 20520, Finland. [13]Foundation for the Finnish Cancer Institute, Tukholmankatu 8, Helsinki 00014, Finland. [14]Turku Bioimaging, University of Turku and Åbo Akademi University, Turku 20520, Finland. [15]These authors contributed equally: Sujan Ghimire, Joanna W Pylvänäinen. ✉E-mail: johanna.ivaska@utu.fi; guillaume.jacquemet@abo.fi

(Vestweber, 2015; Osmani et al, 2019; Dupas et al, 2024). Importantly, passive occlusion alone appears to be insufficient for successful extravasation (Gassmann et al, 2009; Reymond et al, 2012; Offeddu et al, 2021), and even CTC clusters can squeeze through small capillaries (Au et al, 2016).

One challenge in studying CTC-endothelial interactions lies in the rapid and transient nature of these events under flow conditions. Current in vivo models, such as intravital imaging in zebrafish embryos or mice, offer critical insights (Karreman et al, 2023; Paul et al, 2019; Follain et al, 2018a), but are limited by scalability, resolution, and throughput, which hinder systematic and mechanistic studies. Therefore, there is a need for automated, high-throughput methods that enable molecular-level insights into CTC–endothelial interactions.

Here, we introduce FlowVision, a single-cell resolution imaging and analysis framework designed to monitor interactions between circulating cells and an endothelial monolayer under physiological flow conditions. Using this framework, we compare the behavior of multiple PDAC cell lines with each other and with leukocytes to generate unique adhesion and migration profiles under various physiological stimuli and flow conditions. Our analysis reveals previously unreported similarities and differences between leukocyte and cancer cell adhesion, as well as notable heterogeneity in the ability of cancer cell lines to interact with the endothelial monolayer. In addition, we observe significant spatial heterogeneity in cell surface receptor expression on the endothelial monolayer, identifying hotspots with high CD44 levels where PDAC cells tend to arrest. Targeting CD44, either by silencing its expression or inhibiting its function with a monoclonal antibody, significantly reduces PDAC cell attachment to endothelial cells. Moreover, we demonstrate that hyaluronic acid, a key ligand for CD44, plays a crucial role in PDAC cell adhesion under flow conditions. Overall, FlowVision offers a dynamic and quantitative platform for studying cell interactions with the endothelial monolayer, facilitating an understanding of how various features of different cell types influence their arrest on the endothelium.

# Results

## Arrest of PDAC cells on endothelial cells is modulated by flow and intrinsic cancer cell properties

Microfluidics has been widely and successfully used to determine the effects of blood flow on leukocyte adhesion to the endothelium (Lawrence et al, 1987; Grönloh et al, 2022; van der Meer et al, 2024). Here, we adopted a similar system to study PDAC–endothelial interactions by perfusing cancer cells under controlled flow profiles over an endothelial cell monolayer (Follain et al, 2018a; Osmani et al, 2019) (Figs. 1A and EV1). Initially, we validated the expected Poiseuille flow profile in the channel by tracking PDAC cells through BSA-coated channels (Appendix Fig. S1A). As anticipated for our perfusion, the highest flow velocity in the lower plane of the channel was in the center (400 μm/s), gradually decreasing toward the edges (measurable cell velocity ~150 μm/s; Appendix Fig. S1B).

We screened seven PDAC cell lines bearing common mutations found in patients, including constitutive activation of KRAS (G12D, G12C), P53 loss of function (point mutations on exons 5–7), and

loss of P16 and SMAD4 (Bardeesy and DePinho, 2002) (Appendix Fig. S1C), to determine whether they exhibit similarities in their ability to adhere to the endothelial monolayer, both with and without flow (Figs. 1B,C and EV2A,B; Appendix Fig. S1D). Surprisingly, we observed a high level of variability in endothelial cell adhesion among these cell lines that shows no clear link with their driver mutation burden. In both conditions, AsPC-1 and MIA PaCa-2 cells exhibited high adhesion capability, while SW1990 and PANC1 cells exhibited reduced adhesion under flow conditions. Panc 10.05 demonstrated a limited ability to engage the endothelial cell monolayer, both with and without flow. Regardless of the cell line, adhesion events were more favorable near the channel edge, further emphasizing the critical role of the flow velocity during the adhesion process (Fig. 1D).

Theoretically, cell size should influence arrest dynamics, as larger cells experience greater frictional forces under perfusion, causing them to move more slowly (Low Reynolds; Stokes' law, $F = 6\pi\eta rv$). To assess this effect, we measured cell size and observed significant variations, with up to a twofold difference in cell diameter between PDAC cell lines (Fig. EV2C). However, there was no clear correlation between cell size and adhesion capability. Therefore, fluid physics alone is unlikely to fully account for the differences in adhesion between the cell lines, implying a greater role for cell-intrinsic features, such as the differential surface expression of adhesion receptors compatible with the endothelial layer.

## FlowVision reveals dynamic adhesion patterns of PDAC cells under varying flow conditions

To gain dynamic insights into cancer cell adhesion to endothelial cells, we implemented a live-imaging strategy using transmitted light illumination on a widefield microscope at 25 Hz (Figs. 1E and EV1; Movie EV1). By employing label-free imaging, we minimize system perturbation and avoid the phototoxic effects typically associated with fluorescence imaging (Del Rosario et al, 2025; Gómez-de-Mariscal et al, 2024a), ensuring more accurate and physiologically relevant observations. Intrigued by the distinct adhesion behaviors observed in the 10-min fixed-time-point experiments, we focused our analysis on three PDAC cell lines: AsPC-1, MIA PaCa-2, and Panc10.05 (Movies EV1 and 2). We perfused cells as single cells and recorded their adhesion dynamics over 8 min, systematically decreasing flow speeds with 2-min intervals to examine the influence of different flow conditions: 400 μm/s (high), 200 μm/s (medium), and 100 μm/s (low; flow speed found in capillaries) finally increasing the flow speed back to 400 μm/s ("wash") to test the stability of cell adhesions to the endothelial monolayer (Movies EV1 and 2). The resulting wall shear stresses applied to the circulating cells in our system at the surface of the endothelial cells range from 0.4 to 0.05 dyn/cm².

To obtain unbiased, quantitative measurements of cell landing, arrest, and migration under flow conditions, we developed FlowVision, a label-free analysis pipeline that processes brightfield time-lapse movies captured in microfluidic channels (Figs. 1F and EV1; Appendix Fig. S2A). From each movie, cells are segmented using StarDist, a deep-learning instance segmentation method ideally suited for detecting rounded cells, which we custom-trained on our annotated frames (Schmidt et al, 2018). Importantly, out-of-focus cells and very large cell clusters (>10 cells) were not detected at this stage.

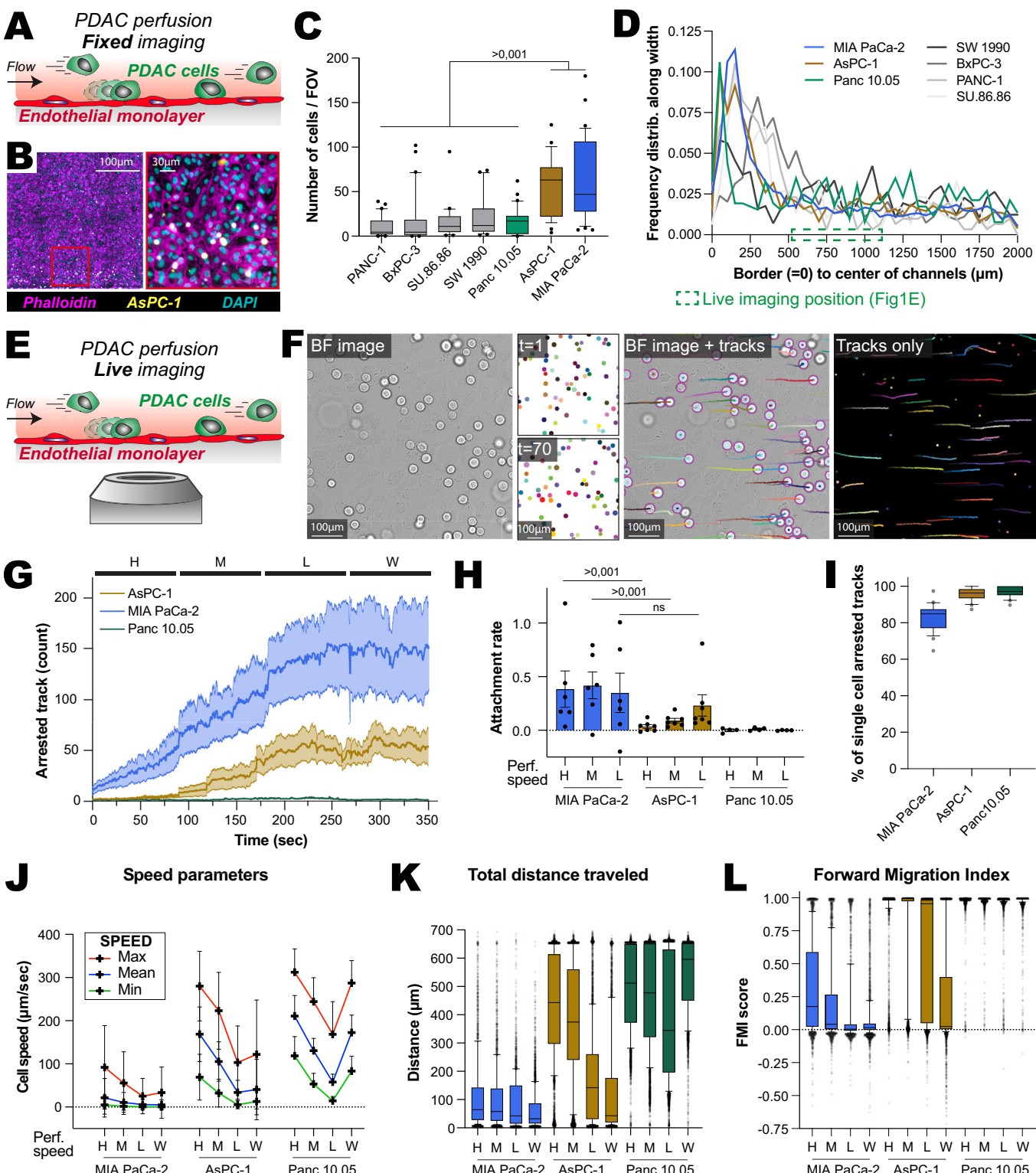

Segmented cells are then linked into trajectories with the Fiji plugin TrackMate (Ershov et al, 2022). The resulting tracks are then aggregated and analyzed using a modified version of CellTracksColab (Gómez-de-Mariscal et al, 2024b). This notebook-based environment enables the batch processing of millions of tracks, calculation of track metrics, event classification (such as landing frequency, arrest onset/

duration, and endothelial migration), and downstream statistical analysis (Fig. EV1; see "Methods" for details). Importantly, all deep-learning models, their training datasets, and the code used are available on a dedicated Zenodo community.

Initially, we quantified cell attachment over time derived from the instantaneous speed of cell tracks (Fig. 1G,H). MIA PaCa-2 cells

◄   **Figure 1.   PDAC cell arrest on endothelial cells is modulated by flow and intrinsic cancer cell properties.**

(A) Schematic representation of the microfluidic system used to study the interactions between pancreatic ductal adenocarcinoma (PDAC) cells and endothelial cells. (B–D) Labeled (CellTrace) PDAC cells were perfused over an endothelial monolayer at 400 µm/s for 10 min. Samples were fixed, stained with phalloidin and DAPI, and imaged using a spinning disk confocal microscope. (B) A representative image shows the endothelial monolayer with attached AsPC-1 cells. The red box highlights a region of interest that is magnified. Scale bars: 100 µm (main), 30 µm (inset). (C) The number of attached cells was quantified for each cell line. The boxes capture the interquartile range, with the median marked by a line within each box. Data points falling outside the whiskers are depicted as individual dots ($n = 25$–36 fields of view, 3–7 biological repeats). The $P$ values were determined using a randomization test. (D) Frequency plot showing the overall distribution of cancer cells across the microfluidic channel from the edge (border) to the center ($n = 302$–2279 cells) and highlighting the region selected for subsequent live-cell imaging. (E–L) AsPC-1, MIA PaCa-2, and Panc 10.05 cells were perfused over an endothelial monolayer and imaged using a brightfield microscope at 25 Hz for 8 min, with flow speeds varied at 2-min intervals: 400 µm/s (High, H), 200 µm/s (Medium, M), and 100 µm/s (Low, L). Each sequence concluded with an increased flow speed of 400 µm/s to test the stability of the adhered cells (Wash, W). (F) A representative brightfield image (BF) showing the detected PDAC cells at two different time points and the resulting tracks. (G) The number of arrested PDAC cells over time for each cell line tested. Bold lines indicate the average, and shaded areas represent the SD (4–7 biological repeats, see "Methods"). The different flow speeds are shown. (H) The attachment rate for each cell line at each flow speed is displayed as a bar chart (mean $+/-$ SEM) with individual data points (4–7 biological repeats, see "Methods"). The $P$ values were determined using a randomization test. (I) Percentage of AsPC-1 and MIA PaCa-2 cells arresting as single cells per movie. Here, a cluster is defined as at least two cells that arrest together (at the same time) within a cell diameter of each other (see "Methods" for details). Results from the various flow speeds are pooled ($n > 24$ videos, 6–7 biological repeats; see "Methods"). (J–L) Analysis of track metrics for MIA PaCa-2, AsPC-1, and Panc 10.05 cells at different flow speeds. (J) Plot of mean, maximum, and minimum track speeds for each cell line. Data are presented as mean ± SD. (K) Total distance traveled by PDAC cells during perfusion. (L) Forward Migration Index (FMI) along the flow direction for each cell line. (I, K, L) Results are presented as boxplots, where the whiskers extend from the 10th to the 90th percentiles. The boxes capture the interquartile range, with the median marked by a line within each box. Data points outside the whiskers are depicted as individual dots ($n = 3011$–8227 tracks, 4–7 biological repeats, see "Methods"). The numerical data and images used for this figure, as well as statistical summaries including pairwise Cohen's $d$ values and results from statistical tests, have been archived on Zenodo (https://doi.org/10.5281/zenodo.17232437).

demonstrated a superior capacity for adhering to endothelial cells, even under the highest flow speed (Movies EV1 and 2). Their adhesion to the endothelial cell monolayer gradually increased with decreasing flow, and the cells adhering at low flow maintained a steady attachment even when the flow speed was returned to 400 µm/s ("wash" step). Conversely, AsPC-1 cells failed to adhere at the highest flow speed (400 µm/s) but displayed increasing adhesion with decreasing flow, maintaining steady adhesion during the wash step. Concordant with the fixed-cell data, Panc 10.05 cells exhibited minimal adhesion at all flow speeds (Fig. 1G,H). While our segmentation strategy excluded large aggregates (tens to hundreds of cells) from analysis, we often observed smaller clusters (2–8 cells) engaging with the endothelium and successfully arresting. We measured arrests by classifying them as single cells versus small clusters (2–8 cells). Overall, the majority of arrests in our microfluidic experiments occur at the single-cell level. In MIA PaCa-2, more than 85% and in AsPC-1, about 96% of arrest events are single cells (Fig. 1I). In MIA PaCa-2, over 50% of cluster arrests involve two cells.

To further analyze adhesion dynamics at the single-cell level, we measured each cell's slowest, fastest, and average speeds based on frame-to-frame movement (Fig. 1J; Appendix Fig. S2B). We also calculated the total distance traveled, which reflects how far each cell moved along its path during the analysis period (Fig. 1K). Directional bias was evaluated using the forward migration index (FMI), indicating how much a trajectory aligns with or against the flow (positive = downstream, 0 = no bias, negative = upstream; Fig. 1L). The cell track mean speeds showed that MIA PaCa-2 cells consistently move slower than the flow speed, suggesting adhesion at all flow speeds. Their low average total distance traveled and FMI near zero indicate stable cell adhesion over time. AsPC-1 tracks exhibited decreased velocity, distance, and FMI, along with lower flow speeds and low FMI during the wash step, indicating many cells remained stably adherent to the endothelial cell monolayer. In contrast, Panc 10.05 cell speeds and distances closely matched flow speeds, with an FMI of 1, indicating no adhesion.

Altogether, FlowVision enabled a nuanced differentiation between PDAC cell adhesion behaviors, revealing that both flow

speed and the intrinsic properties of cancer cells play critical roles in their adhesion to endothelial cells. These results highlight the utility of our dynamic, label-free imaging approach for studying cancer cell interactions under physiologically relevant conditions, providing a solid foundation for further exploration of the molecular mechanisms underlying cancer cell adhesion.

## PDAC cells can adhere to endothelial cells as efficiently as immune cells

We next compared cancer-endothelium interactions with leukocyte-endothelium interactions. For these experiments, we isolated neutrophils and a PBMC fraction (primarily comprising lymphocytes and monocytes) directly from whole human blood (Fig. 2A). Leukocyte interaction with endothelial cells is augmented during inflammation. Therefore, we expanded our endothelial adhesion experiments to include both untreated ("resting", CTRL) and monolayers activated with the inflammatory cytokine IL-1β (Dinarello, 2018).

First, we tested the effects of short (2 h) and prolonged (16 h) stimulation with IL-1β on our endothelial monolayers. Analysis of endothelial adhesion molecules known to influence leukocyte attachment showed expected activation patterns (Shen et al, 1997; Smith et al, 1988; Chappell et al, 1998): ICAM-1 and VCAM-1 increased after 2 h and continued to rise by 16 h, while E-selectin was elevated at 2 h but decreased by 16 h (Fig. EV2D,E). In contrast, endothelial CD44 expression mostly remained unchanged by IL-1β (Fig. EV2D,E).

Next, we performed live perfusion experiments (Fig. 2B; Movie EV3). As expected, IL-1β significantly increased neutrophil attachment and led to transmigration within the 8 min of acquisition, consistent with previous reports (Woodfin et al, 2009; Smith et al, 1988) (Movie EV4). PBMCs showed minimal adhesion to the resting monolayer, but IL-1β treatment noticeably enhanced their adhesion (Fig. 2C–H; Appendix Fig. S3) to a lesser extent than previously described in static conditions (Chappell et al, 1998). Extended observation under IL-1β treatment revealed a tendency for PBMCs

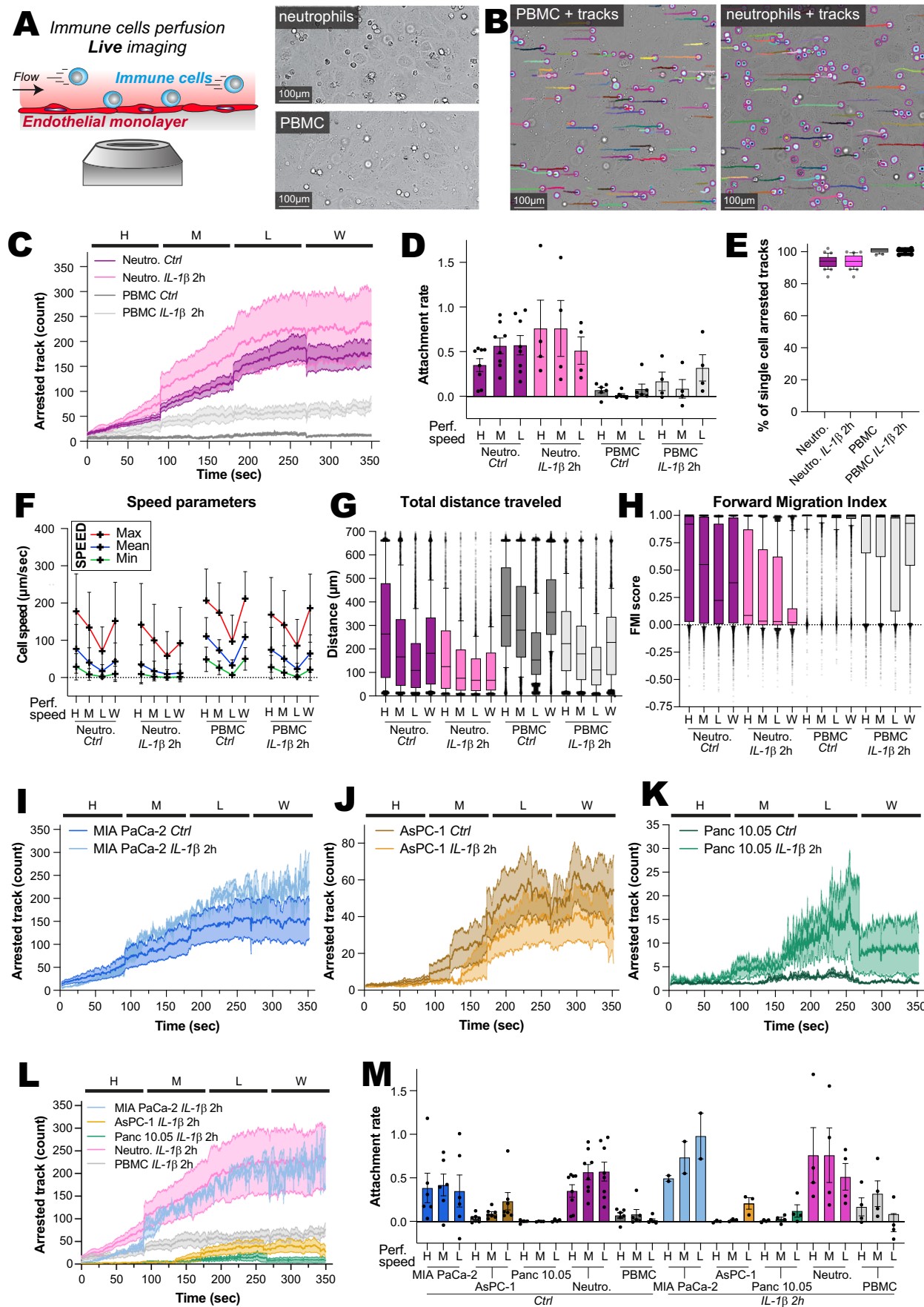

**Figure 2.  PDAC cells can interact with endothelial cells as efficiently as immune cells.**

(A–M) Neutrophils, PBMCs, AsPC-1, MIA PaCa-2, and Panc 10.05 cells were perfused over an endothelial monolayer in the presence or absence of IL-1β stimulation (10 ng/ml for 2 h) and imaged using a brightfield microscope at 25 Hz for 8 min. Flow speeds varied in 2-min intervals: 400 µm/s (High, H), 200 µm/s (Medium, M), 100 µm/s (Low, L), and 400 µm/s (Wash, W). (A–H) Analysis of neutrophils and PBMCs attachment to the endothelial monolayer in the presence or absence of IL-1β stimulation (10 ng/ml for 2 h of pretreatment). (A, B) Representative brightfield images showing neutrophils and PBMCs and their resulting tracks. (C) The number of arrested Neutrophils (Neutro.) and PBMCs over time, with bold lines indicating the average and shaded areas representing the SD (4–8 biological repeats, see "Methods"). (D) The attachment rates for neutrophils and PBMCs at each flow speed are displayed as a bar chart (mean ± SEM) with individual data points (4–8 biological repeats, see "Methods"). (E) Percentage of neutrophils and PBMCs cells arresting as single cells per movie in the presence or absence of IL-1β stimulation (10 ng/ml for 2 h). Here, a cluster is defined as at least two cells that arrest together (at the same time) within a cell diameter of each other (see "Methods" for details). Results from the various flow speeds are pooled (n > 16 videos, 4–8 biological repeats; see "Methods"). (F–H) Analysis of track metrics for neutrophils and PBMCs at different flow speeds. (F) Plot of mean, maximum, and minimum track speeds. Data are presented as mean ± SD (4–8 biological repeats, see "Methods"). (G) Total distance traveled by neutrophils and PBMCs during perfusion. (H) FMI along the flow direction for each condition. (E, G, H) Results are presented as boxplots, where the whiskers extend from the 10th to the 90th percentiles. The boxes capture the interquartile range, with the median marked by a line within each box. Data points outside the whiskers are depicted as individual dots (n = 3044–12,905 tracks, 4–8 biological repeats, see "Methods"). (I–K) Impact of IL-1β stimulation (10 ng/ml for 2 h) on PDAC cell attachment to the endothelial monolayer. The number of arrested MIA PaCa-2 (I), AsPC-1 (J), and Panc 10.05 (K) cells over time, with bold lines indicating the average and shaded areas representing the standard deviation (2–7 biological repeats, see "Methods"). Different flow speeds are shown. (L, M) Comparison of PDAC and immune cells' attachment to endothelial cells in the presence of IL-1β stimulation (10 ng/ml for 2 h). The number of arrested cells over time, with bold lines indicating the average and shaded areas representing the standard deviation (2–8 biological repeats, see "Methods"). (M) The attachment rate for each condition is displayed as a bar chart (mean ± SEM) with individual data points. The numerical data and images used for this figure, as well as statistical summaries including pairwise Cohen's d values and results from statistical tests, have been archived on Zenodo (https://doi.org/10.5281/zenodo.17232437).

to exhibit cyclic adhesion and detachment, except in interactions with specific endothelial cells, where they exhibited pronounced accumulation (Movie EV4). Interestingly, a 2-h stimulation with IL-1β was sufficient to induce near-maximal adhesion of neutrophils and PBMCs to the endothelium, as the 16-hour activation resulted in only a slight additional increase over the 2-hour treatment (Fig. EV2F,G). As for PDAC cells, neutrophils and PBMCs primarily attached as single cells, yet neutrophils tended to attach as clusters more frequently than PBMCs, a phenomenon that was not influenced by IL-1β stimulation (Fig. 2E).

For comparative purposes, we also analyzed PDAC cell adhesion to the endothelium stimulated with IL-1β for 2 h. We observed varied responses when examining the attachment of AsPC-1, MIA PaCa-2, and Panc 10.05 cancer cells to activated (i.e., IL-1β-treated) endothelial cells. IL-1β facilitated a minor increase in MIA PaCa-2 adhesion (Fig. 2I) and reduced AsPC-1 adhesion (Fig. 2J). Interestingly, IL-1β enabled poorly adherent Panc 10.05 cells to start adhering at low flow. However, they became displaced with the increased "wash" flow, indicating transient rather than stable adhesion (Fig. 2K). In contrast to neutrophils, PDAC cells were unable to rapidly transmigrate within the 8-min imaging period. Comparing the adhesion profiles and track metrics of cancer cells to those of immune cells revealed similar adhesion profiles between MIA PaCa-2 cells and neutrophils (Figs. 2L,M and EV2H,I). In contrast, Panc 10.05 adhesion profiles resemble those of PBMCs.

## PDAC and immune cells exhibit delayed arrest following the initial landing on endothelial monolayers

Given the similarities in the adhesion profiles between PDAC cells and immune cells, we investigated the initial adhesion and stabilization phases of cells arresting on the endothelial cell monolayer. To do this, we narrowed our analysis to include only tracks exhibiting a prominent attachment profile (Figs. 3A,B and EV1; see the figure legend and "Methods" for details). All flow speeds were combined in these analyses to collect sufficient tracks. Due to their infrequent adhesion events, Panc 10.05 cells were excluded from this analysis.

Examining the track metrics from the point of initial contact (landing) to the first moment of arrest, we found that cells do not halt immediately upon encountering the endothelial layer (Fig. 3B–F). Instead, they continue to move slowly, predominantly in the direction of the flow for more than the length of a cell diameter (around 17 µm for MIA PaCa-2, Fig. 3C). After the first arrest, the cells transition several times between arrest and movement leading to longer distances traveled (Fig. 3D). This was particularly evident in MIA PaCa-2 cells and neutrophils. The AsPC-1 cells moved a shorter distance, and all three cell types were insensitive to IL-1β in their movement. In contrast, IL-1β activation triggered a rapid arrest of PBMCs after landing (Fig. 3C), and they moved a markedly shorter distance before establishing stable adhesion (Fig. 3D). Interestingly, during the phase of periodic attachment and movement, the FMI decreased (Fig. 3E), indicating that the cells no longer strictly followed the flow direction and even displayed migration against the flow (negative FMI values). While negative FMI was observed in all cell types, it was more pronounced in immune cells.

To better characterize cell behavior, we counted the number of track speed peaks after the first arrest to quantify transient arrest cycles (Fig. 3F). Despite significant heterogeneity, our quantitative analyses revealed distinct phenotypes within the cell populations. MIA PaCa-2 cells transitioned between arrest and movement multiple times, explaining the high distance traveled. In contrast, AsPC-1 cells display fewer speed peaks and stabilize more rapidly, resulting in shorter migration on the endothelial surface. Neutrophils stabilized efficiently but migrated above or below the endothelial layer, displaying high-speed peaks. PBMCs adhered and detached quickly, depending on the state of the endothelial cells, and did not migrate upon landing.

## PDAC cells and immune cells land and arrest in the proximity of endothelial cell–cell junctions

Having characterized the adhesion and movement dynamics of cancer and immune cells, we aimed to map their spatial locations within the endothelial layer with sub-cellular resolution. We

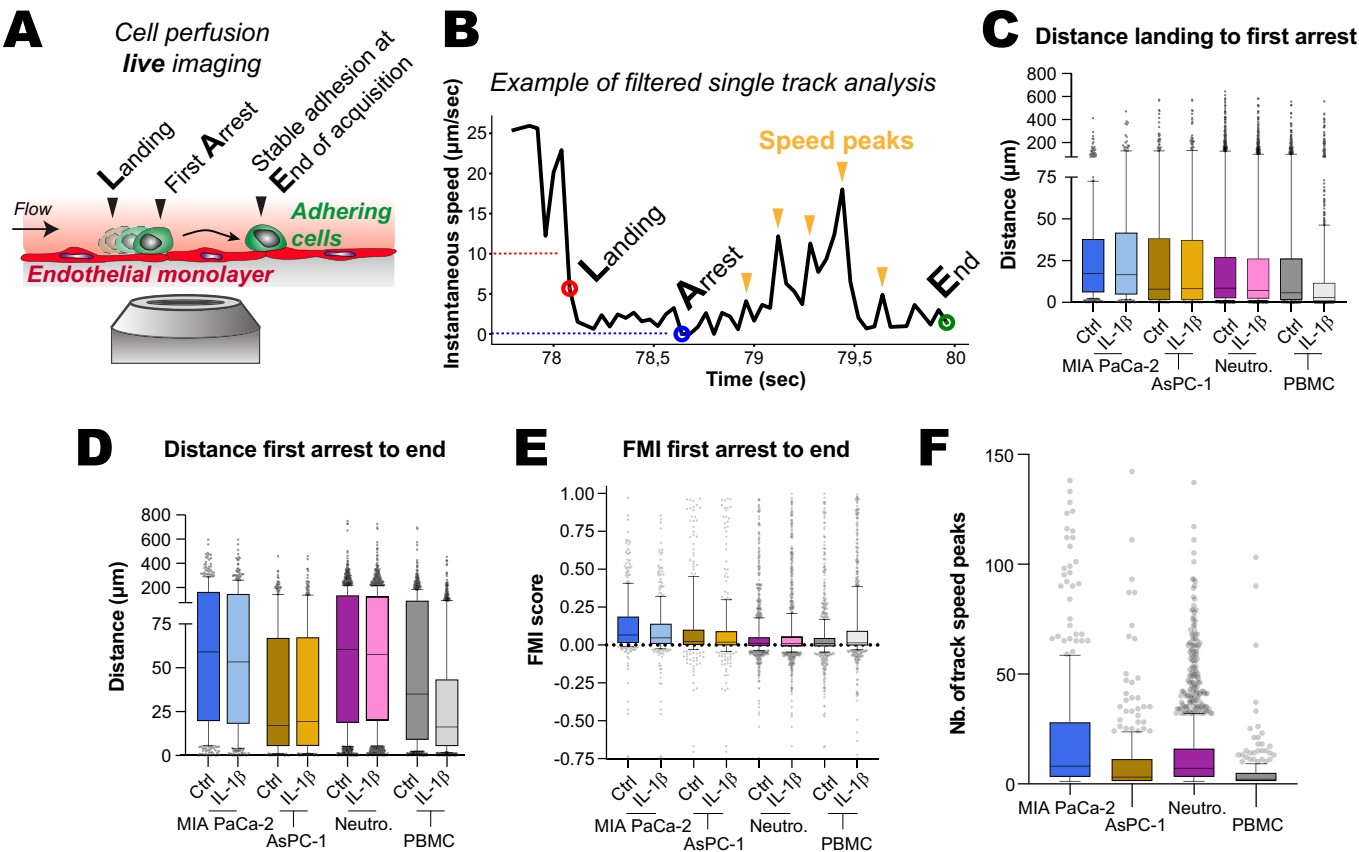

**Figure 3. Quantitative analysis of PDAC and immune cell landing, arrest, and migration patterns on endothelial cell monolayers.**

(A–F) To delineate cell trajectories indicative of a definitive arrest pattern, we refined our analysis to include only those tracks in which cells demonstrated an arrest pattern (see "Methods" for details). The initial point of landing was identified as the first coordinate at which the velocity of these designated landing tracks decreased to below 10 μm/s. Furthermore, the earliest point of arrest within these trajectories was identified as the coordinate at which cells first reached their minimum speed. Detachment peaks were then identified. (A) Schematic representation of the experiment setup. (B) A representative track displaying an arresting pattern is shown. The instantaneous velocity of the track is plotted over time. The points of Landing, Arrest, track end, and speed peaks are highlighted. (C) The total distance traveled between the initial landing and the point of arrest is displayed for each condition. (D) The total distance traveled between the point of arrest and the track end is shown. (E) The Forward Migration Index (FMI) between the point of arrest and the track end is displayed. (F) The number of detachment peaks per condition is displayed. (C–F) Results are presented as boxplots, where the whiskers extend from the 10th to the 90th percentiles. The boxes capture the interquartile range, with the median marked by a line within each box. Data points outside the whiskers are depicted as individual dots ($n = 334$–1711 tracks, 2–8 biological repeats, see "Methods"). The numerical data and images used for this figure, as well as statistical summaries including pairwise Cohen's $d$ values and results from statistical tests, have been archived on Zenodo (https://doi.org/10.5281/zenodo.17232437).

enhanced FlowVision by adding an artificial-labeling module that predicts pseudo-fluorescent markers for endothelial cell junctions and nuclei directly from a brightfield video. Specifically, a pix2pix conditional generative adversarial network (Isola et al, 2017), trained on paired brightfield/fluorescence images (PECAM-1 for junctions and DAPI for nuclei), was applied frame by frame to generate junction and nuclear probability maps. These maps were then segmented with Cellpose to identify individual endothelial nuclei and junctions. The resulting masks were registered with the circulating cell single-cell track data, allowing per-frame measurements such as the distance from a circulating cell to the nearest endothelial junction or nucleus (Figs. 4A and EV1; Appendix Fig. S4 and Movie EV5; see "Methods" for details) (Elmalam et al, 2024; von Chamier et al, 2021). This approach enabled the high-precision inference of endothelial junction and nuclear locations without the use of fluorescent markers (Appendix Fig. S4). Using this strategy, we quantified the proximity of cells to the nearest

endothelial nuclei and junctions during both the landing and arrest phases (Fig. 4B; Appendix Fig. S5A,B). Our analysis revealed that PDAC and immune cells predominantly land and arrest closer to endothelial junctions than nuclei, irrespective of IL-1β treatment (Fig. 4B; Appendix Fig. S5C). However, by comparing the sizes of PDAC and endothelial cells with the distances traveled during the arrest phase, we observed that the initial landing and final arrest sites at cell junctions are not always identical (Appendix Fig. S5D).

To corroborate these findings, we perfused PDAC cells on endothelial cell monolayers, fixed them, and stained them for an endothelial junction marker (Fig. 4C). This approach allowed us to distinguish and quantify the stable adhesions. Remarkably, over 95% of arrested cancer cells were found in direct contact with endothelial cell junctions (Fig. 4D). As proof of concept, we extended these observations to an in vivo context by injecting cancer cells into zebrafish embryos with labeled endothelial cell junctions (VE-Cadherin-GFP). Consistent with our in vitro

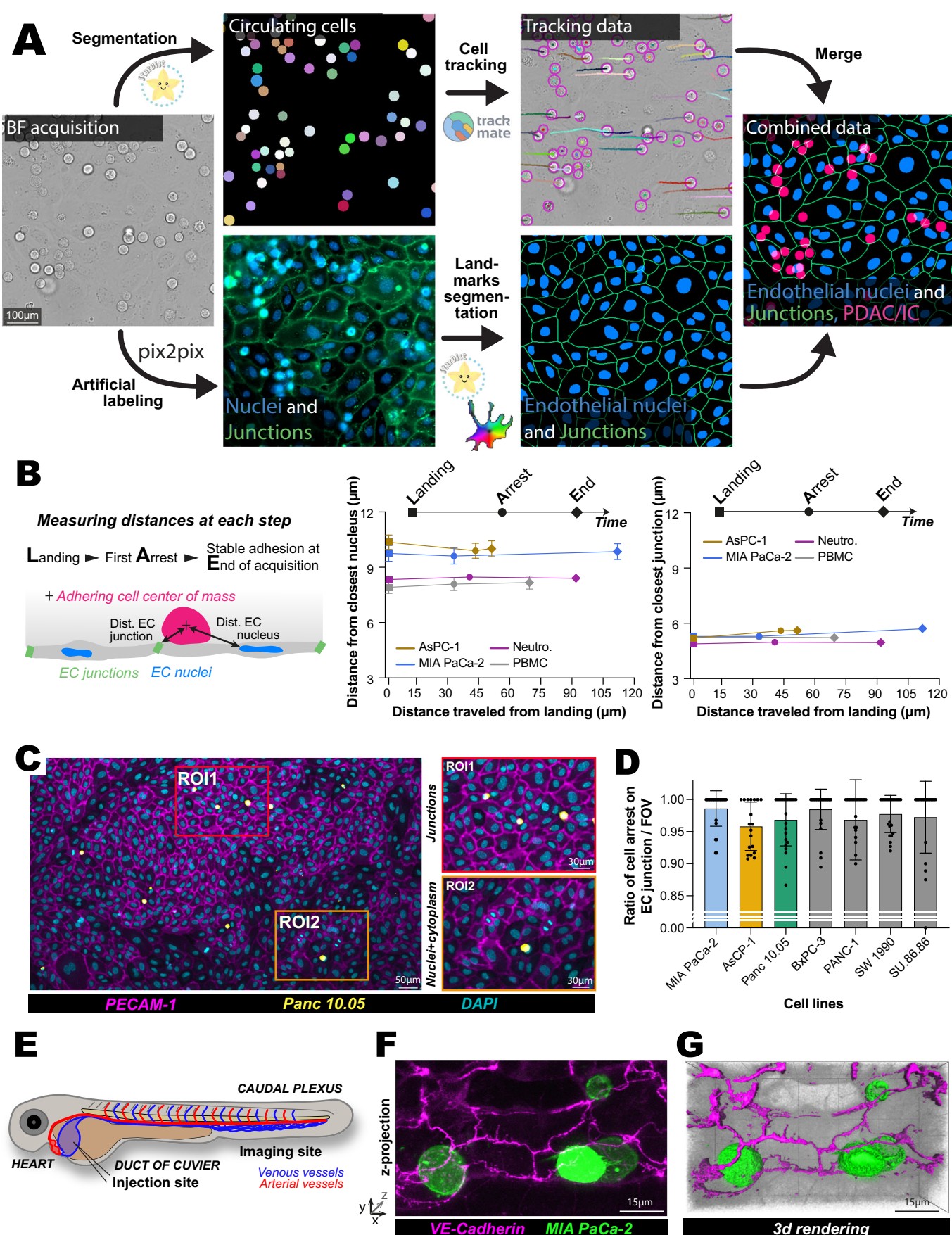

**Figure 4. PDAC cells and immune cells land and arrest in proximity to endothelial cell–cell junctions.**

(A, B) Analysis pipeline for predicting and segmenting endothelial cell junctions and nuclei from brightfield videos using deep-learning-based artificial labeling. (A) Brightfield videos were used to detect and track perfused cells, and to predict DAPI (all nuclei) and PECAM-1 (endothelial cell–cell junction) staining using artificial labeling. Endothelial cell nuclei were segmented using StarDist based on the predicted DAPI staining. Using both predicted DAPI and PECAM-1 staining, endothelial cells were segmented with Cellpose (see "Methods" for details). The distance between tracked cells and the nearest endothelial structures (junctions and nuclei) was calculated using the Euclidean distance transform algorithm from the SciPy library. Scale bar: 100 μm. (B) For each condition, the average distance from the cell center to the nearest nucleus and the average distance from the cell center to the nearest endothelial cell–cell junction are plotted (mean +/− SD) as a function of the total distance traveled from Landing to the first Arrest and the End of the track (n = 84–1139 tracks, 2–8 biological repeats, see "Methods"). (C, D) Labeled PDAC cell lines were perfused over an endothelial monolayer at 400 μm/s for 10 min. Samples were then fixed, stained with PECAM-1 and DAPI, and imaged using a spinning disk confocal microscope. (C) A representative image showing the endothelial monolayer with attached PDAC cells. Boxes highlight two regions of interest that are magnified. Scale bars: 50 μm (main), 30 μm (inset). (D) The ratio of PDAC cells arrested on top of junctions to those arrested on top of nuclei is displayed as a bar chart (mean +/− SD) with individual data points (n = 19–26 field of views, 3–4 biological repeats). (E-G) Lifeact-mScarlet-I MIA PaCa-2 cells were injected into 48 h post-fertilization *Tg(5xUAS:cdh5-EGFP)* zebrafish embryos. Three hours post-injection, embryos were imaged live using an Airyscan confocal microscope. A representative image and a 3D rendering are displayed. Scale bar: 15 μm. The numerical data and images used for this figure, as well as statistical summaries including pairwise Cohen's *d* values and results from statistical tests, have been archived on Zenodo (https://doi.org/10.5281/zenodo.17232437).

findings, these in vivo experiments demonstrated that arrested cancer cells are always near endothelial junctions (Fig. 4E–G). Altogether, our results clearly demonstrate preferential arrest near endothelial junctions.

## Endothelial cell junctions are stiffer and provide access to the basal extracellular matrix

The preferred adhesion of PDAC cells near endothelial cell–cell junctions prompted us to interrogate the biophysical and topological properties of the endothelial cell monolayer. First, we characterized the monolayer using atomic force microscopy to determine the endothelial cell surface topography and stiffness. Our data show that the endothelial cell monolayer is predominantly flat, with an average height difference of 2.4 μm between the tops of nuclei and the cell junctions. The nuclei are spaced, on average, 39 μm apart (Fig. 5A–C). These findings suggest that the monolayer topology is unlikely to determine the localization of cell arrest. Stiffness measurements indicated that the endothelial junctions were 28% more rigid (mean Young Modulus (YM) = 1218 Pa) compared to other monolayer regions (mean YM = 949 Pa) (Fig. 5D,E). This could facilitate adhesion at the junctions, given that increased stiffness promotes adhesion of most cancer cell types (Mathieu et al, 2024), and has also been shown to support the adhesion of neutrophils in vitro (Stroka and Aranda-Espinoza, 2009; Brandon et al, 2024).

Next, we hypothesized that endothelial cell–cell junctions could be a source of accessible extracellular matrix (ECM) components on the endothelial surface (Mana et al, 2016). We tested this by staining the endothelial cell monolayers for fibronectin and thrombospondin without permeabilization to avoid visualizing the intracellular ECM and the inaccessible basal ECM (Fig. 5F) (Ball et al, 2024). We chose fibronectin (FN) as it was previously implicated in CTC attachment to endothelial cells (Chen et al, 2016) and thrombospondin (THSD) due to its high expression in endothelial cells (Follain et al, 2021). Unexpectedly, we observed basal, rather than apical (Barbazán et al, 2017), patches of fibrillar fibronectin and thrombospondin under some endothelial cell–cell junctions, with the vast majority of the patches directly under a junction (Fig. 5G). Thus, we speculate that these basal ECM patches are exposed and accessible for antibody staining under junctions that undergo active reorganization, a normal physiological process for maintaining endothelial monolayers (Claesson-Welsh et al, 2021).

To assess the relevance of these ECM patches on PDAC cell adhesion, we analyzed the spatial relationship between arrested PDAC cells and ECM patches (Fig. 5H). We found that cancer cells arrest at varying distances from these ECM patches, with only 40% of adhesion events occurring directly on or near them (Fig. 5I). Yet, by randomizing our dataset, we found that proximity of cancer cells to ECM patches is higher than would be expected by chance (Fig. 5I). Using structured illumination microscopy, we observed that when cancer cells arrest on or close to junctions, they extend multiple filopodia-like protrusions toward the junctions, as well as to nearby endothelial actin cables. These filopodia also contact the underlying basal ECM when accessible (Fig. 5J). Altogether, our data suggest that PDAC cells are unlikely to require fibronectin or thrombospondin to attach to the endothelial monolayer. Instead, arresting near endothelial junctions provides PDAC cells access to the underlying basal ECM, which may facilitate the extravasation process (Wang et al, 2004; Chen et al, 2016).

## Local flow dynamics contribute to PDAC cell clustering on CD44-positive endothelial hotspots

Intrigued by the observation that PDAC cells tend to arrest near each other (Movies EV1 and 2), we utilized our tracking data to map the cells' arrest locations within each field of view over time (Fig. 6A). To investigate the spatial distribution and clustering patterns of these arrested cells, we applied a modified Ripley's L Function, which is a statistical tool used to assess whether objects in space are randomly distributed or tend to cluster in certain areas. Using Ripley's L function and Monte Carlo simulations, we revealed a pronounced tendency for arrested cells to cluster on the endothelial monolayer, especially at higher flow speed (Figs. 6B and EV3A,B). Notably, PDAC cells showed more clustering than neutrophils. AsPC-1 cells clustered the most, and the MIA PaCa-2 cells clustered similarly to PBMCs (Figs. 6B and EV3B). This finding suggests a non-random pattern of arrest on endothelial monolayers, with cells preferentially arresting at specific hotspots.

We initially hypothesized that arrest hotspots might result from differences in receptor expression on the apical surface of endothelial cells. To test this, we performed immunostaining of various endothelial cell adhesion molecules, including ICAM-1, ICAM-2, VCAM-1, E-selectin, CD44, and fibronectin, to analyze their distribution patterns in relation to cancer cell arrest sites (Fig. 6C). Importantly, stainings were performed without

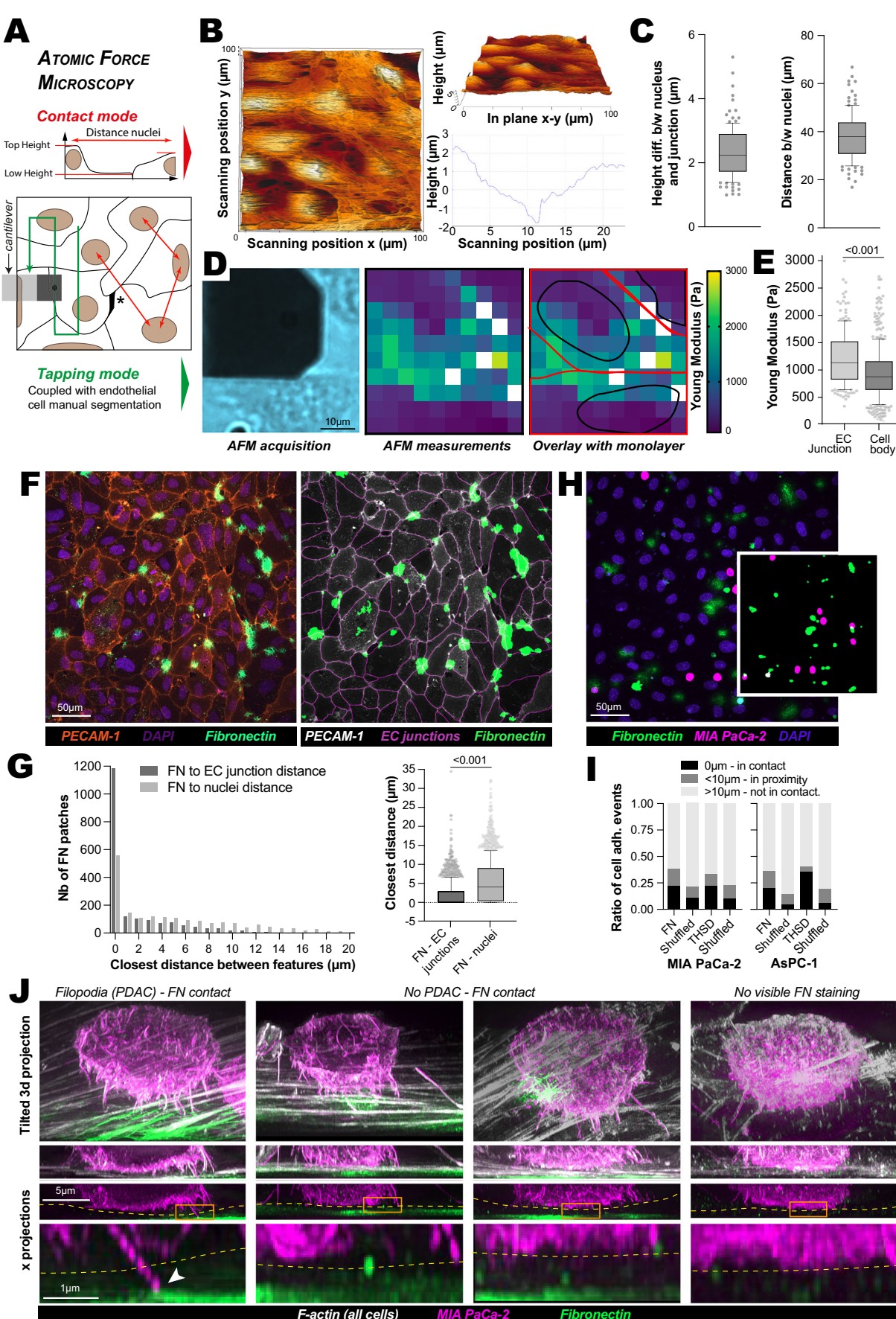

◀

**Figure 5. Endothelial cell junctions are stiffer and provide access to the basal extracellular matrix.**

(A–E) Biophysical characterization of the endothelial monolayer using atomic force microscopy (AFM). AFM was used in both contact mode (**B, C**) and tapping mode (**D, E**) to characterize the biophysical properties of the endothelial monolayer surface. The asterisk (*) indicates a gap in the endothelial monolayer. Stiffnesses measured are outside the biological range and are represented as white squares (**D**); thus, they are removed from the analysis. (**B**) A representative image of the monolayer's height is displayed. Note that the x-y and height axes are of different scales to exaggerate the 3D topology of the layer. A representative height profile between two adjacent nuclei is also displayed. (**C**) The height difference between (b/w) nuclei and adjacent cell–cell junctions, and the distance between two adjacent nuclei, were manually extracted from AFM imaging data and are displayed as a boxplot ($n = 124$ measurements, 3 biological repeats). (**D**) Representative image showing AFM results acquired in contact mode. Nuclei (black) and EC junctions (red) manual segmentation. Scale bar: 10 μm. (**E**) The stiffness (Young's modulus) measured over the cell body and at the cell junctions was manually extracted from the AFM data and displayed as a boxplot ($n = 190$–399 measurements, >3 biological repeats). (**F, G**) Spatial relationship between fibronectin (FN) and endothelial cell–cell junctions. Endothelial cell monolayers were fixed and stained without permeabilization for FN, PECAM-1, and DAPI, then imaged using a spinning disk confocal microscope. Endothelial junctions and FN patches were automatically segmented, and the distance between the FN patches and the closest endothelial cell–cell junction was analyzed (edge-to-edge distance). (**F**) A representative max projection image and associated segmentation labels are displayed. Scale bar: 50 μm. (**G**) Frequency plot showing the number of FN patches as a function of the distance to the closest endothelial cell–cell junction. The data distribution is also shown as a boxplot ($n = 1867$ patches, 65 fields of view, one experiment). (**H, I**) The spatial relationship between FN patches and arrested PDAC cells (MIA PaCa-2 and AsPC-1). Lifeact-mScarlet-I expressing PDAC cell lines were perfused over an endothelial monolayer at 400 μm/s for 10 min, fixed, and stained to visualize FN, thrombospondin (THSD), and DAPI, then imaged using a spinning disk confocal microscope. PDAC cells and FN/THSD patches were automatically segmented, and the distance between the PDAC cells and the closest patches was analyzed (edge-to-edge distance). (**H**) Representative images showing the endothelial monolayer with attached MIA PaCa-2 cells as segmentation labels. Scale bar: 50 μm. (**I**) The proportion of PDAC cells on top, near, or away from visible FN/THSD patches is displayed as a stacked histogram ($n = 162$–2462 cells, 69–80 fields of view, 3 biological repeats). The shuffled conditions indicate that FN and THSD fluorescent channel images were randomly paired with cancer cell images from the same group before performing the distance analysis to generate randomized controls for distance measurement. (**J**) Structured illumination microscopy images of MIA PaCa-2 cells attached to an endothelial monolayer. Lifeact-mNeonGreen MIA PaCa-2 cells were perfused over an endothelial monolayer at 400 μm/s for 30 min, fixed, stained to visualize FN and F-actin, and imaged using a structured illumination microscope. Representative images of 3D renderings as well as selected projections are displayed. Scale bars: 5 μm and 1 μm. (**C, E, G**) Results are presented as boxplots, with whiskers representing the 10th to 90th percentiles and boxes indicating the interquartile range, with the median value marked. Outliers are displayed as individual data points. The P values were determined using a randomization test. (**E**) $P$ value = 0.0001. (**G**) $P$ value = 0.0001. The numerical data and images used for this figure, as well as statistical summaries including pairwise Cohen's d values and results from statistical tests, have been archived on Zenodo (https://doi.org/10.5281/zenodo.17232437).

permeabilization to label specifically the surface-accessible adhesion molecules. Notably, we observed a mosaic distribution for VCAM-1, E-selectin, CD44, ICAM-1, and fibronectin, which could support the formation of arrest hotspots. Conversely, ICAM-2 showed a more uniform distribution, mainly at cell–cell junctions. Further analysis of ICAM-1, fibronectin, and CD44 levels near arrested cancer cells indicated that about 20% of MIA PaCa-2 cells arrest on fibronectin patches or ICAM-1-positive endothelial cells, while over 95% attach to CD44-positive endothelial cells (Fig. 6D). To determine whether the frequency of arrests in CD44-positive areas exceeds chance levels, we used a Monte Carlo test: for each image, we kept the number of arrests fixed and randomly reassigned those points within the endothelial area thousands of times, recording how often they landed on CD44-positive regions to generate a "by-chance" baseline. The actual overlap was higher than this baseline (Fig. 6D), suggesting a possible correlation between the presence of CD44 on the endothelial surface and the specific locations where cancer cells arrest.

To investigate the possible mechanisms underlying PDAC cell clustering, we developed a cell adhesion simulator designed to closely replicate our experimental conditions. This simulator employs a probabilistic model for cell attachment, where the adhesion strength of PDAC cells affects the attachment probability, the adhesion properties of the endothelial background, and the flow speed (see "Methods" for details, Movie EV6). In particular, we examined both uniform and CD44-based adhesion backgrounds, where higher CD44 intensity corresponds to an increased probability of attachment (Fig. EV3C). Surprisingly, clustering was not observed at higher flow speeds in either adhesion background, regardless of PDAC cell adhesion strength, which did not align with our experimental findings (Fig. 6E; Movie EV6). This suggests that CD44 patterning alone cannot trigger the clustering we observed within our simulation parameters and assumptions.

In our movies, we observe that cells in flow can be deflected by previously attached cells, prompting us to investigate the influence of local flow dynamics on the attachment process. Consequently, we simulated a dynamic flow field, recalculating flow profiles for the simulation space using the Navier-Stokes equations after each cell attachment (Fig. EV3D; Movie EV6). Remarkably, introducing a dynamic flow field in our simulations was sufficient to induce PDAC cell clustering, irrespective of the background type used (Figs. 6E and EV3E). Together, our results suggest that while CD44 patterning alone may not initiate clustering, an endothelial receptor/ligand interaction likely facilitates initial cell attachment, with subsequent clustering driven by interactions between cancer cells and endothelial receptors, as well as alterations in local dynamic flow profiles upon cancer cell adhesion to the endothelium.

## CD44 modulates PDAC cell attachment to endothelial cell monolayers

Intrigued by the observation that most PDAC cells arrest on CD44-positive endothelial cells, we further investigated the role of CD44 in this process. Interestingly, CD44 is known to form homophilic interactions (Liu et al, 2019), and MIA PaCa-2 and AsPC-1 cells exhibited high CD44 surface expression (Fig. 6C). Western blot analyses showed differential expression of CD44 across the PDAC cell lines, with MIA PaCa-2 showing the highest CD44 protein levels, followed by AsPC-1, while Panc 10.05 had barely detectable levels (Appendix Fig. S6A). Notably, both endothelial cells, AsPC-1, and MIA PaCa-2 cells predominantly expressed the 88 kDa isoform, likely corresponding to the standard CD44s variant (Appendix Fig. S6A; Fig. EV4A). In addition, MIA PaCa-2 cells displayed a higher molecular weight CD44 isoform, likely one of the CD44v variants (Chen et al, 2018; Xu et al, 2020). These findings suggest a possible correlation between CD44 protein levels

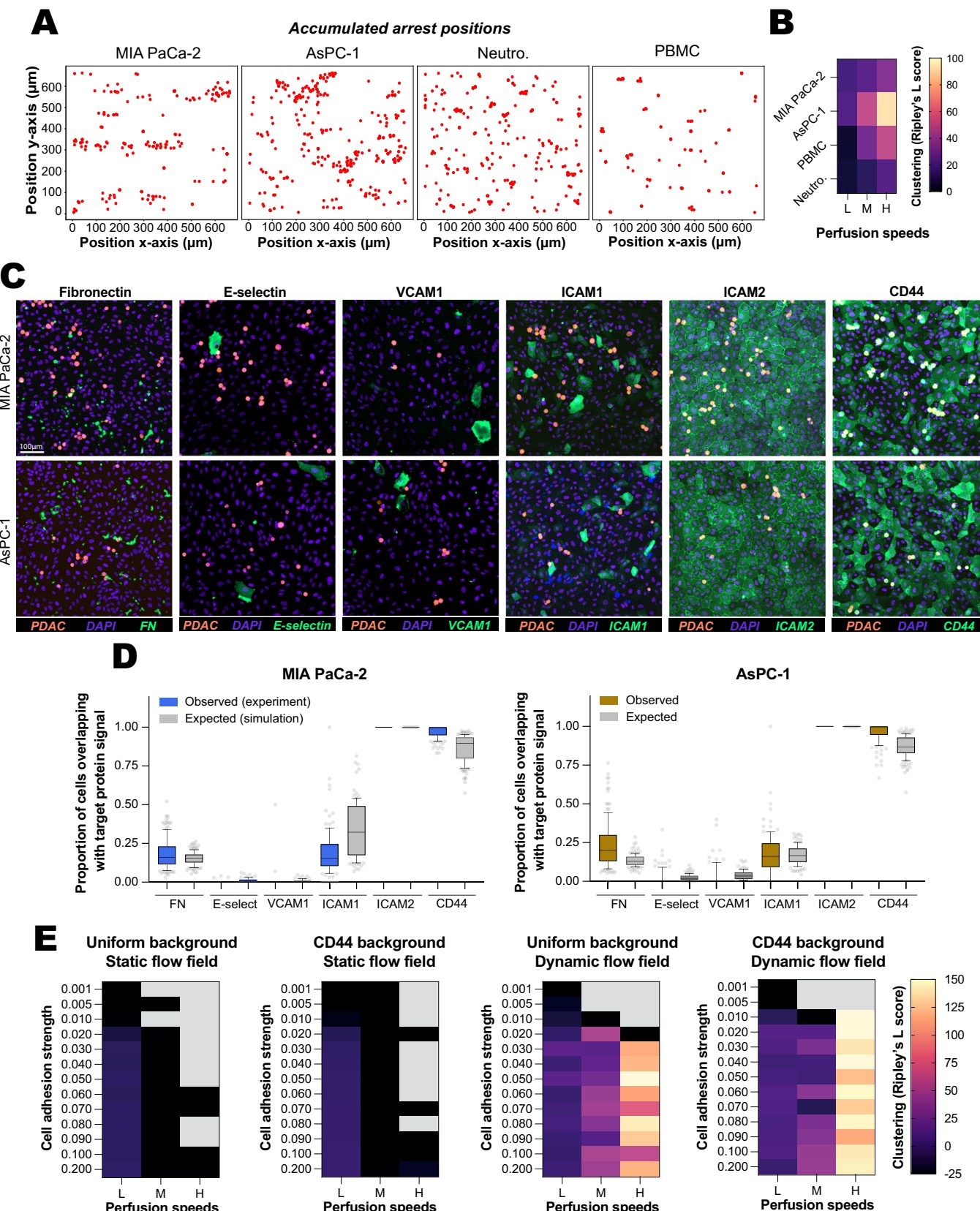

**Figure 6. PDAC cells arrest at CD44-positive hotspots on endothelial cell monolayers.**

(A, B) Spatial analysis of arrested PDAC cells and immune cells on endothelial cell monolayers. Using our tracking data (Figs. 1 and 2), we recorded all the events where cells were arrested in each field of view (instantaneous speed below 5 μm/s). Modified Ripley's L functions and Monte Carlo simulations were employed to quantify the spatial density of arrested cell events, comparing the observed density to what is expected under a random distribution. (A) Representative images showing the accumulation of arrest events in a field of view for each condition. (B) Ripley's L scores for each condition (CTRL and IL-1β are combined) are presented as a heatmap ($n = 8$–12 fields of view, 8–12 biological repeats; see "Methods"). (C, D) Spatial relationship between arrested PDAC cells (MIA PaCa-2 and AsPC-1) and endothelial adhesion molecules. Labeled PDAC cells were perfused over an endothelial monolayer for 10 min, fixed, and stained to visualize E-selectin (E-select), VCAM1, ICAM-1, ICAM-2, CD44, and fibronectin (FN). Stainings were performed without permeabilization to label specifically surface-accessible adhesion molecules. Images were captured using a spinning disk confocal microscope. (C) Representative fields of view are displayed. Scale bar: 100 μm. (D) PDAC cells were automatically segmented, and the positivity of the endothelial cells in contact with the arrested PDAC cells for each specific marker was manually scored. The percentage of PDAC cells in contact with marker-positive areas is shown (observed). The expected rate of cells in contact with each marker was calculated using Monte Carlo simulations, which took into account the cell diameter and the area of the field of view covered by each marker (see "Methods" for details) ($n = 71$–143 fields of view, 3 biological replicates). (E) Simulations conducted using our cell adhesion simulator to explore the factors contributing to PDAC cell clustering. We examined the effects of flow speed, PDAC adhesion strength, and the characteristics of the endothelial adhesion background. The uniform background features a constant attachment probability throughout the simulation space, while the CD44 background utilizes CD44 images to modulate attachment probabilities spatially. A static flow field indicates that cell attachment does not influence flow dynamics. In contrast, a dynamic flow field indicates that the local flow dynamics are recomputed across the whole simulation space by solving the Navier-Stokes equations after each attachment event. For each parameter tested, Ripley's L score was calculated only when at least two cells were attached in the simulation space. Results are presented as heatmaps. (B, D) Results are presented as boxplots, with whiskers extending from the 10th to the 90th percentiles. The boxes capture the interquartile range, with the median marked by a line within each box. Data points outside the whiskers are depicted as individual dots. The numerical data and images used for this figure, as well as statistical summaries including pairwise Cohen's *d* values and results from statistical tests, have been archived on Zenodo (https://doi.org/10.5281/zenodo.17232437).

and the attachment capabilities of these PDAC cell lines, indicating that CD44 expression may play a pivotal role in mediating their adhesion to the endothelial monolayer.

To investigate the role of CD44 in mediating interactions between PDAC cells and endothelial cells under flow conditions, we employed siRNAs targeting all CD44 isoforms to silence CD44 gene expression. Silencing CD44 in endothelial cells resulted in an approximately 50% reduction in CD44 expression (Fig. EV4A–C). However, this did not consistently impact the attachment of MIA PaCa-2 and AsPC-1 cells to the endothelial monolayers (Fig. EV4D–I; Movie EV7), possibly due to the incomplete loss of CD44. In contrast, silencing CD44 in PDAC cells was more efficient, resulting in a reduction of approximately 90% in overall CD44 expression (Fig. EV5A–C). Notably, this significant reduction in CD44 expression impaired the PDAC cells' ability to form clusters when kept in suspension (Movie EV8). It also nearly abolished PDAC cells' capacity to attach to endothelial cell monolayers (Figs. 7A–C and EV5D,E; Movie EV8).

To further explore the role of CD44 in PDAC cell arrest on endothelial monolayers, we employed a monoclonal pan-anti-CD44 function-blocking antibody (HERMES-1, which recognizes the N-terminal hyaluronate-binding domain of CD44; Jalkanen et al, 1986). Cancer and/or endothelial cells were pre-incubated with the CD44-blocking antibody for 10 min before perfusion. Interestingly, treatment of the endothelial monolayer with a 5 μg/ml antibody concentration led to a significant disruption of the endothelial monolayer, prompting us to use a lower concentration (1.5 μg/ml) (Fig. EV5F). Incubating the endothelial monolayer with the lower antibody concentration did not disturb the monolayer, nor did it impact PDAC cell attachment. In contrast, treatment of PDAC cells with 5 μg/ml of anti-CD44 antibody impaired their ability to form cell clusters in suspension. It nearly abolished their ability to attach to the endothelial layer, consistent with the results obtained with RNAi (Figs. 7D–F and EV5G,H; and Movie EV9).

Intriguingly, when CD44 was targeted in PDAC cells using either siRNA or the anti-CD44 antibody, the few cells that did manage to attach exhibited stronger adhesion, as evidenced by shorter distances traveled upon landing and fewer speed peaks.

These cells also demonstrated more efficient clustering patterns on the endothelial monolayer than control cells (Fig. 7G–L). Altogether, our findings indicate that CD44 mediates the attachment of PDAC cells to endothelial cells under flow conditions, particularly on the side of the cancer cells.

## Hyaluronic acid modulates PDAC cell attachment to endothelial cell monolayers

CD44 is known to form homophilic interactions, but it also binds to a variety of ligands, including hyaluronic acid (HA), and other ECM molecules such as osteopontin, collagens, fibronectin, and laminin (Chen et al, 2018; Jalkanen and Jalkanen, 1992). Since CD44 and HA are expressed on the surfaces of MIA PaCa-2, AsPC-1, and endothelial cells (Fig. 8), we next investigated whether HA contributes to the attachment of PDAC cells to endothelial cells. We treated PDAC or endothelial cells with hyaluronidase to reduce HA levels, effectively depleting HA at the cell surface (Fig. 8A,B).

Significantly, treatment of either cancer cells or endothelial cell monolayers with hyaluronidase almost completely abolished the ability of AsPC-1 and MIA PaCa-2 cells to attach to the endothelial monolayer under flow conditions (Fig. 8C–E; Movie EV10). This finding indicates that HA, whether on the surface of endothelial or PDAC cells, plays a significant role in facilitating the arrest of PDAC cells.

Interestingly, similar to the anti-CD44 treatments, the few cancer cells attaching after HA targeting exhibited stronger adhesion, as demonstrated by shorter distances traveled upon landing and fewer speed peaks (Fig. 8F–H). These cells also showed more efficient clustering on the endothelial monolayer than control cells. Altogether, our data indicate that both CD44 and HA are crucial for the ability of PDAC cells to adhere to endothelial cell monolayers.

## Discussion

In this study, we developed an imaging framework, called FlowVision, that combines microfluidics, fast live-imaging, and

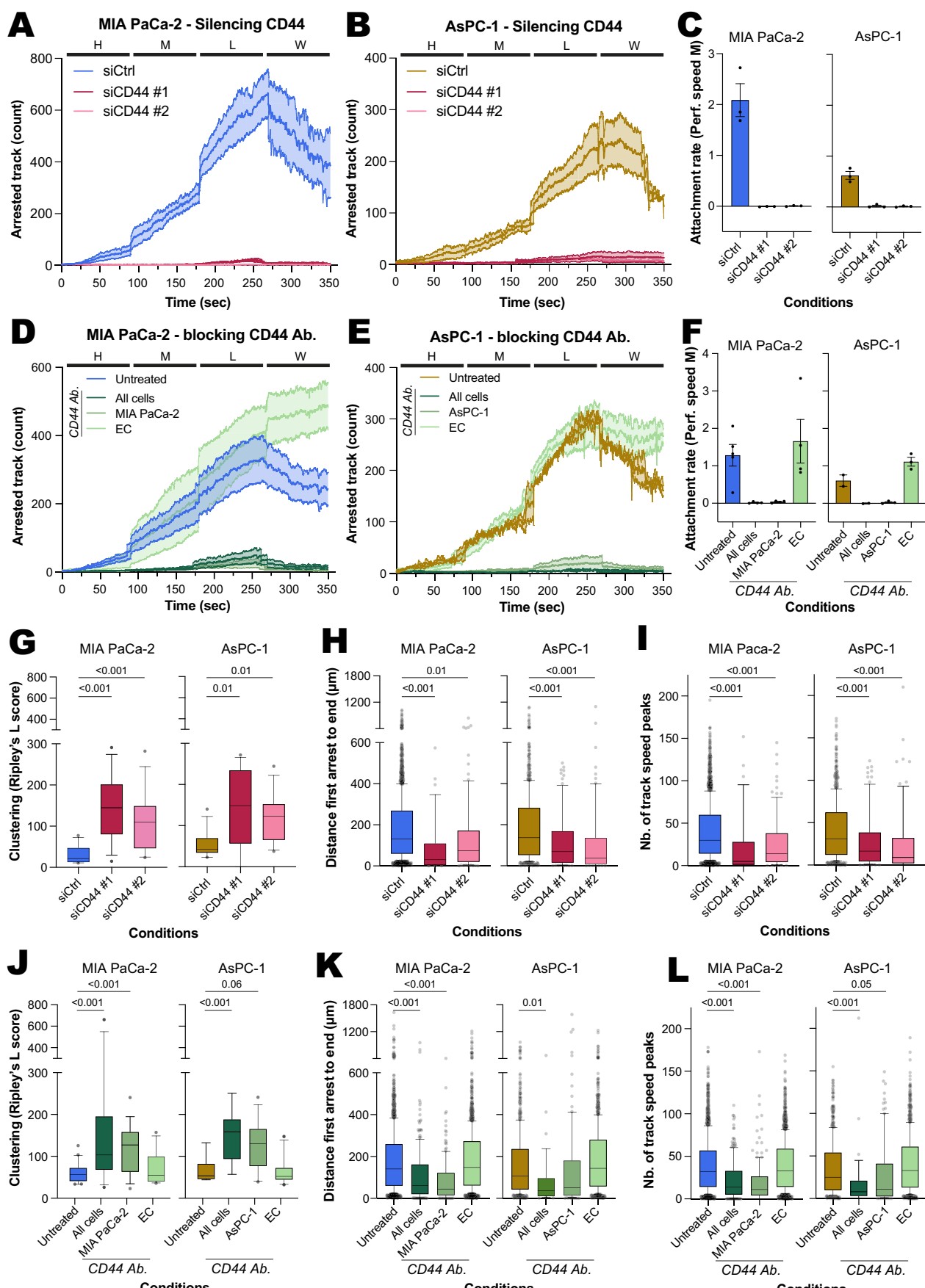

**Figure 7. CD44 modulates PDAC cell attachment to endothelial cell monolayers.**

(A–C) Effect of CD44 silencing in PDAC cells on PDAC cell attachment to the endothelial monolayer. MIA PaCa-2 and AsPC-1 cells were treated with either CD44-targeting siRNAs (siCD44 #1 or siCD44 #2) or control siRNA (siCTRL) and then perfused over endothelial monolayers. Cell attachment was recorded as described in Fig. 1. (D–F) Effect of anti-CD44 blocking antibody treatment on PDAC cell attachment to the endothelial monolayer. PDAC and/or endothelial cells were pre-treated with an antibody that blocks CD44 function. PDAC cells were then perfused over endothelial monolayers, and cell attachment was recorded as in Fig. 1. The number of arrested MIA PaCa-2 (A, D) and AsPC-1 (B, E) cells over time is presented, with bold lines indicating the average and shaded areas of the SD (2–5 biological repeats, see "Methods"). The data are shown for different flow speeds. (C, F) The attachment rate for each cell line and condition at the medium (M) flow speed is displayed as a bar chart (mean ± SEM) with individual data points (2–5 biological repeats, see "Methods"). (G–I) Effect of CD44 silencing in PDAC cells on the spatial clustering of PDAC cells. (G) Using the tracking data from (A–C) and as in Fig. 6B, we recorded all arrest events within each field of view. Modified Ripley's L functions were applied to quantify the spatial density of arrested cells. Ripley's L scores for each condition are presented as boxplots ($n = 12$ fields of view, 3 biological repeats, see "Methods"). (H, I) Similar to Fig. 3, the analysis was refined to include only cell trajectories showing a definitive arrest pattern. (H) The total distance traveled from the point of arrest to the end of the track is displayed for each condition. (I) The number of detachment peaks per condition is shown ($n = 35$–2172 tracks). (J–L) Effect of anti-CD44 blocking antibody on the spatial clustering of PDAC cells. (J) Using the tracking data from (D, E) and as in Fig. 6B, all arrest events within each field of view were recorded. Modified Ripley's L functions were applied to quantify the spatial density of arrested cells. Ripley's L scores for each condition are presented as boxplots ($n = 8$–20 fields of view, 2–5 biological repeats, see "Methods"). (K, L) As in Fig. 3, the analysis was refined to include only cell trajectories showing a definitive arrest pattern. (K) The total distance traveled from the point of arrest to the end of the track is displayed for each condition. (L) The number of detachment peaks per condition is shown ($n = 35$–1721 tracks). (G–L) Results are presented as boxplots, with whiskers extending from the 10th to the 90th percentiles. The boxes capture the interquartile range, with the median marked by a line within each box. Data points falling outside the whiskers are depicted as individual dots. The P values were determined using a randomization test. (G) MIA PaCa-2. siCtrl vs siCD44#1, P value = 0.0001. siCtrl vs siCD44#2, P value = 0.0001. (H) MIA PaCa-2. siCtrl vs siCD44#1, P value = 0.0001. AsPC-1. siCtrl vs siCD44#1, P value = 0,0001. siCtrl vs siCD44#2, P value = 0.0001. (I) MIA PaCa-2. siCtrl vs siCD44#1, P value = 0.0001. siCtrl vs siCD44#2, P value = 0,0001. AsPC-1. siCtrl vs siCD44#1, P value = 0,0001. siCtrl vs siCD44#2, P value = 0.0001. (J) MIA PaCa-2. Untreated vs All cells, P value = 0.0001. Untreated vs MIA PaCa-2, P value = 0.0001. Untreated vs EC, P value = 0.34. AsPC-1. Untreated vs All cells, P value = 0.0001. (K) MIA PaCa-2. Untreated vs All cells, P value = 0.0001. Untreated vs MIA PaCa-2, P value = 0.0001. (L) MIA PaCa-2. Untreated vs All cells, P value = 0.0001. Untreated vs MIA PaCa-2, P value = 0.0001. AsPC-1. Untreated vs All cells, P value = 0.0001. The numerical data and images used for this figure, as well as statistical summaries including pairwise Cohen's d values and results from statistical tests, have been archived on Zenodo (https://doi.org/10.5281/zenodo.17232437).

single-cell tracking analysis to investigate how circulating cells interact with an endothelial monolayer under capillary-like flow conditions.

Capturing dynamic events, such as the arrest of immune and cancer cells under flow, is challenging using traditional fluorescence microscopy due to the speed and unpredictability of the arrest event, as well as the potential phototoxic effects of continuous laser imaging, which can alter cell physiology (Del Rosario et al, 2025). To overcome these challenges, we developed a framework that combines fast label-free imaging, deep-learning-based segmentation, artificial labeling (Elmalam et al, 2024), and single-cell tracking to phenotype cell behaviors. This approach builds on previous work, such as Traject3d and CellPhe, which have pioneered methods for analyzing cellular dynamics in complex environments through live imaging and advanced tracking techniques (Wiggins et al, 2023; Freckmann et al, 2022). The current framework focuses on the arrest phase of the interaction with endothelial monolayers, and future work will expand this framework to analyze the later stages of the interaction, including extravasation in vitro and in vivo. By leveraging widely accessible, open-source tools and software, our pipeline is fully adaptable for diverse research applications. Furthermore, we have made all code, deep learning models, and training datasets available to the scientific community (Laine et al, 2021). Therefore, the tools established here will support the broader study of circulating cell-endothelial monolayer interactions under flow conditions.

Using our framework, we report that PDAC and immune cells frequently arrest near endothelial cell junctions. Our findings align with previous studies, which show that PDAC cells tend to arrest near tricellular junctions (Nakai et al, 2005). However, whether this arrest is random or directed remains unclear. We also observed that, in vivo, cancer cells are always positioned near junctions. Three critical consequences of this proximity are a favorable topology and increased stiffness, which may favor more stable adhesion and the increased accessibility of the underlying ECM, which could facilitate transendothelial migration (Resnikoff and

Schwarzbauer, 2024; Chen et al, 2016). Future work will explore how transendothelial migration is regulated and whether other factors contribute to the preferential arrest of immune and cancer cells near endothelial cell junctions.

We found that PDAC and immune cells exhibit similar behavior when arresting on endothelial cell monolayers, frequently localizing near endothelial cell junctions and halting at specific hotspots on the endothelial surface. Interestingly, while leukocytes are known to adhere and transmigrate at ICAM-1-rich hotspots (Grönloh et al, 2022), only a small proportion (20%) of PDAC cells arrested on ICAM-1-positive endothelial cells, suggesting distinct mechanisms for PDAC cell versus immune cell adhesion. Similarly, PDAC cells differ from other cancers, such as colon cancer cell models, in which the central role of E-selectin was established (Laferrière et al, 2001). Previous studies have also suggested that breast cancer CTCs arrest on endothelial cells are promoted by apical endothelial fibronectin and require integrin-mediated adhesion (Osmani et al, 2019; Barbazán et al, 2017). However, in our study, only 20% of arrested PDAC cells were found in contact with fibronectin patches. Moreover, structured illumination microscopy revealed that most fibronectin patches were located basally, rather than apically, which limits their direct role in mediating adhesion. Nevertheless, in cases where apical endothelial fibronectin is present, it is likely to contribute to CTC attachment (Barbazán et al, 2017; Malik et al, 2010).

We demonstrate that targeting CD44 in PDAC cells effectively disrupts their interaction with endothelial cells. Our findings underscore the pivotal role of CD44 in pancreatic cancer metastasis, where it is highly expressed and closely linked to poor prognosis and metastatic progression. Previous studies have shown that targeting CD44 can reduce pancreatic tumor growth and metastasis, highlighting its potential as a therapeutic target (Chen et al, 2018; Xu et al, 2020). Notably, CD44 also facilitates the attachment of breast cancer cells to endothelial cells, suggesting a broader role for CD44 in mediating the attachment of cancer cells to endothelial cells (Osmani et al, 2019; Baccelli et al, 2013; Boral et al, 2017).

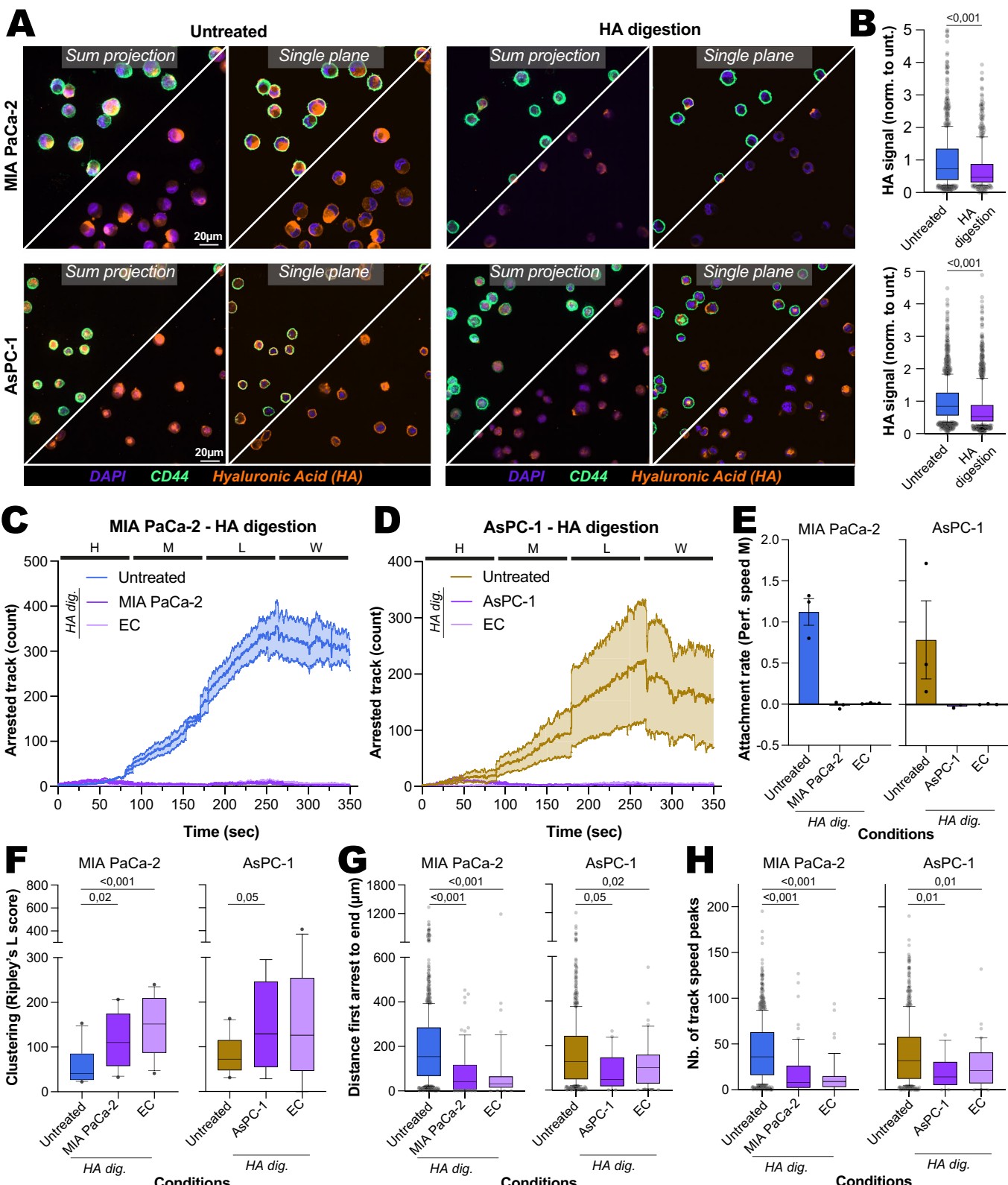

**Figure 8. HA mediates PDAC cell attachment to endothelial cell monolayers.**

(A, B) Validation of hyaluronidase (HA digestion; HA dig.) treatment in PDAC cells. MIA PaCa-2 and AsPC-1 were treated with hyaluronidase and plated on poly-Lysine-coated coverslip, fixed, stained, and imaged using a spinning disk confocal microscope. (A) Representative SUM projections and single planes are displayed. Note that the Hyaluronic signal in the digested sample is inside the cell. (B) The HA signal per cell was quantified from the SUM projections ($n = 601$–1542, 3 biological repeats). Scale bar: 20 µm. (C–E) Impact of hyaluronidase treatments in PDAC cells or endothelial cells (ECs) on PDAC cell attachment to the endothelial monolayer. PDAC or ECs were pre-treated with hyaluronidase. PDAC cells were then perfused over endothelial monolayers, and cell attachment was recorded as in Fig. 1. The number of arrested MIA PaCa-2 (C) and AsPC-1 (D) cells over time is displayed, with bold lines indicating the average and shaded areas representing the SD (2–3 biological repeats, see "Methods"). The data is shown for different flow speeds. (E) The attachment rate for each cell line and condition at the medium (M) flow speed is displayed as a bar chart (mean ± SEM) with individual data points (2–3 biological repeats, see "Methods"). (F–H) Effect of hyaluronidase treatments on the spatial clustering of PDAC cells. (F) Using the tracking data from (C–E) and as in Fig. 6B, we recorded all arrest events within each field of view. Modified Ripley's L functions were applied to quantify the spatial density of arrested cells. Ripley's L scores for each condition are presented as boxplots ($n = 8$–12 fields of view, 2–3 biological repeats, see "Methods"). (G, H) As in Fig. 3, the analysis was refined to include only cell trajectories showing a definitive arrest pattern. (G) The total distance traveled from the point of arrest to the end of the track is displayed for each condition. (H) The number of detachment peaks per condition is shown ($n = 20$–1055 tracks). (B, F–H) Results are presented as boxplots, with whiskers extending from the 10th to the 90th percentiles. The boxes capture the interquartile range, with the median marked by a line within each box. Data points falling outside the whiskers are depicted as individual dots. The $P$ values were determined using a randomization test. (B) MIA PaCa-2, $P$ value = 0.0001. AsPC-1, $P$ value = 0.0001. (F) MIA PaCa-2, Untreated vs EC, $P$ value = 0.0001. (G) MIA PaCa-2, Untreated vs MIA PaCa-2, $P$ value = 0.0001. Untreated vs EC, $P$ value = 0.0001. (H) MIA PaCa-2, Untreated vs MIA PaCa-2, $P$ value = 0.0001. Untreated vs EC, $P$ value = 0.0001. The numerical data and images used for this figure, as well as statistical summaries including pairwise Cohen's $d$ values and results from statistical tests, have been archived on Zenodo (https://doi.org/10.5281/zenodo.17232437).

CD44 is known to mediate homophilic interactions (Liu et al, 2019) and bind to a variety of ligands, including HA, osteopontin, serglycin, collagens, fibronectin, and laminin (Chen et al, 2018). We found that disrupting HA on endothelial cells nearly abolished PDAC cell adhesion to the monolayer, suggesting a mechanism where CD44 on cancer cells interacts with HA on endothelial cells. However, we also observed that removing HA from cancer cells had a similar effect, abolishing PDAC adhesion. This dual dependency suggests a more complex interplay between CD44 and HA in cancer and endothelial cells, indicating that these molecules may participate in more intricate and cooperative interactions than previously understood. Future work will aim to dissect the molecular details of how CD44-HA interactions are regulated at the adhesion interface and explore how these mechanisms contribute to the transendothelial migration of PDAC cells.

### Limitations of the study

We used HUVECs to model the interactions between PDAC cells and endothelium. Although HUVECs are venous in origin and may not fully capture the features of liver- or lung-derived endothelial cells, which are common metastatic sites in PDAC (diSibio and French, 2008), they are widely adopted in microfluidic platforms due to their robustness and suitability for in vitro studies. Past work demonstrates that HUVEC-based observations can be translated in vivo (Santio et al, 2024; Follain et al, 2018a; Osmani et al, 2019).

Our microfluidic system does not replicate the full complexity of in vivo vascular geometry, such as circumferential vessel walls, where cancer cells may arrest through passive occlusion. However, eliminating potential physical occlusion in our model is a necessary trade-off to focus on studying the mechanisms of active adhesion between cancer and endothelial cells. In addition, while the flow speed values used in our study are comparable to those found in capillaries, the shear stresses experienced by the cells are lower due to the flat geometry of our monolayer. Nonetheless, our system offers precise flow control and scalability for quantitative analysis of the effects of molecular interventions. It also enables high-resolution imaging, making it ideal for investigating the active mechanisms governing cancer cell adhesion to endothelial cells under flow.

Adapting FlowVision segmentation models to new imaging settings or cell/monolayer combinations might require a quick re-annotation and fine-tuning step (usually involving about 40–60 images and taking 6–8 h from start to finish). We see this as a practical and manageable limitation, which is balanced by the benefits in scale and reproducibility.

## Methods

**Reagents and tools table**

| Reagent/resource | Reference or source | Identifier or catalog number |
|---|---|---|
| **Experimental models** | | |
| AsPC-1 cells (*H. sapiens*) | ATCC | ATCC-CRL-1682 |
| BxPC-3 cells (*H. sapiens*) | ATCC | ATCC-CRL-1687 |
| MIA PaCa-2 cells (*H. sapiens*) | ATCC | ATCC-CRL-1420 |
| PANC-1 cells (*H. sapiens*) | ATCC | ATCC-CRL-1469 |
| Panc 10.05 cells (*H. sapiens*) | ATCC | ATCC-CRL-2547 |
| SU.86.86 cells (*H. sapiens*) | ATCC | ATCC-CRL-1837 |
| SW 1990 cells (*H. sapiens*) | ATCC | ATCC-CRL-2172 |
| HUVECs (*H. sapiens*) | PromoCell | C-12203 |
| HEK293FT (*H. sapiens*) | ATCC | ATCC-CRL-1573 |
| Neutrophils (*H. sapiens*) | Isolated from donor blood | N/A |
| PBMC (*H. sapiens*) | Isolated from donor blood | N/A |
| **Recombinant DNA** | | |
| pENTR2b-Lifeact-mNeonGreen | This study | N/A |
| pENTR2b-Lifeact-mScarlet-I | This study | N/A |
| pLenti6.3/TO/V5-DEST | ThermoFisher Scientific | V53306 |
| pENTR2b | ThermoFisher Scientific | A10463 |
| pLenti6.3/TO/V5-DEST-Lifeact-mNeonGreen | This study | N/A |

| Reagent/resource | Reference or source | Identifier or catalog number |
|---|---|---|
| pLenti6.3/TO/V5-DEST-Lifeact-mScarlet-I | This study | N/A |
| pMD2.G | Addgene | 12259 |
| pMDLg/pRRE | Addgene | 12251 |
| pRSV-Rev | Addgene | 12253 |
| **Antibodies** | | |
| Anti-fibronectin | Sigma-Aldrich | F3648 |
| Anti-thrombospondin-1 | Abcam | ab1823 |
| Anti-PECAM-1 | Invitrogen | 37-0700 |
| Anti-ICAM-1 | Invitrogen | MA5-11569 |
| Anti-ICAM-2 | Invitrogen | 14-1029-82 |
| Anti-CD44 (HERMES-3) | Abcam | ab254530 |
| Anti-GAPDH | HyTest | 5G4MaB6C5 |
| Anti-VCAM-1 | Abcam | ab316314 |
| Anti-E-selectin | Abcam | ab300557 |
| Anti-alpha-tubulin | Abcam | ab18251 |
| Alexa Fluor 647-conjugated anti-mouse IgG | Invitrogen | A21235 |
| Alexa Fluor 488-conjugated anti-mouse IgG | Invitrogen | A11001 |
| Alexa Fluor 568-conjugated anti-mouse IgG | Invitrogen | A10037 |
| Alexa Fluor 488-conjugated anti-rat IgG | Invitrogen | A21208 |
| Alexa Fluor 568-conjugated anti-rat IgG | Invitrogen | A11077 |
| Alexa Fluor 647-conjugated anti-rabbit IgG | Invitrogen | A21244 |
| Alexa Fluor 488-conjugated anti-rabbit IgG | Invitrogen | A11008 |
| Alexa Fluor 568-conjugated anti-rabbit IgG | Invitrogen | A10042 |
| Goat anti-mouse 650 | Azure Biosystems | AC2166 |
| Anti-CD44 blocking antibody (HERMES-1) | DSHB | AB_528148 |
| **Oligonucleotides and other sequence-based reagents** | | |
| TaqMan pan CD44 primers | ThermoFisher Scientific | Hs01075864_m1 |
| TaqMan GAPDH primers | ThermoFisher Scientific | Hs02786624_g1 |
| Negative Control siRNA (endothelial cells) | ThermoFisher Scientific | 4390843 |
| siCD44 #1 (endothelial cells) | ThermoFisher Scientific | s2681 |
| siCD44 #2 (endothelial cells) | ThermoFisher Scientific | s2682 |
| AllStars Negative Control siRNA | Qiagen | 1027281 |
| siCD44 #1 (PDAC) | Sigma-Aldrich | custom-designed, GCG CAG AUC GAU UUG AAU A |

| Reagent/resource | Reference or source | Identifier or catalog number |
|---|---|---|
| siCD44 #2 (PDAC) | Sigma-Aldrich | custom-designed, CCG CUU UGC AGG UGU AUU C |
| **Chemicals, enzymes, and other reagents** | | |
| Endothelial cell growth medium (ECGM) | PromoCell | C-22010 and C-39215 |
| Penicillin–streptomycin | Sigma-Aldrich | P0781 |
| Fetal bovine serum (FBS) | BioWest | S1810 |
| L-glutamine | Sigma-Aldrich | G7513 |
| Human insulin | Sigma-Aldrich | I0908 |
| DMEM high glucose | Sigma-Aldrich | D6171-500ML |
| RPMI1640 | Sigma-Aldrich | R5886-100ML |
| CellTrace | ThermoFisher Scientific | C34570 |
| DAPI | ThermoFisher Scientific | D1306 |
| Alexa Fluor 647-conjugated phalloidin | Invitrogen | A30107 |
| Poly-D-Lysine | Gibco | A3890401 |
| fibronectin | Sigma-Aldrich | 341631 |
| IL-1β | R&D Systems | 201-LB |
| OptiMEM | Gibco | 31985070 |
| Lipofectamine™ RNAiMAX | ThermoFisher Scientific | 13778150 |
| Kpn1 | ThermoFisher Scientific | FD0524 |
| Xhol | ThermoFisher Scientific | FD0694 |
| LR clonase II | Invitrogen | 56485 |
| Lipofectamine 3000 | ThermoFisher Scientific | 100022052 |
| Blasticidin | Gibco | A11139-03 |
| Percoll gradients | Sigma-Aldrich | p1644 |
| Complete protease inhibitors mix | Roche | 05056489001 |
| PhosStop phosphatase inhibitors mix | Roche | 04906837001 |
| GENIUS Nuclease | Santa Cruz Biotechnology | sc-391121 |
| DC protein assay | BioRad | 5000113; 5000114; 5000115 |
| PVDF membrane | Millipore | IPFL00005 |
| AdvanBlock-Fluor blocking solution | Advansta | R-03729-E10 |
| NucleoSpin RNA Plus kit | Macherey-Nagel | 740984.250 |
| High-Capacity cDNA Reverse Transcription Kit | Applied Biosystems | 4368814 |
| Paraformaldehyde | ThermoFisher Scientific | 28908 |
| ProLong Glass media | Invitrogen | P36984 |
| Glass-bottom ibidi microfluidic channels | ibidi | 80177 |
| Vectashield | Vector Laboratories | H-1000-10 |
| HA-binding protein | This study | N/A |
| Alexa 594-conjugated streptavidin | Invitrogen | S11227 |

| Reagent/resource | Reference or source | Identifier or catalog number |
|---|---|---|
| 1-channel μ-slides LUER 0.4 | ibidi | 80176 |
| 6-channel μ-slides LUER 0.4 | ibidi | 80606 |
| Hyaluronidase | Sigma-Aldrich | H3506 |
| HEPES | Sigma-Aldrich | H0887 |
| **Software** | | |
| QuantStudio Design & Analysis Software version 1.5.2 | https://www.thermofisher.com/fi/en/home/technical-resources/software-downloads/quantstudio-3-5-real-time-pcr-systems.html | |
| SlideBook 6 | https://www.intelligent-imaging.com/slidebook | |
| Zen Black | https://www.micro-shop.zeiss.com/en/us/softwarefinder/software-categories/zen-black/zen-black-system/ | |
| NIS-Elements | https://www.microscope.healthcare.nikon.com/fr_EU/products/software/nis-elements | |
| Fiji | https://fiji.sc/ | |
| Cellpose | https://github.com/MouseLand/cellpose | |
| StarDist | https://github.com/stardist/stardist | |
| DiAna plugin | https://imagej.net/plugins/distance-analysis | |
| FlowVision | This study, https://github.com/CellMigrationLab/PDAC_DL | |
| BIOP LaRoMe plugin | https://github.com/BIOP/ijp-LaRoMe | |
| ZeroCostDL4Mic | https://github.com/HenriquesLab/ZeroCostDL4Mic | |
| DL4MicEverywhere | https://github.com/HenriquesLab/DL4MicEverywhere | |
| Arivis Vision4D | https://www.zeiss.com/microscopy/en/products/software/arivis-pro.html | |
| Adhesion Flow Simulator | This study, https://github.com/CellMigrationLab/AdhesionFlowSimulator | |
| CellTracksColab | https://github.com/CellMigrationLab/CellTracksColab | |
| GraphPad Prism | https://www.graphpad.com/features | |
| Adobe Illustrator | https://www.adobe.com/products/illustrator.html | |
| **Other** | | |

| Reagent/resource | Reference or source | Identifier or catalog number |
|---|---|---|
| Masterflex Ismatec peristaltic pump | Ismatec | 78018-24 |
| Nikon Eclipse Ti2-E | Nikon | |
| NanoDrop Lite Spectrophotometer | ThermoFisher Scientific | |
| Sapphire Biomolecular Imager | Azure Biosystems | |
| QuantStudio 3 | ThermoFisher Scientific | A28567 |
| Marianas spinning-disk imaging system with a Yokogawa CSU-W1 scanning unit on an inverted Zeiss Axio Observer Z1 | Intelligent Imaging Innovations | |
| DeltaVision OMX v4 | GE Healthcare Life Sciences | |
| LSM880 | Zeiss | |

## Cells and cell culture

Pancreatic ductal adenocarcinoma (PDAC) cell lines AsPC-1 (ATCC, CRL-1682), BxPC-3 (ATCC, CRL-1687), MIA PaCa-2 (ATCC, CRL-1420), PANC-1 (ATCC, CRL-1469), Panc 10.05 (ATCC, CRL-2547), SU.86.86 (ATCC, CRL-1837), and SW 1990 (ATCC, CRL-2172) were cultured in standard conditions on 10 cm plastic culture dishes using cell-specific recommended media (ATCC). The basal media used were DMEM high glucose (DMEM HG) for MIA PaCa-2, PANC-1, and SW 1990, or RPMI1640 for BxPC-3, SU.86.86, and Panc 10.05. All media were supplemented with 10% fetal bovine serum (FBS) (BioWest, S1810), 1% L-glutamine (Sigma-Aldrich, G7513), and 1% penicillin–streptomycin (Sigma-Aldrich, P0781). The Panc 10.05 media was also supplemented with 10 U/mL of human insulin (Sigma-Aldrich, I0908). PDAC cell lines were authenticated at the beginning of the project by the Leibniz Institute DSMZ (Deutsche Sammlung von Mikroorganismen und Zellkulturen) using STR profiling.

HEK293FT (ATCC, CRL-1573) cells were cultured in DMEM HG, supplemented with 10% FBS, 1% L-glutamine, and 1% penicillin–streptomycin. Human Umbilical Vein Endothelial Cells (HUVECs) (PromoCell, C-12203) were grown in ready-to-use Endothelial Cell Growth Medium (ECGM) (PromoCell, C-22010 and C-39215), supplemented with 1% penicillin–streptomycin (Sigma-Aldrich, P0781). Primary endothelial cells from "P0" (commercial vial) were expanded to a P3 stock and stored at −80 °C until used to standardize the experimental replicates. Vials were thawed, and cells were transferred to a 10-cm culture dish, where they were grown for at least 2 days.

All cell lines were routinely tested for mycoplasma infections and were found to be free of mycoplasma infections.

## Antibodies and reagents

The following primary antibodies were used in this study for immunofluorescence (IF) and Western blot (WB) analysis: anti-fibronectin (FN) (1:200 for IF, Sigma-Aldrich, F3648), anti-thrombospondin (1:200 for IF, Abcam, ab1823), anti-PECAM-1

(1:200 for IF, Invitrogen, 37-0700), anti-ICAM-1 (1:200 for IF, Invitrogen, MA5-11569), anti-ICAM-2 (1:200 for IF, Invitrogen, 14-1029-82), anti-CD44 (1:200 for IF and 1:1000 for WB, HERMES-3, Abcam, ab254530), anti-GAPDH (1:1000 for WB, HyTest, 5G4MaB6C5) anti-VCAM-1 (1:200, Abcam ab316314), anti-E-selectin (1:200, Abcam ab300557) and anti-alpha-tubulin (1:1000 for WB, Abcam, ab18251).

The secondary antibodies used in 1:400 dilution for immuno-fluorescence, all from Invitrogen, include Alexa Fluor 647-conjugated anti-mouse IgG (A21235), Alexa Fluor 488-conjugated anti-mouse IgG (A11001), Alexa Fluor 568-conjugated anti-mouse IgG (A10037), Alexa Fluor 488-conjugated anti-rat IgG (A21208), Alexa Fluor 568-conjugated anti-rat IgG (A11077), Alexa Fluor 647-conjugated anti-rabbit IgG (A21244), Alexa Fluor 488-conjugated anti-rabbit IgG (A11008), and Alexa Fluor 568-conjugated anti-rabbit IgG (A10042).

For western blot analysis, secondary antibodies included goat anti-mouse 650 (Azure Biosystems, AC2166) at 1:3000 dilution, Alexa Fluor 647-conjugated anti-mouse IgG (Invitrogen, A21235), Alexa Fluor 488-conjugated anti-mouse IgG (Invitrogen, A11001), Alexa Fluor 647-conjugated anti-rabbit IgG (Invitrogen, A21244), and Alexa Fluor 488-conjugated anti-rabbit IgG (Invitrogen, A11008), all used at 1:5000 dilution. The anti-CD44 blocking antibody used was HERMES-1 (DSHB, AB_528148).

Additional reagents used in this study include CellTrace (ThermoFisher Scientific, C34570), DAPI (ThermoFisher Scientific, D1306), Alexa Fluor 647-conjugated phalloidin (Invitrogen, A30107), Poly-D-Lysine (Gibco™, A3890401), fibronectin (Sigma-Aldrich, 341631), and IL-1β (R&D Systems, 201-LB).

## siRNA-mediated gene silencing

For endothelial cells, siRNAs used were siCTRL (Negative Control siRNA, 4390843), siCD44 #1 (s2681, UAU UCC ACG UGG AGA AAA A), siCD44 #2 (s2682, GCG CAG AUC GAU UUG AAU A), and siCD44 # 3 (s2683, GAA UAU AAC CUG CCG CUU U), all from ThermoFisher Scientific (4427038). For PDAC cells, the siRNAs used were siCTRL (AllStars Negative Control siRNA, Qiagen, 1027281), siCD44 #1 (custom-designed, GCG CAG AUC GAU UUG AAU A), and siCD44 #2 (custom-designed, CCG CUU UGC AGG UGU AUU C), all from Sigma-Aldrich. SiRNAs were mixed with Lipofectamine™ RNAiMAX Transfection Reagent (ThermoFisher Scientific, 13778150) according to the manufacturer's instructions in OptiMEM (Gibco, 31985070). The transfection mixture was added to the cells and left for 2 h. After 2 h, twice the amount of ECGM was added, and the cells were incubated for an additional 72 h before use. For PDAC cells, the transfection mixture was added and left for 24 h. After 24 h, the media were changed to DMEM HG, supplemented with 10% FBS, 1% L-glutamine, and 1% penicillin–streptomycin. Cells were left for an additional 48 h and a total of 72 h after transfection before use.

## Production of lifeact cell lines

Lifeact-expressing cell lines were generated using a third-generation lentiviral system. Expression plasmids encoding Lifeact fused to fluorescent markers were constructed via Gateway cloning. First, gene blocks containing the lifeact sequence (Riedl et al, 2008) in frame with either mNeonGreen (Shaner et al, 2013) or mScarlet-I

(Bindels et al, 2017) and flanked with Kpn1 and Xho1 restriction sites were purchased from IDT (Integrated DNA Technologies, Inc.). Through enzymatic digestion (Kpn1, ThermoFisher Scientific, FD0524, and XhoI, ThermoFisher Scientific, FD0694) of the two sites on both gene blocks and the pENTR2b entry vector (ThermoFisher Scientific, A10463), we obtained pENTR2b-Lifeact-mNeonGreen and pENTR2b-Lifeact-mScarlet-I. Second, LR reaction (LR clonase II (Invitrogen, 56485) was carried out following the manufacturer's instructions in the pLenti6.3/TO/V5-DEST destination vector (ThermoFisher Scientific, V53306). The 4 plasmids, pENTR2b-Lifeact-mNeonGreen, pENTR2b-Lifeact-mScarlet-I, pLenti6.3/TO/V5-DEST-Lifeact-mNeonGreen, and pLenti6.3/TO/V5-DEST-Lifeact-mScarlet-I, were verified using sequencing. Generated plasmids are being deposited on Addgene (https://www.addgene.org/Guillaume_Jacquemet/).

HEK293FT cells were used to produce lentiviral particles. Cells were transfected with a third-generation lentiviral packaging system: the envelope plasmid pMD2.G (a gift from Didier Trono, Addgene #12259), packaging plasmids pMDLg/pRRE (a gift from Didier Trono, Addgene #12251) (Dull et al, 1998), pRSV-Rev (a gift from Didier Trono, Addgene #12253, (Dull et al, 1998)), and one of the lifeact plasmids described above. The plasmids were co-transfected into HEK293FT cells at a 1:22:15:50 ratio, respectively, using 50 μL of Lipofectamine 3000 (ThermoFisher Scientific, 100022052) in 8 mL of OptiMEM (Gibco, 31985070). Twenty hours post-transfection, the media was replaced with the standard growth medium. Viral particles were collected after 36 h, filtered through 0.45 μm filters, and stored as 1 mL aliquots at −80 °C.

PDAC cells were transduced with the lentiviral supernatant, mixed at a 1:1 ratio with fresh culture medium, and supplemented with 1% Lipofectamine 3000. After 48 h of incubation, transduced cells were selected by adding 2 μg/mL blasticidin (Gibco, A11139-03) to the culture medium. Selection was maintained for two weeks until non-transduced cells were eliminated. The surviving, stably transduced PDAC cells were then sorted using fluorescence-activated cell sorting (FACS) on a BD FACSAria II cell sorter to isolate populations expressing lifeact-mNeonGreen or lifeact-mScarlet-I.

## Isolation of neutrophils and PBMC from human blood samples

A total of 30 mL of human blood was collected in heparin-treated tubes from different anonymous healthy donors for each experiment, with ethical permission from VARHA (ETMK Dnro: 43/1801/2015). Neutrophils and PBMC cells were isolated from whole blood using density gradient centrifugation techniques, with neutrophils isolated using Percoll gradients (Sigma-Aldrich, p1644) and PBMC using Ficoll gradients (Kirveskari et al, 2000). Neutrophils were concentrated beneath the Percoll layer, along with red blood cells. An osmotic shock procedure was performed to remove red blood cells by adding 1 mL of 0.2% NaCl to the cell pellet, vortexing for 18 s, and then adding 1 mL of 1.8% NaCl, followed by pelleting the cells. This process was repeated three times. PBMCs were concentrated at the interface between the Ficoll layer and the platelet-rich plasma. This was followed by careful pipetting of the PBMC layer and differential centrifugation to minimize platelet contamination. Both neutrophil and PBMC cell

fractions were washed with PBS and kept on ice for 3 h before being used in perfusion experiments.

## Western blot

Cells were lysed using a lysis buffer containing 50 mM Tris-HCl pH 7.1, 150 mM NaCl, 2 mM CaCl$_2$, 1 mM EDTA, 2 mM MgCl$_2$, 0.5% Triton X-100, supplemented with Complete protease inhibitors mix (05056489001; Roche), PhosStop phosphatase inhibitors mix (04906837001; Roche), and GENIUS Nuclease (sc-391121; Santa Cruz Biotechnology) in recommended concentrations. The lysates were incubated on ice for 20 min, after which 2% SDS in 50 mM Tris, pH 7.1, was added to the final concentration of 1%. The lysates were heated for 5 min at 95 °C and cleared by centrifugation for 10 min at 13,000×g. The concentration of the protein extracts was quantified using the DC protein assay (Reagent A, #5000113; Reagent B, #5000114; Reagent S, #5000115; BioRad). Finally, samples were heated in the reducing sample buffer, resolved by SDS-PAGE, and transferred onto a low fluorescence PVDF membrane (IPFL00005; Millipore). The membrane was blocked for 30 min at room temperature using AdvanBlock-Fluor blocking solution (R-03729-E10; Advansta). The membranes were incubated overnight in the blocking solution with the indicated primary antibodies at 1:1000 dilution and for 1 h at RT with the respective AzureSpectra fluorescent secondary antibodies at 1:3000 dilution (AC2166; Azure Biosystems). The blots were visualized using the Sapphire Biomolecular Imager (Azure Biosystems).

## Quantitative PCR

RNA extracts were collected using the NucleoSpin RNA Plus kit (740984.250; Macherey-Nagel), and the concentration was measured using a NanoDrop Lite Spectrophotometer (ThermoFisher Scientific). Subsequently, cDNA synthesis was performed using the High-Capacity cDNA Reverse Transcription Kit (4368814; Applied Biosystems). Real-time quantitative PCR (RT-qPCR) was performed on QuantStudio 3 (A28567; ThermoFisher Scientific) using pre-designed single-tube TaqMan gene expression assays (pan CD44: Hs01075864_m1 and GAPDH: Hs02786624_g1). Each sample was analyzed in triplicate, with gene expression levels normalized to GAPDH expression. Data analysis was performed using QuantStudio Design & Analysis Software version 1.5.2.

## Immunofluorescence stainings

PDAC cells were labeled with CellTrace (ThermoFisher Scientific, C34570) at a 1:1000 dilution in cell culture medium for 30 min before trypsinization and perfusion. Samples were fixed using 4% paraformaldehyde (ThermoFisher Scientific, 28908) in PBS by perfusion through microfluidic channels or incubating at 37 °C for 10 min. Primary antibodies (diluted 1:200) were incubated for 1 h at room temperature, followed by incubation with species-appropriate secondary antibodies (1:400 dilution) for 45 min. Alexa-647 phalloidin (1:800) and DAPI (1:3000 from a 5 mg/ml stock) were added alongside the secondary antibodies. Between each step, samples were washed with PBS, and upon completion of the staining protocol, microfluidic channels were stored in PBS at 4 °C. ProLong Glass media (Invitrogen, P36984) was used for coverslip mounting.

Structured illumination microscopy (SIM) samples were prepared in glass-bottom ibidi microfluidic channels (ibidi, 80177). Cells were stained using the aforementioned immunofluorescence protocol and then mounted in Vectashield (Vector Laboratories, H-1000-10) for subsequent imaging.

To stain cells in suspension, coverslips were pre-coated with 10 µg/ml Poly-D-Lysine at 37 °C for 1 h. Cells were pipetted onto the coverslips for 10 min before fixation, allowing low binding without spreading.

Hyaluronic acid (HA) was detected using an HA-binding protein linked to biotin (HABP-biotin) at a 1:20 dilution from the stock solution (50 µg/ml in 1% BSA-PBS). HABP-biotin was produced in-house, as previously described (Arasu et al, 2017). Incubation was performed for 4 h at room temperature. Detection was achieved using Alexa 594-conjugated streptavidin (Invitrogen, S11227) at a 1:500 dilution for 2 h at room temperature.

## Light microscopy setup

The spinning-disk confocal microscope used was a Marianas spinning-disk imaging system with a Yokogawa CSU-W1 scanning unit on an inverted Zeiss Axio Observer Z1 microscope controlled by SlideBook 6 (Intelligent Imaging Innovations, Inc.). Images were acquired using an Orca Flash 4 sCMOS camera (chip size 2048 × 2048; Hamamatsu Photonics). The objectives used were a ×63 (NA 1.4 oil immersion, Zeiss PLN Apo), a ×40 (NA 1.1 Water, Zeiss LD C-Apochromat), and a ×20 (NA 0.8 Air, Zeiss Plan-Apochromat).

The structured illumination microscope used was DeltaVision OMX v4 (GE Healthcare Life Sciences) fitted with a ×60 Plan-Apochromat objective lens, 1.42 NA (immersion oil RI of 1.516), used in SIM illumination mode (five phases × three rotations). Emitted light was collected on a front-illuminated pco.edge sCMOS (pixel size 6.5 mm, readout speed 95 MHz; PCO AG) controlled by SoftWorx.

The confocal microscope used was a laser scanning confocal microscope, LSM880 (Zeiss), equipped with an Airyscan detector (Carl Zeiss) and a ×20 Plan-Apochromat objective, NA 0.8. The microscope was controlled using Zen Black (2.3), and the Airyscan was used in standard super-resolution mode.

The widefield microscope used was a Nikon Eclipse Ti2-E fitted with a 10x Nikon CFI Plan-Fluor objective lens, 0.3 NA, and a ×20 Nikon CFI S Plan Fluor ELWD, NA 0.45. The microscope was controlled using NIS-Elements AR 5.11.01 64-bit software.

## Microfluidic setup

Microfluidic channels used in this study were either 1-channel µ-slides LUER 0.4 (ibidi, 80176) or 6-channel µ-slides LUER 0.4 (ibidi, 80606). For high-resolution imaging, glass-bottom channels of the same type were employed (ibidi, 80177). Before cell seeding, channels were coated with 10 µg/ml fibronectin (Sigma-Aldrich, 341631) for 1 h at 37 °C. Endothelial cells were seeded at a density of 800,000 cells/ml and cultured with media changes twice daily for 3–4 days. Flow priming, when required, was performed overnight. A peristaltic pump (Masterflex® Ismatec® 78018-24) was used to maintain flow for all perfusion experiments. The flow system, including tubing assembly, was custom-built in-house using components from Ibidi and IDEX: ISMATEC-070535-04i-ND

SC0052T, IBIDI-10840, IBIDI-10842, IBIDI-10829, IBIDI-10802, and IBIDI-10827. Detailed assembly instructions are described in Osmani et al, MMB (Osmani et al, 2021).

## Live microfluidic experiments

The microfluidic pump, channel slides, and tubing setup were installed inside the heated chamber of a Nikon Eclipse Ti2-E Widefield microscope, equipped with either a ×10 Nikon CFI Plan-Fluor objective lens (NA 0.3) or a ×20 Nikon CFI Plan Apo Lambda objective lens (NA 0.75). Before perfusion, warm culture medium was used to prime the tubing before connecting it to the channels. Imaging regions were carefully selected and positioned 500 μm from the channel's top border and at least 1.5 cm from the perfusion entry well to minimize disturbances and maintain consistent flow conditions across experiments.

During the setup process, a cell suspension (300,000 cells/ml) of PDAC or immune cells was prepared in endothelial cell media. A total volume of 5 ml was used for the 8-min perfusion. The tubing was washed or replaced between experiments before setting up the following perfusion. Brightfield images were acquired with a 5-ms exposure time and a frame rate of 25 frames per second. Continuous imaging was performed over four perfusion speeds (400, 200, 100, and 400 μm/s), with 2 min of acquisition per speed, resulting in ~12,000 frames per experiment (8 min total). The raw movies were then segmented into four standardized extracts, each consisting of 2250 frames, for analysis.

To assess the perfusion speed within the channels, PDAC cells were perfused and tracked at various heights and positions. Channels were coated with 1% BSA for 1 h at 37 °C before the perfusion of cancer cells (500,000 cells/ml in PBS) at a speed of 300 μm/s as previously described (Fazeli et al, 2020).

For experiments involving IL-1β treatment, cells were either incubated in culture media with 10 ng/ml IL-1β for 2 h or with 5 ng/ml IL-1β for 16 h before perfusion, as indicated. For blocking antibody treatments, the HERMES-1 anti-CD44 antibody (DSHB, AB_528148) was used at a concentration of 5 μg/mL (for PDAC cells) or 1.5 μg/mL (for endothelial cells). Cells were incubated for 10 min with the antibody and washed before perfusion. Hyaluronic acid digestion treatments were performed using hyaluronidase (HYAL, Sigma-Aldrich, H3506) in culture media for 20 min. In total, 120 U of hyaluronidase was applied to PDAC cells, while 40 U was used for the HUVEC monolayer.

## Distance analyses

The distance between a fibronectin patch and the nearest endothelial cell junctions and the distance between a fibronectin patch and the nearest nucleus were measured from images of endothelial monolayers stained for fibronectin (FN), PECAM-1, and DAPI. 3D stack images were acquired using a spinning disk confocal microscope using a ×40 (NA 1.1 Water, Zeiss LD C-Apochromat) objective and a Z-step size of 0.8 μm. Fibronectin patches were extracted from maximum intensity projections using the "Default" thresholding method and the "Analyze Particles" function in Fiji (Schindelin et al, 2012). Pixels with an intensity below 50 were removed using the "Remove Outliers" function, and objects smaller than 10 pixels were filtered out. Endothelial cell junctions were segmented using a custom-trained Cellpose model based on the Cyto2 model (Stringer et al, 2021) via the Cellpose GUI

(Pachitariu and Stringer, 2022). The model was trained on 14 images using default training parameters. For predictions, the disk size was set to 300 px, the flow threshold to 0.4, and the probability threshold to 0. After segmentation, the junctions were manually curated and post-processed by dilating them by 3 pixels, followed by skeletonization to create one-pixel-thick junction masks and to remove segmentation errors. The nuclei were segmented using a pre-trained StarDist versatile nuclei model (Schmidt et al, 2018). The distances between the fibronectin patches and the nearest junctions (Edge-to-Edge) and the distances between the fibronectin patches and the nearest nuclei were measured using the DiAna plugin in Fiji (Gilles et al, 2017).

The distance between fibronectin or thrombospondin patches and the nearest attached PDAC cell was measured using images of endothelial monolayers previously perfused with Lifeact-mScarlet-I-positive PDAC cells at 400 μm/s. The monolayers were then fixed and stained for fibronectin or thrombospondin, PECAM-1, and DAPI. 3D stack images were acquired using a spinning disk confocal microscope with a 40x (NA 1.1 Water, Zeiss LD C-Apochromat) objective and a Z-step size of 0.8 μm. Maximal intensity projections of the fibronectin or thrombospondin images were used to segment ECM patches using the "Default" thresholding method in Fiji. Pixels with an intensity below 50 were removed using the "Remove Outliers" function, and objects smaller than 10 pixels were filtered out. PDAC cells were segmented using the Triangle thresholding method, and segmented masks were converted into label images. Segmentation masks were manually curated to ensure accurate measurement of distances. The distance between the edges of fibronectin or thrombospondin patches and segmented cancer cells (Edge-to-Edge) was measured using the DiAna plugin in Fiji (Gilles et al, 2017). fibronectin and thrombospondin images were shuffled and randomly paired with cancer cell images from the same group before performing a similar distance analysis to generate randomized controls for distance measurement.

## Deep-learning segmentation models

All segmentation models were validated on test datasets (Laine et al, 2021), summarized on the GitHub page accompanying this paper (https://github.com/CellMigrationLab/PDAC_DL/), and archived on Zenodo together with the training dataset used (https://zenodo.org/communities/pdac_dl). To segment cancer, immune cells, and endothelial nuclei from live brightfield images and fixed fluorescent images, we trained multiple StarDist 2D models using the ZeroCostDL4Mic and DL4MicEverywhere platforms (von Chamier et al, 2021; Hidalgo-Cenalmor et al, 2024; Schmidt et al, 2018). Images used for training were manually annotated in Fiji (Schindelin et al, 2012), and labeled images were exported using the BIOP LaRoMe plugin (https://github.com/BIOP/ijp-LaRoMe). All models were trained using the mean absolute error (mae) loss function with an initial learning rate of 0.0003.

The StarDist model, used to detect cancer cells perfused in BSA-coated channels, was previously described (Fazeli et al, 2020). The training dataset was composed of 57 paired images. The model was trained for 200 epochs using a patch size of 1024 ×1024. The generated StarDist model demonstrated high performance, yielding an average F1-score of 0.933 on the test dataset.

A training dataset of 66 images was generated to train the StarDist model used to detect cancer cells labeled with CellTrace from fixed images (Follain et al, 2024a). The model was trained for

200 epochs using a patch size of 1024 × 1024 and a batch size of 2. The generated StarDist model demonstrated high performance, yielding an average F1-score of 0.877 on the test dataset.

To train the StarDist model, capable of segmenting cancer cells from endothelial cells (20x) (Follain et al, 2024b), we manually annotated 20 paired images. We computationally augmented this dataset to 160 paired images using the Augmentor ZeroCostDL4-Mic notebook (Bloice et al, 2019; von Chamier et al, 2021). The model was trained for 400 epochs using a patch size of 992 × 992 and a batch size of 2. This approach produced a model demonstrating an average F1-score of 0.921 on our test dataset.

To train the StarDist model, capable of segmenting cancer cells from endothelial cells (10x) (Follain et al, 2024h), we manually annotated 77 paired images. We computationally augmented this dataset by 8 during training. The model was trained for 500 epochs using a patch size of 992 ×992 and a batch size of 2. This approach produced a model demonstrating an average F1-score of 0.968 on our test dataset.

We manually annotated 36 paired images to train the StarDist model, which is capable of detecting neutrophils on endothelial cells (Follain et al, 2024c). The model was trained for 400 epochs using a patch size of 992 × 992 and a batch size of 2. This approach produced a model demonstrating an average F1-score of 0.969 on our test dataset.

We manually annotated 27 paired images to develop the StarDist model, which is capable of detecting PBMCs on endothelial cells (Follain et al, 2024d). The model was trained for 400 epochs using a patch size of 992 × 992 and a batch size of 2. This approach produced a model demonstrating an average F1-score of 0.941 on our test dataset.

To train the StarDist model capable of segmenting endothelial nuclei while ignoring cancer cells (Follain et al, 2024f), 17 images were first segmented using the StarDist Versatile nuclei model in Fiji, and the nuclei of cancer cells were manually removed. We then computationally augmented this dataset to 68 paired images using the Augmentor ZeroCostDL4Mic notebook (Bloice et al, 2019; von Chamier et al, 2021), employing rotation and flipping. The model was trained for 200 epochs using a patch size of 1024 × 1024 and a batch size of 2. This approach produced a model demonstrating an average F1-score of 0.976 on our test dataset.

To train the StarDist model capable of detecting individual PDAC cells from the Lifeact fluorescence channel (Follain et al, 2024k), we first manually annotated 10 paired images and then computationally augmented this dataset to 40 images using the Augmentor ZeroCostDL4Mic notebook (Bloice et al, 2019; von Chamier et al, 2021). The model was trained for 400 epochs using a patch size of 1024 × 1024 and a batch size of 2. This approach produced a model demonstrating an average F1-score of 0.967 on our test dataset.

To train the StarDist model capable of detecting MIA PaCa-2 cells from the CD44 fluorescence channel (Follain et al, 2024k), we manually annotated 8 paired images and computationally augmented this dataset to 40 images using the Augmentor Zero-CostDL4Mic notebook (Bloice et al, 2019; von Chamier et al, 2021). The model was trained for 400 epochs using a patch size of 896 × 896 and a batch size of 2. This approach produced a model demonstrating an average F1-score of 0.950 on our test dataset.

To train the StarDist model capable of detecting MIA PaCa-2 cell nuclei from images containing nuclei of both MiaPaca cells and HUVECs (Follain et al, 2024l), we manually annotated 48 paired images. The model was trained for 200 epochs using a patch size of 896 × 896 and a batch size of 2. This approach produced a model demonstrating an average F1-score of 0.793 on our test dataset.

## Artificial labeling

Artificial labeling of brightfield images was achieved by training pix2pix models using the ZeroCostDL4Mic pix2pix notebook (von Chamier et al, 2021; Isola et al, 2017). The artificial labeling models used are summarized on the GitHub page accompanying this paper (https://github.com/CellMigrationLab/PDAC_DL/) and archived on Zenodo (https://zenodo.org/communities/pdac_dl). Pix2pix models were trained using an initial learning rate of 0.0002.

To generate fake nuclei and PECAM-1 staining from brightfield images, we first created a training dataset using fixed microfluidic channels containing endothelial and PDAC cells, as well as endothelial and immune cells. Samples were stained with PECAM-1 and DAPI, and images were acquired using a Nikon Eclipse Ti2-E microscope with a ×20 objective.

To train the pix2pix model capable of generating fake nuclear staining from brightfield images, we used a training dataset comprising 258 paired images of circulating cancer cells (Follain et al, 2024g) and 226 paired images of circulating immune cells (Follain et al, 2024g). Both training datasets were computationally augmented using the ZeroCostDL4Mic Augmentor notebook (Bloice et al, 2019; von Chamier et al, 2021). Both models were trained for 400 epochs using a patch size of 512 × 512, a batch size of 1, and a vanilla GAN loss function. The best models were selected based on quality metric scores and a visual inspection of the generated images compared to the ground truth. These approaches produced models with an average SSIM score of 0.755 and an LPIPS score of 0.120 for the circulating cancer cell images, and an average SSIM score of 0.756 and an LPIPS score of 0.130 for the circulating immune cell images (on the test dataset).

To train the pix2pix model capable of generating fake PECAM-1 staining (Follain et al, 2024e) from brightfield images, we used a training dataset of 484 paired images. We computationally augmented this dataset to 2904 paired images using the Augmentor ZeroCostDL4Mic notebook (Bloice et al, 2019; von Chamier et al, 2021). The pix2pix model was trained using 245 epochs, a patch size of 512 × 512, a batch size of 1, and a vanilla GAN loss function. The best model was chosen based on quality metric scores and a visual inspection of the generated images compared to the ground truths. This approach produced a model demonstrating an average SSIM score of 0.273 and an LPIPS score of 0.360 on our test dataset.

## Segmentation of endothelial cell nuclei and cell–cell junctions from brightfield video

Segmenting endothelial cell nuclei and cell–cell junctions from live brightfield imaging involved a multi-step process. Initially, the brightness and contrast of each frame were normalized using adaptive histogram equalization. Each frame was then assessed for sharpness using Sobel and LoG operators. Based on sharpness, the top 25% of frames underwent artificial labeling with pix2pix models to predict DAPI and PECAM-1 staining. Sum projections of all the predicted frames were created to generate consensus DAPI and PECAM-1 images. Endothelial nuclei were segmented using a StarDist model from the consensus DAPI image. Endothelial cells were segmented from the merged DAPI and PECAM-1 consensus images using Cellpose (Cyto2

model, (Stringer et al, 2021)). The endothelial cell–cell junctions were determined using cell boundary delineation with pyclesperanto_prototype's detectLabelEdges function (Haase et al, 2023).

## Cell tracking and quantitative analysis of tracking data

Brightfield movies were first segmented frame by frame using the appropriate StarDist model, depending on the cells being perfused, with a custom-made notebook based on ZeroCostDL4Mic and DL4MicEverywhere (von Chamier et al, 2021; Hidalgo-Cenalmor et al, 2024). Generated labeled images were then tracked using TrackMate, the TrackMate label detector, and the Simple LAP tracker (Ershov et al, 2022). The following settings were used: Linking max distance: 15.0 μm, gap-closing max distance: 15.0 μm, gap-closing max frame gap: 4, number of spots in track: 11. Upon completion of cell tracking, the datasets were compiled and analyzed using a customized version of CellTracksColab (Gómez-de-Mariscal et al, 2024b). The adaptations to the original notebooks are publicly accessible via the GitHub repository associated with this publication. Furthermore, the tracking datasets have been deposited on Zenodo (Follain et al, 2024i, 2024j, 2024q, 2024o, 2024p).

In CellTracksColab, a preliminary filtration step was performed to retain only tracks containing a minimum of 50 spots. The Forward Migration Index (FMI) in the direction of flow was computed to assess directional persistence quantitatively. To evaluate the temporal dynamics of cell arrest within the endothelial cell monolayer, we enumerated tracks exhibiting an instantaneous speed of 5 μm/s or lower at sequential time points, facilitating an investigation into the temporal patterns of cell adhesion under flow. The cell adhesion rate was quantitatively assessed using a linear fit to these temporal data.

To define co-arrest and determine whether circulating cells arrest as single cells or within clusters, we analyzed "slowdown" events detected along each track (instantaneous speed ≤5 μm·s$^{-1}$; see above). We classified arrests as clusters when two or more cells stopped at the same time and within one cell diameter of each other in the same field of view. The spatial threshold was based on the mean diameter of the corresponding cell type, measured directly from the microscopy videos (AsPc-1: 17 μm; MiaPaCa-2: 20 μm; Panc10: 20 μm; PBMC: 15 μm; Neutrophil: 12 μm). Events that did not meet these criteria were considered single-cell arrests.

To evaluate the dynamic arrest patterns of circulating cells within endothelial monolayers, we first identified events where cell velocity dropped below 5 μm/s, marking the initial arrest point within a cell trajectory. Ripley's K and L functions were employed to analyze these spatial point patterns. Ripley's K function was used to quantify the spatial density of arrested cell events within a given radius r, facilitating the assessment of clustering or dispersion by comparing the observed density to that expected under a random distribution. The L function, derived from the K function, offers a more intuitive interpretation of spatial patterns; values near zero indicate randomness, positive values suggest clustering, and negative values denote dispersion. Additionally, Monte Carlo simulations were conducted to establish a baseline for spatial distributions under various conditions. For comparative analysis across different conditions and cell types, we utilized a clustering radius of 50 μm, leveraging Ripley's L function to discern the underlying spatial organization of cell arrest within the endothelial monolayer.

To delineate cell trajectories indicative of a definitive arrest pattern, we refined our analysis to include only those tracks where cells demonstrated an initial velocity exceeding 20 μm/s, decelerating to a minimum speed below 1 μm/s, and remaining at a speed below 5 μm/s at the track's conclusion. The initial point of landing was identified as the first coordinate at which the velocity of these designated landing tracks decreased to below 10 μm/s. Furthermore, the earliest point of arrest within these trajectories was identified as the coordinate at which cells first reached their minimum speed. We utilized the find_peaks function from the SciPy library to identify peaks in cell velocity profiles. Peaks were identified in the cell track segments following the first arrest event. The criteria for peak detection included parameters for height (10), threshold (0.1), and distance (5) between peaks. The proximity between tracked cells and endothelial structures, specifically junctions and nuclei, was calculated employing the Euclidean distance transform algorithm from the SciPy library, specifically distance_transform_edt (Virtanen et al, 2020).

Dataset comparing PDAC cells and immune cells arrest on endothelial monolayers in the presence and absence of IL-1β stimulation (https://zenodo.org/records/17160727).

| Cells perfused | Treatment (endothelial cells) | Number of biological repeats | Total number of tracked objects | Number of tracks |
|---|---|---|---|---|
| AsPC-1 | CTRL | 7 | 3,881,052 | 22,747 |
| AsPC-1 | IL-1β 2 h | 3 | 1,244,877 | 7473 |
| MIA PaCa-2 | CTRL | 6 | 7,064,923 | 13,223 |
| MIA PaCa-2 | IL-1β 2 h | 2 | 3,155,777 | 7045 |
| PBMC | CTRL | 6 | 4,576,576 | 35,147 |
| PBMC | IL-1β 2 h | 4 | 3,929,661 | 22,886 |
| Neutrophil | CTRL | 8 | 12,804,225 | 33,084 |
| Neutrophil | IL-1β 2 h | 4 | 7,225,020 | 13,610 |
| Panc 10.05 | CTRL | 4 | 2,567,833 | 21,726 |
| Panc 10.05 | IL-1β 2 h | 4 | 2,611,582 | 16,111 |

Dataset investigating the influence of IL-1β stimulation duration on the arrest of immune cells on endothelial monolayers (https://zenodo.org/records/17159634).

| Cells perfused | Treatment (endothelial cells) | Number of biological repeats | Total number of tracked objects | Number of tracks |
|---|---|---|---|---|
| PBMC | CTRL | 5 | 1,002,937 | 8728 |
| PBMC | IL-1β 2 h | 4 | 1,350,171 | 7219 |
| PBMC | IL-1β 16 h | 7 | 1,681,719 | 7143 |
| Neutrophil | CTRL | 4 | 3,492,863 | 19,890 |
| Neutrophil | IL-1β 2 h | 5 | 8,495,407 | 23,607 |
| Neutrophil | IL-1β 16 h | 7 | 15,547,177 | 55,872 |

Dataset investigating the influence of CD44 silencing in endothelial cells on PDAC cell arrest on endothelial monolayers (https://zenodo.org/records/13377961).

| Cells perfused | Treatment (endothelial cells) | Number of biological repeats | Total number of tracked objects | Number of tracks |
|---|---|---|---|---|
| AsPC-1 | siCD44 #1 | 2 | 3,858,650 | 12,136 |
| AsPC-1 | siCD44 #2 | 2 | 3,930,915 | 10,295 |
| AsPC-1 | siCD44 #3 | 2 | 3,054,893 | 10,213 |
| AsPC-1 | siCTRL | 2 | 2,875,073 | 7204 |
| MIA PaCa-2 | siCD44 #1 | 4 | 6,977,672 | 24,809 |
| MIA PaCa-2 | siCD44 #2 | 4 | 5,856,882 | 20,557 |
| MIA PaCa-2 | siCD44 #3 | 4 | 8,102,836 | 28,842 |
| MIA PaCa-2 | siCTRL | 4 | 9,669,657 | 28,139 |

Dataset investigating the influence of CD44 silencing in PDAC cells on PDAC cell arrest on endothelial monolayers (https://zenodo.org/records/13379627).

| Cells perfused | Treatment (PDAC cells) | Number of biological repeats | Total number of tracked objects | Number of tracks |
|---|---|---|---|---|
| AsPC-1 | siCD44 #1 | 3 | 4,566,249 | 30,505 |
| AsPC-1 | siCD44 #3 | 3 | 3,082,446 | 18,466 |
| AsPC-1 | siCTRL | 3 | 6,786,224 | 17,684 |
| MIA PaCa-2 | siCD44 #1 | 3 | 4,951,490 | 33,334 |
| MIA PaCa-2 | siCD44 #3 | 3 | 3,980,478 | 28,066 |
| MIA PaCa-2 | siCTRL | 3 | 16,741,485 | 34,104 |

Dataset investigating the influence of pre-treating cells (PDAC cells and/or endothelial cells) with an anti-CD44 blocking antibody on PDAC cell arrest on endothelial monolayers (https://zenodo.org/records/13584215).

| Cells perfused | Treatment (cell treated) | Number of biological repeats | Total number of tracked objects | Number of tracks |
|---|---|---|---|---|
| AsPC-1 | Endothelial cells | 3 | 8,432,292 | 17,772 |
| AsPC-1 | PDAC cells | 3 | 4,588,334 | 30,850 |
| AsPC-1 | All cells | 2 | 2,325,186 | 14,754 |
| AsPC-1 | Untreated | 2 | 5,721,330 | 17,095 |
| MIA PaCa-2 | Endothelial cells | 4 | 15,175,979 | 21,825 |
| MIA PaCa-2 | PDAC cells | 4 | 6,716,590 | 40,724 |
| MIA PaCa-2 | All cells | 4 | 6,463,932 | 42,144 |
| MIA PaCa-2 | Untreated | 5 | 15,002,179 | 28,151 |

Dataset investigating the influence of Hyaluronic acid digestion treatments on PDAC cell arrest on endothelial monolayers (https://zenodo.org/records/13627037).

| Cells perfused | Treatment (cell treated) | Number of biological repeats | Total number of tracked objects | Number of tracks |
|---|---|---|---|---|
| AsPC-1 | Endothelial cells | 3 | 2,655,144 | 19,128 |
| AsPC-1 | PDAC cells | 2 | 2,091,538 | 13,118 |
| AsPC-1 | Untreated | 3 | 6,862,132 | 19,066 |
| MIA PaCa-2 | Endothelial cells | 3 | 1,817,623 | 12,366 |
| MIA PaCa-2 | PDAC cells | 3 | 3,691,910 | 22,791 |
| MIA PaCa-2 | Untreated | 3 | 8,807,315 | 14,941 |

## Analysis of cell attachment patterns to CD44. VCAM-1, E-selectin, ICAM-1, ICAM-2, and fibronectin

Using the microfluidic setup detailed above, coupled with a single channel and "Y" tubing, lifeact-mScarlet-I MIA PaCa-2 and AsPC-1 cells were perfused at a concentration of 500,000 cells/ml over an endothelial cell monolayer at a flow speed of 200 μm/s for 10 min. The channels were then fixed by perfusing warm 4% paraformaldehyde (PFA, ThermoFisher Scientific, 28908) in PBS for 10 min at the same flow rate. Following fixation, the microfluidic channels were stained without permeabilization according to a previously described immunofluorescence staining protocol. Primary antibodies used included CD44 (1:200, Abcam ab254530), ICAM-1 (1:200, Invitrogen MA5-11569), ICAM-2 (1:200, Invitrogen 14-1029-82), VCAM-1 (1:200, Abcam ab316314), E-selectin (1:200, Abcam ab300557) and fibronectin (FN) (1:200, Invitrogen PA5-98811). Cell nuclei were also stained with DAPI (ThermoFisher Scientific, D1306). Images were acquired using a spinning disk confocal microscope (3i CSU-W1) equipped with a 20x Zeiss Plan-Apochromat objective (NA: 0.8). Z-stacks were acquired with a 20 μm depth, a 1 μm step size, and 2 × 2 binning.

From the acquired images, PDAC cells were segmented using custom StarDist models based on either the lifeact signal, the DAPI signal, or the CD44 signal (Follain et al, 2024m, 2024n, 2024l). The segmented cell masks were manually validated for all images to ensure accuracy and consistency. Segmented cell masks were

overlaid onto summed projections of the respective marker channels to correlate attached cells with specific markers. Cell positivity for each marker was manually scored by visual inspection, and the observed percentage of marker-positive cells was computed. To calculate the expected number of cancer cells attaching to specific markers, marker-specific masks were generated for each field of view using intensity-based thresholding in Fiji. Attachment simulations were performed by randomly placing the same number of cancer cells observed in each field of view onto the generated marker masks. Each simulated cell was checked for overlap with the marker mask, and the number of overlaps was recorded. The number of simulations per mask was set to 1,000 for statistical robustness. Simulated overlap counts were compared to the observed overlaps to determine whether cell attachment occurred randomly or in a marker-specific manner. The Jupyter Notebook used to perform the attachment simulation is available on the GitHub repository associated with this manuscript.

## Analysis of CD44 and HA levels in cells in suspension

MIA PaCa-2 and AsPC-1 cells were treated with either control siRNA, siRNA targeting CD44, or hyaluronidase as indicated. Cells were plated on poly-L-lysine-coated coverslips for 10 min to ensure minimal adherence without spreading. Following this, cells were fixed with 4% paraformaldehyde and stained for DAPI, CD44, and HA, as previously described. The samples were then imaged using a spinning disk confocal microscope. SUM projections of the confocal stacks were performed using the microscope acquisition software. Images were then analyzed using a custom Python Jupyter notebook, publicly available in the GitHub repository linked to this manuscript. The Cellpose algorithm was employed for cell segmentation using the Cyto3 model, using the CD44 and DAPI images. Objects smaller than 3000 pixels square were filtered out to exclude non-cellular debris. The integrated density of the CD44 and HA signal was measured for each segmented cell to assess the levels of these proteins following the treatments.

## Zebrafish models and injection with cancer cells

Adult zebrafish were maintained in a 12-h light–dark cycle in stand-alone housing racks (Aqua-Schwartz, Göttingen, Germany). Embryos were obtained through natural spawning in specialized 1.7 L sloped-bottom mating tanks (Tecniplast, Buguggiate, Italy). The *Tg(5xUAS:cdh5-EGFP)* transgenic zebrafish line (Lenard et al, 2013; Herwig et al, 2011; ZFIN identifier: ubs12Tg), kindly provided by Dr. Markus Affolter and Dr. Heinz-Georg Belting (University of Basel, Switzerland), was used in these experiments. All zebrafish housing and experimental procedures were conducted under licenses MMM/465/712–93 (Finnish Ministry of Agriculture and Forestry) and GTLK/004/E/2016 (Finnish Ministry of Social Affairs and Health) and carried out in the Zebrafish Core Facilities at Turku Bioscience Centre.

Embryos were maintained in E3 medium (17.4 mM NaCl, 0.2 mM KCl, 0.1 mM $MgSO_4$, 0.2 mM $Ca(NO_3)_2$), buffered with 0.15 mM HEPES (pH 7.6), and supplemented with 200 mM 1-phenyl-2-thiourea (Sigma-Aldrich, P7629) to inhibit melanogenesis, as previously described (Goetz et al, 2014). At 48 h post-fertilization (hpf), embryos were immobilized in 0.8% low-melting-point agarose and chemically anesthetized using E3 medium

supplemented with 650 mM tricaine (ethyl-3-aminobenzoate-methanesulfonate). PDAC cells were injected into the duct of Cuvier using a Nanoject microinjector 2 (Drummond) with micro forged glass capillaries (25 to 30 μm inner diameter) filled with mineral oil (Sigma-Aldrich). Injections consisted of 13 nL of a cell suspension at a concentration of $100 \times 10^6$ cells/ml. The embryos were positioned under an AxioZoom microscope (Zeiss) for injections. A more detailed version of the protocol is available (Follain et al, 2018b). Following injection, embryos were maintained on an agarose pad for 3 h before imaging the caudal plexus region. Imaging was performed using an Airyscan microscope (Zeiss) equipped with a 40x objective. 3D rendering of the acquired images was performed using arivis Vision4D 3.5.0.

## Atomic force microscopy

AFM experiments were performed using a JPK NanoWizard II system (Bruker Nano GmbH), which includes a CellHesion@ module, mounted on a Zeiss LSM510 confocal microscope (Carl Zeiss NTS Ltd.). Endothelial cells were grown until confluent on fibronectin-coated coverslips. These coverslips were mounted on a heated holder (JPK BioCell for AFM, Bruker Nano GmbH), and the media was supplemented with 20 mM HEPES (Sigma, H0887), allowing live imaging for several hours. For calibration and live recording of the experiments, we used a 10x and a 20x objective connected to a webcam camera attached to the LSM510 Zeiss microscope. Both experiments utilized 0.01 N/m force constant AFM cantilevers (Nanosensors, qp-SCONT) with tips featuring a resonance frequency range of 8–13 kHz, a length of 120–130 μm, a mean width of 32–36 μm, and a thickness of 320–380 nm. The cantilever spring constant and deflection sensitivity were calibrated in fluid via the thermal noise method (Hutter and Bechhoefer, 1993). In the case of contact mode AFM, 10 fields of view, ranging from $100 \times 100$ to $150 \times 150$ μm, were acquired and rendered in the AFM software (JPK DP version 4.2). Quantification was performed based on line profiles drawn from the nucleus to the nucleus of neighboring cells using the same software ($n = 124$).

In the case of tapping mode AFM, five fields of view, ranging from $100 \times 150$ to $100 \times 150$ μm, were acquired with three replicate force curve measurements taken every 3–5 μm. The elastic modulus for each force curve was calculated using JPK data processing software (JPK DP version 4.2), assuming a Hertz impact model. Each measurement position result was carefully curated in the software, resulting in a mean value per position. These are reported using a color-coded square on the corresponding brightfield image. In parallel, cell–cell junctions and nuclei were manually drawn based on the same brightfield image and used as an overlay for the final image. Only one example image is shown in the figure. Graphs depict data from all fields of view. Classification of the values to "junctions" or "nuclei" was done manually using the overlay.

## Simulation of adhesion under flow

Simulations were performed to model the attachment of PDAC cells under various flow conditions. The simulations were conducted within a $512 \times 512$ pixel grid, where each pixel was assigned a flow vector aligned along the $y$ axis to simulate unidirectional flow. Each simulation ran for 500 frames (iterations), during which cells, modeled as frictionless spheres, were introduced

into the field from the left at random x-axis positions. To prevent overlapping, a constraint ensured that no two cells occupied the same space. Once introduced, cells moved through the simulation space at each iteration, guided by local flow vectors.

A probabilistic model was employed to simulate cell attachment. At each iteration, a random probability was drawn for each cell, and cells were attached only if this random value was smaller than the calculated attachment probability, *Pattach*. The attachment probability, *Pattach*, is defined by the following equation:

$$Pattach = A \cdot e^{(-flow_{speed})} \cdot B$$

where *A* represents the adhesion strength of the PDAC cells and *B* represents the adhesion properties of the background (i.e., the endothelial cells).

We applied an exponential relationship between flow speed and cells' adhesion strength, consistent with previous studies (Robert et al, 2011; Hammer and Apte, 1992). Flow vectors were sampled from grid points surrounding the cell. Simulations examined how attachment strength and flow speed affected cell attachment by varying these parameters across multiple runs. Cell sizes, densities, and time intervals were matched to experimental data.

To model B (the adhesion properties of the background), simulations were carried out using two environments: uniform and receptor-based backgrounds. In the receptor-based scenario, CD44 receptor maps were used to simulate varying attachment probabilities across the surface. Ten fields of view of endothelial monolayers stained for CD44 were used, with background subtracted and intensities rescaled between 0 and 1 using a 99th percentile normalization. Here, we assumed a linear relationship between the adhesion properties of the background and the amount of CD44 at the surface of the endothelial cells. For the uniform background, an average intensity value of 0.3, derived from the CD44 images, was used. At each iteration of the simulation, the adhesion properties of the background were computed for each cell, which were the average background values underneath the cell.

To further evaluate the effect of flow, simulations were performed with either a static flow field (where cell attachment did not alter the flow) or a dynamic flow field (where the flow was updated after each attachment). In the dynamic mode, the flow field was recalculated following each attachment using the Navier-Stokes equations, solved by the phiFlow Python package (PyTorch implementation) (Holl and Thuerey, 2024). The flow started with a uniform inflow speed at one boundary, with zero-gradient or open boundary conditions at the sides to simulate an infinite source. The attached cells were treated as obstacles in the flow, and the fluid dynamics were updated accordingly. The fluid properties were set to values measured for cell culture media with 10% serum at 37 °C (ibidi, application note 11), with a viscosity of 0.0007 Pa·s and a density of 993 kg/m³.

After each simulation, the coordinates of attached cells were analyzed using a modified version of Ripley's L function to evaluate how the simulation parameters influenced cell clustering. Only the simulations where at least two cells were attached were considered for these analyses. The code used for these simulations is available on GitHub (https://github.com/CellMigrationLab/AdhesionFlowSimulator) and archived on Zenodo (Jacquemet, 2024).

## PDAC lines genetic mutation analysis

The Cancer Cell Line Encyclopedia (CCLE) mutation annotation format (MAF) data was downloaded from The Cancer Dependency Map Portal (DepMap). The data was queried for mutations in genes such as KRAS, BRAF, CDKN2A, PIK3CA, SMAD4, ARID1A, TGFBR2, TP53, FN1, MUC16, and BRCA2 in the cell lines involved in this study. Tumor mutational burden (TMB) was defined as the number of mutations per megabase pair (Mbp), which approximates the size of the human exome at around 30 Mbp. The data was visualized using the ComplexHeatmap R package with R version 4.3.3.

## Statistical analysis

The numerical data used to make the figures and statistical summaries, including pairwise Cohen's d values and results from statistical tests, have been archived on Zenodo (https://doi.org/10.5281/zenodo.17232437). The code for performing randomization tests and *t* tests is available on GitHub (https://github.com/CellMigrationLab/PDAC_DL/). Randomization tests used Cohen's *d* as the effect size metric, with 10,000 iterations for each test. For *t* tests, data were assumed to follow a normal distribution, though this was not formally tested.

## Manuscript preparation

Figures were prepared using Fiji and Adobe Illustrator 2024 and CS6. Data were plotted using Microsoft Excel and GraphPad Prism v10. GPT-5 (OpenAI) and Grammarly (Grammarly, Inc.) were used as writing aids while preparing this manuscript. The author further edited and validated all text sections. GPT-5 did not provide references.

# Data availability

Plasmids generated in this study are being deposited in Addgene (https://www.addgene.org/Guillaume_Jacquemet/). A summary of all the deep learning models and associated training datasets used in this study can be found on the associated GitHub page (https://github.com/CellMigrationLab/PDAC_DL/) and have been deposited in the Zenodo community (https://zenodo.org/communities/pdac_dl). The tracking datasets generated in this study have been deposited on Zenodo. The raw microscopy images and the numerical data used to make the figures have been archived on Zenodo (https://doi.org/10.5281/zenodo.17232437). The authors declare that the data supporting the findings of this study are available within the article and from the authors upon request. Any additional information required to reanalyze the data reported in this paper is available from the corresponding authors. The code used in this study is available on the GitHub page (https://github.com/CellMigrationLab/PDAC_DL/) associated with this manuscript. The AdhesionFlowSimulator code is available on GitHub (https://github.com/CellMigrationLab/AdhesionFlowSimulator) and has been archived on Zenodo (Jacquemet G, 2024).

The source data of this paper are collected in the following database record: biostudies:S-SCDT-10_1038-S44318-025-00678-9.

## Peer review information

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

## Acknowledgements

This study was supported by the Research Council of Finland (338537 and 371287 to GJ; 325464 to JI; 338585 and 360775 to JRWC, and 332402 to GF), the Sigrid Juselius Foundation (to GJ and to JI), the Cancer Society of Finland (Syöpäjärjestöt; to GJ, MS, and JI), and the Solutions for Health strategic funding to Åbo Akademi University (to GJ). This research was supported by the InFLAMES and GeneCellNano Flagships Programme of the Research Council of Finland (decision numbers: 337530, 337531, 357910, 35791, and 337120). This work was also supported by the Finnish Cancer Institute (K. Albin Johansson Professorship, JI), the Research Council of Finland Centre of Excellence program (346131 & 364182 to JI), and the Jane and Aatos Erkko Foundation (JI). JRWC and MD were supported by the European Union's Horizon 2020 research and innovation program under the Marie Sklodowska-Curie grant agreement (841973 and 101108089). The Turku Collegium for Science, Medicine, and Technologies supported GF. The authors acknowledge Euro-BioImaging ERIC (https://ror.org/05d78xc36) for providing access to imaging technologies and services via the Finnish Advanced Microscopy Node (FiAM) in Turku, Finland (https://ror.org/023f5cj14). The Cell Imaging and Cytometry Core facility, the Zebrafish Core (both at Turku Bioscience, University of Turku, Åbo Akademi University, and supported by Biocenter Finland), Turku Bioimaging and the Genome Biology Unit (Research Programs Unit, HiLIFE Helsinki Institute of Life Science, Faculty of Medicine, University of Helsinki, Biocenter Finland) are acknowledged for services, instrumentation, and expertise. We thank Hamidi H and Harlepp S for critical reading of the manuscript. We also thank Laasola P. for blood isolation and preparation, and express our gratitude to the anonymous blood donors for their contributions to this study.

## Author contributions

**Gautier Follain**: Conceptualization; Resources; Data curation; Formal analysis; Supervision; Funding acquisition; Validation; Investigation; Visualization; Methodology; Writing—original draft; Writing—review and editing. **Sujan Ghimire**: Data curation; Formal analysis; Supervision; Validation; Investigation; Visualization; Methodology; Writing—review and editing. **Joanna W Pylvänäinen**: Data curation; Formal analysis; Visualization; Writing—review and editing. **Monika Vaitkevičiūtė**: Data curation; Formal analysis;

Investigation; Writing—review and editing. **Iván Hidalgo-Cenalmor**: Software; Methodology; Writing—review and editing. **Diana Wurzinger**: Investigation; Writing—review and editing. **Camilo Guzmán**: Formal analysis; Writing—review and editing. **James RW Conway**: Methodology; Writing—review and editing. **Michal Dibus**: Methodology; Writing—review and editing. **Jouni Härkönen**: Formal analysis; Writing—review and editing. **Sanna Oikari**: Resources; Writing—review and editing. **Kirsi Rilla**: Resources; Writing—review and editing. **Marko Salmi**: Resources; Methodology; Writing—review and editing. **Johanna Ivaska**: Conceptualization; Resources; Supervision; Funding acquisition; Visualization; Methodology; Writing—original draft; Project administration; Writing—review and editing. **Guillaume Jacquemet**: Conceptualization; Resources; Data curation; Software; Formal analysis; Supervision; Funding acquisition; Validation; Investigation; Visualization; Methodology; Writing—original draft; Project administration; Writing—review and editing.

Source data underlying figure panels in this paper may have individual authorship assigned. Where available, figure panel/source data authorship is listed in the following database record: biostudies:S-SCDT-10_1038-S44318-025-00678-9.

## Disclosure and competing interests statement

The authors declare no competing interests.

# Expanded View Figures

**Figure EV1. Summary of the FlowVision framework used in this study to investigate the interaction between circulating cells and endothelial cells under flow.** ▶

Illustration of the microfluidic setup (**A**) and the main imaging and analysis pipelines (**B**) used in the study to investigate the interaction between circulating cells and endothelial cell monolayers under flow. The main deep learning models used in this study, along with the primary tracking datasets, are also highlighted. Full details are available in the Methods section and on the GitHub page accompanying this paper (https://github.com/CellMigrationLab/PDAC_DL/), as well as archived on Zenodo, along with the training dataset used (https://zenodo.org/communities/pdac_dl).

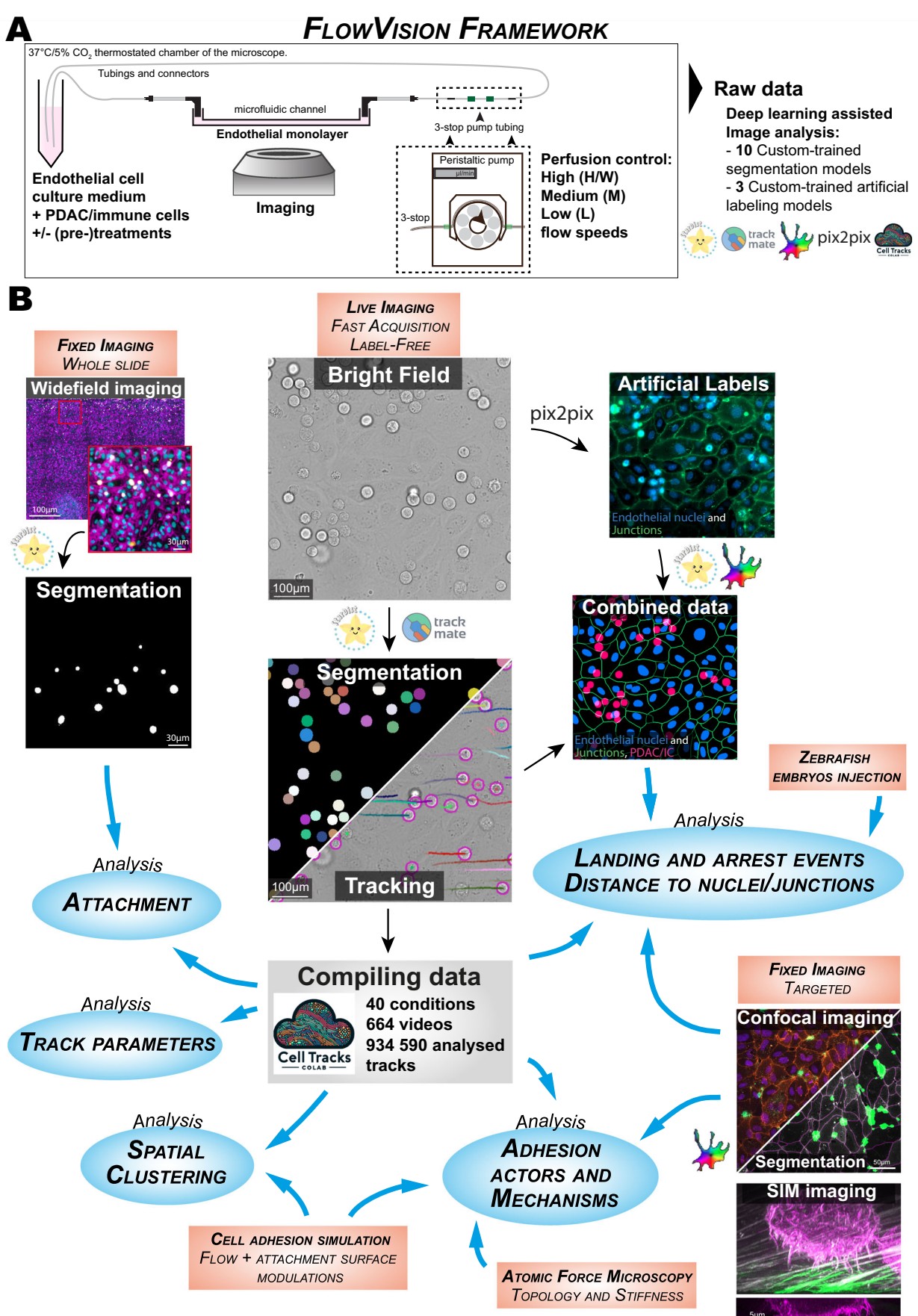

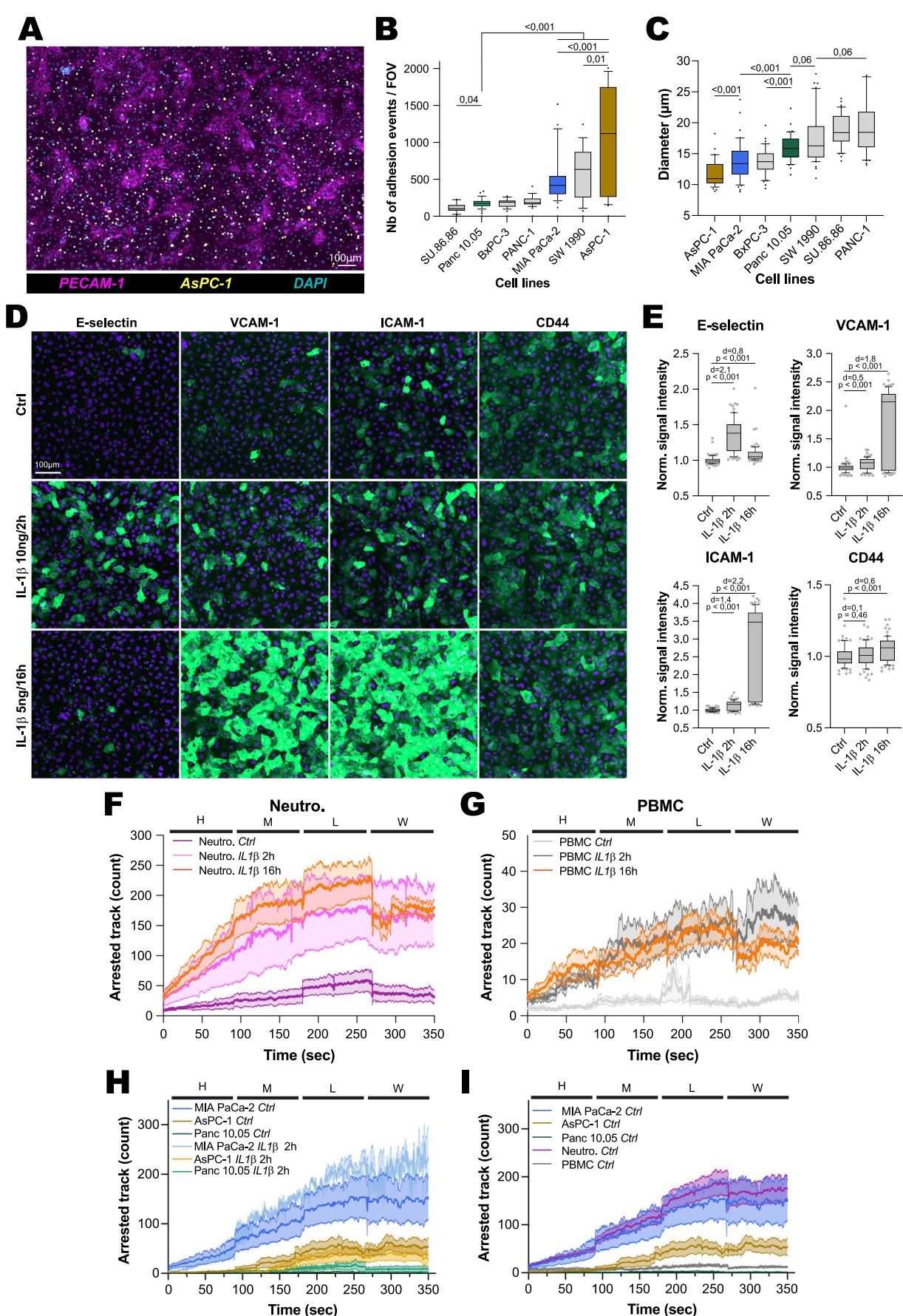

◀

**Figure EV2. Additional characterizations of PDAC and immune cells tracking results.**

(A, B) Adhesion of labeled PDAC cell lines to an endothelial monolayer without flow. Cells were fixed, stained with phalloidin and DAPI, and imaged using a spinning disk confocal microscope. (A) A representative image of the endothelial monolayer with attached PDAC cells. Scale bar: 100 µm. (B) The number of attached cells was quantified for each cell line. Results are presented as boxplots, where the whiskers extend from the 10th to the 90th percentiles. The boxes capture the interquartile range, with the median marked by a line within each box. Data points falling outside the whiskers are depicted as individual dots ($n = 10$–27 fields of view, 2–9 biological repeats). The P values were determined using a randomization test. MIA PaCa-2 vs AsPC-1, *P* value = 0.0001. MIA PaCa-2/SW1990/AsPC-1 (grouped on graph) vs Panc 10.05, *P* value = 0.0001. (C) Size distribution of PDAC cell lines. Cells in suspension were imaged using a brightfield microscope, and their diameters were manually measured using Fiji. Results are presented as boxplots, where the whiskers extend from the 10th to the 90th percentiles. The boxes capture the interquartile range, with the median marked by a line within each box. Data points falling outside the whiskers are depicted as individual dots ($n = 39$–40 cells). The P values were determined using a randomization test. Non-significant comparisons not shown. AsPC-1 vs MIA PaCa-2, *P* value = 0.0001. MIA PaCa-2 vs Panc 10.05, *P* value = 0.0001. BxPC-3 vs Panc 10.05, *P* value = 0.0001. (D, E) Endothelial monolayers, either untreated or treated with IL-1β (10 ng/ml for 2 h and 5 ng/ml for 16 h), were fixed and stained to visualize E-selectin, VCAM1, ICAM-1, and CD44. Stainings were performed without permeabilization to specifically label surface-accessible adhesion molecules. Images were captured using a spinning disk confocal microscope. (D) Representative fields of view are shown. Scale bar: 100 µm. (E) Quantification of the marker per field of view is presented. Intensities were normalized to the number of nuclei per field of view, as well as the average intensity measured in the control in each repeat. Results are presented as boxplots, where the whiskers extend from the 10th to the 90th percentiles. The boxes capture the interquartile range, with the median marked by a line within each box. Data points falling outside the whiskers are depicted as individual dots ($n = 45$ field of view, 3 biological repeats). The P values were determined using a randomization test. E-selectin, Ctrl vs IL-1β 2 h, *P* value = 0.0001 Ctrl vs IL-1β 16 h, *P* value = 0.0001. VCAM-1, Ctrl vs IL-1β 2 h, *P* value = 0.0001 Ctrl vs IL-1β 16 h, *P* value = 0.0001. ICAM-1, Ctrl vs IL-1β 2 h, *P* value = 0.0001 Ctrl vs IL-1β 16 h, *P* value = 0.0001. CD44, Ctrl vs IL-1β 16 h, *P* value = 0.0001. (F, G) The number of arrested neutrophils (F) or PBMCs (G) over time, in the presence or absence of IL-1β stimulation (10 ng/ml for 2 h and 5 ng/ml for 16 h). Bold lines indicate the average, and shaded areas represent the SD (4–7 biological repeats, see "Methods"). (H) The number of arrested PDAC cells over time for each cell line tested, with (2 h) and without IL-1β stimulation (PDAC Ctrl results were already displayed in Fig. 1). Bold lines indicate the average, and shaded areas represent the SD (2–7 biological repeats, see "Methods"). (I) The number of arrested PDAC and immune cells over time without IL-1β stimulation (PDAC Ctrl results were already displayed in Fig. 1). Bold lines indicate the average, and shaded areas represent the SD (3–8 biological repeats, see "Methods"). The numerical data and images used for this figure, as well as statistical summaries including pairwise Cohen's *d* values and results from statistical tests, have been archived on Zenodo (https://doi.org/10.5281/zenodo.17232437).

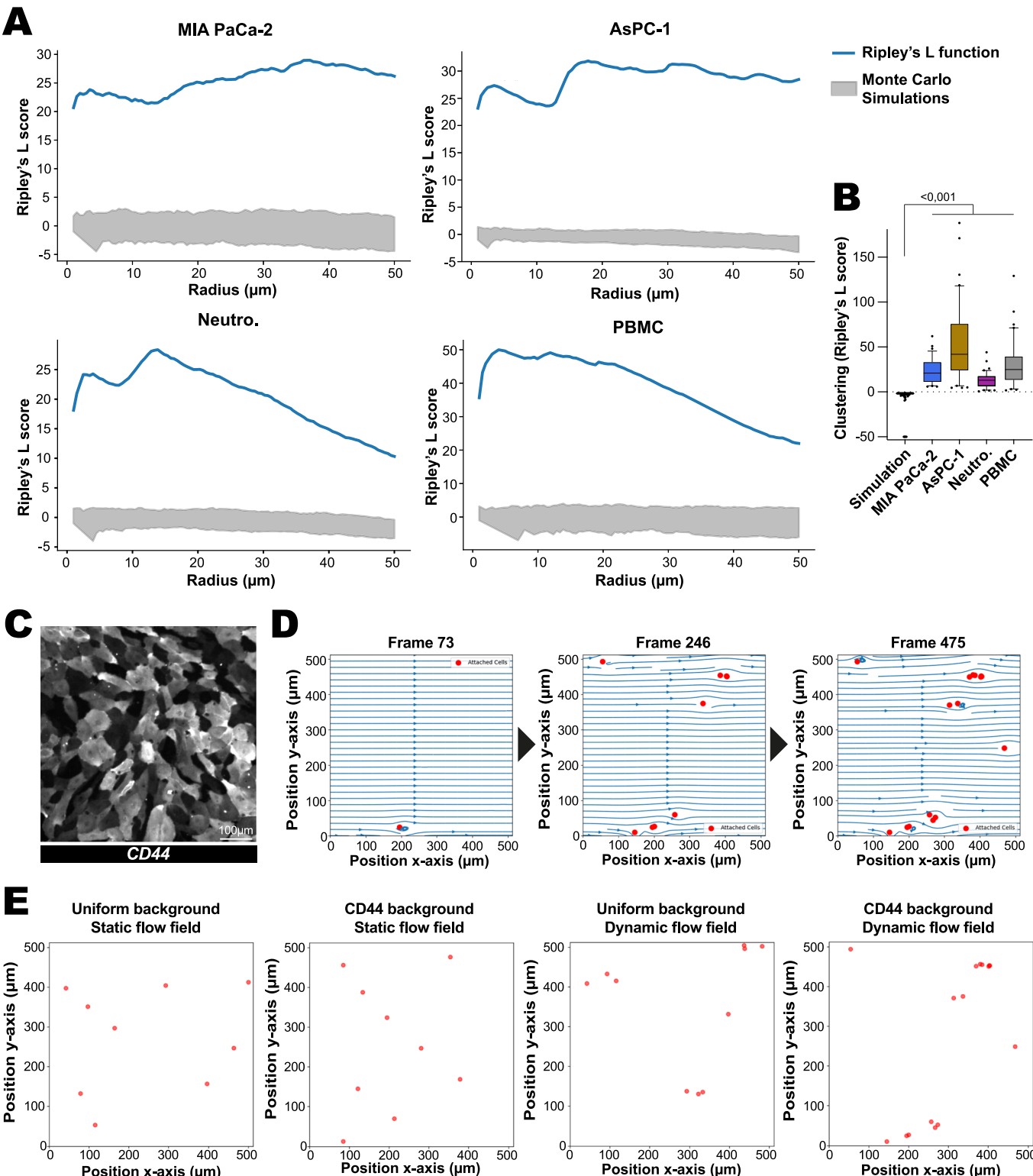

◀ **Figure EV3. Spatial clustering analysis of arrested PDAC and immune cells on endothelial monolayers.**

(A, B) Analysis of spatial clustering for arrested PDAC and immune cells on endothelial cell monolayers using Ripley's L function and Monte Carlo simulations. (A) These graphs depict the spatial distribution of arrested cell tracks, with a blue curve above the zero line indicating significant clustering at specific radii within the field of view. The Monte Carlo simulations provide a statistical framework to assess the significance of the observed clustering patterns. (B) Ripley's L scores and associated Monte Carlo simulations for each condition are presented as a heatmap ($n = 32$–$48$ fields of view, 8–12 biological repeats). The $P$ values were determined using a randomization test. (C–E) Simulations performed with our cell adhesion simulator to investigate factors driving PDAC cell clustering. (C) A representative field of view is used as the CD44 background. Scale bar: 100 µm. (D) Example showing changes in the flow field (blue lines) recalculated after each cell attachment at frames 73, 246, and 475. (E) Representative results showing the arrest locations of cells under medium flow speed with an adhesion strength of 0.2 (see also Movie EV6). Examples include simulations with a uniform background, a CD44 background, a static flow field, and a dynamic flow field. The numerical data and images used for this figure, as well as statistical summaries including pairwise Cohen's $d$ values and results from statistical tests, have been archived on Zenodo (https://doi.org/10.5281/zenodo.17232437).

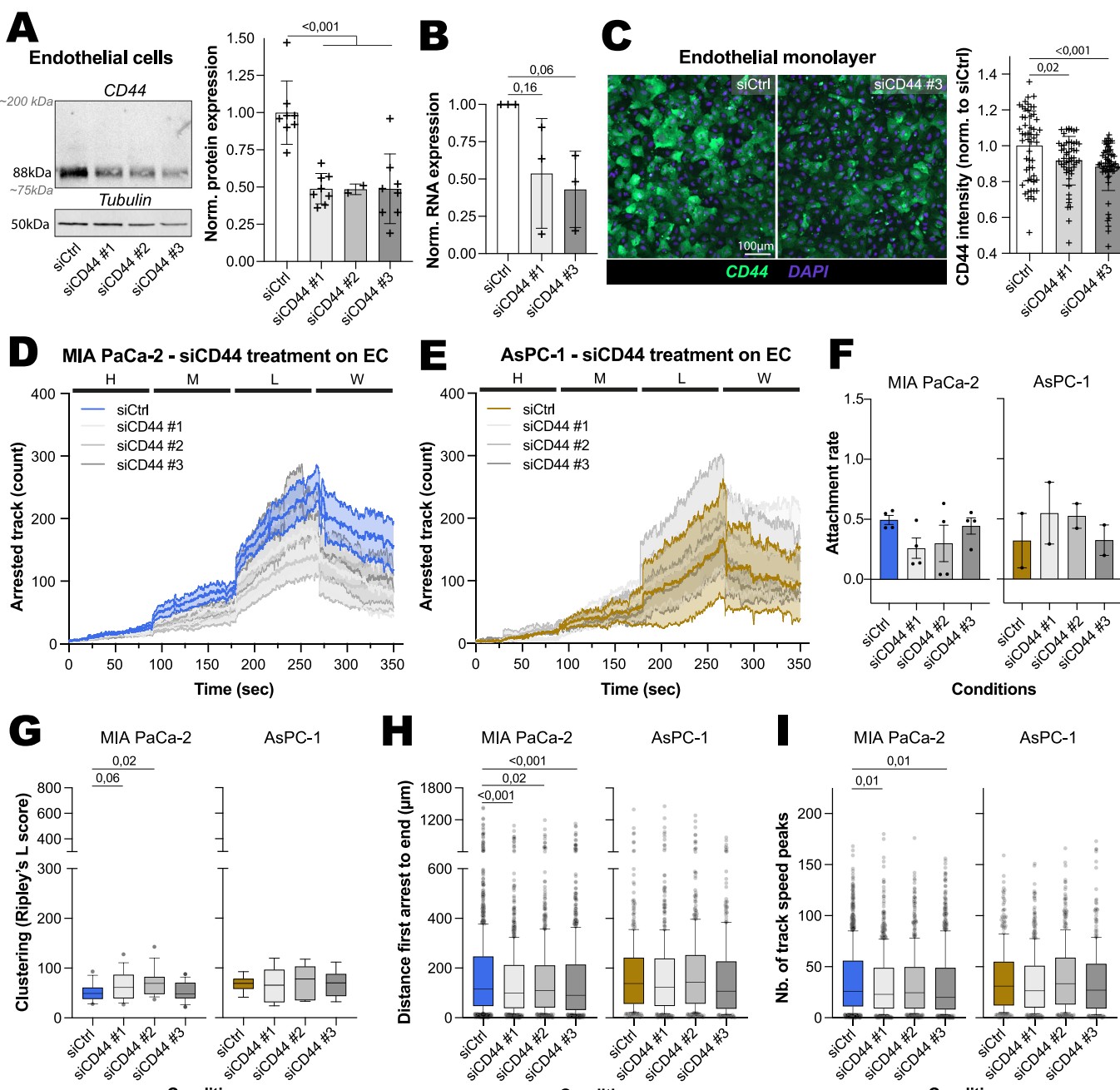

◀ **Figure EV4. Effect of targeting CD44 in endothelial cells on PDAC cell attachment to endothelial monolayers.**

(A–C) Validation of CD44 silencing in endothelial cells (ECs). ECs were treated with either CD44-targeting siRNAs (siCD44 #1, siCD44 #2 or siCD44 #3) or control siRNA (siCTRL), and CD44 levels were assessed using western blotting (A) ($n = 2$–8 biological repeats), qPCR (B) ($n = 3$ biological replicates), and immunofluorescence (C) ($n = 53$–60 field of views, 3 biological repeats). Scale bar: 100 μm. (D–F) Effect of CD44 silencing in endothelial cells on PDAC cell attachment to the endothelial monolayer. ECs were treated with either CD44-targeting siRNA or control siRNA (siCTRL) before perfusion of MIA PaCa-2 or AsPC-1 cells. Cell attachment was recorded as described in Fig. 1. The number of arrested MIA PaCa-2 (D) and AsPC-1 (E) cells over time is presented, with bold lines indicating the average and shaded areas representing the standard deviation (2–4 biological repeats, see "Methods"). The data is shown for different flow speeds. (F) The attachment rate for each cell line and condition at the medium (M) flow speed is displayed as a bar chart (mean ± SEM) with individual data points (2–4 biological repeats, see "Methods"). (G–I) Impact of CD44 silencing in ECs on the spatial clustering of PDAC cells. (G) Using the tracking data from (D, E) and as in Fig. 6B, all arrest events within each field of view were recorded. Modified Ripley's L functions were applied to quantify the spatial density of arrested cells. Ripley's L scores for each condition are presented as boxplots ($n = 8$–16 fields of view, 2–4 biological repeats, see "Methods"). (H, I) As in Fig. 3, the analysis was refined to include only cell trajectories showing a definitive arrest pattern. (H) The total distance traveled from the point of arrest to the end of the track is displayed for each condition. (I) The number of detachment peaks per condition is shown ($n = 331$–1248 tracks). The results are presented as boxplots, with whiskers representing the 10th to 90th percentiles and boxes indicating the interquartile range, with the median value marked. Outliers are displayed as individual data points. (A, B) The P values were determined using an unequal variance *t* test. siCtrl vs siCD44#1, P value = 0.0001. siCtrl vs siCD44#2, P value = 0.0002. siCtrl vs siCD44#3, P value = 0.0004. (C, G–I) The P values were determined using a randomization test. (C) siCtrl vs siCD44#3, P value = 0.0001. (H) siCtrl vs siCD44#1, P value = 0.0001. siCtrl vs siCD44#3, P value = 0.0001. The numerical data and images used for this figure, as well as statistical summaries including pairwise Cohen's *d* values and results from statistical tests, have been archived on Zenodo (https://doi.org/10.5281/zenodo.17232437).

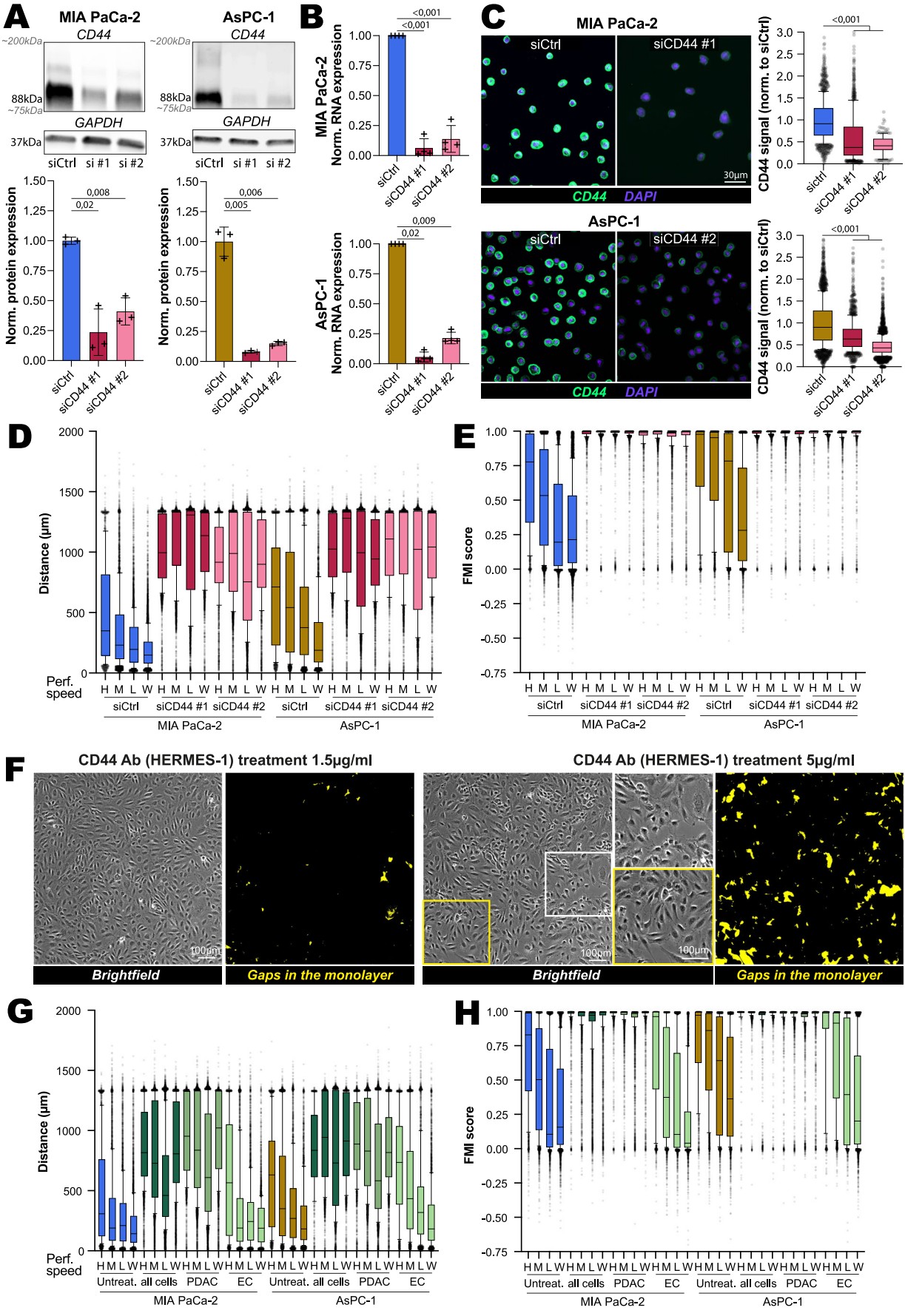

◀ **Figure EV5. Effect of targeting CD44 in PDAC cells on PDAC cell attachment to endothelial monolayers.**

(A–C) Validation of CD44 silencing in PDAC cells. MIA PaCa-2 and AsPC-1 cells were transfected with control siRNA (siCTRL) or two independent CD44-targeting siRNAs (siCD44 #1 or siCD44 #2), and CD44 levels were assessed using western blotting (A) ($n = 3$ biological repeats), qPCR (B) ($n = 4$ biological repeats), and immunofluorescence (C) ($n = 199$–2451 cells, 3 biological repeats). Scale bar: 30 µm. (D, E) Impact of CD44 silencing on PDAC cell attachment to the endothelial monolayer, related to Fig. 7A–C ($n = 2166$–15,056 tracks, 3 biological repeats). (D) The total distance traveled by siCTRL and siCD44-treated MIA PaCa-2 and AsPC-1 cells during perfusion at different flow speeds is displayed. (E) The Forward Migration Index (FMI) along the flow direction for each condition is shown. (F) The effect of anti-CD44 blocking antibody treatment on endothelial monolayer integrity. Endothelial cell (EC) monolayers were treated with two concentrations of anti-CD44 blocking antibody, followed by fixation and staining with phalloidin-Alexa488. Gaps in the monolayer were identified based on the absence of phalloidin signal, and they are shown as segmented masks to delineate these areas. Scale bar: 100 µm. (G, H) The effect of anti-CD44 blocking antibody on PDAC cell attachment to the endothelial monolayer, related to Fig. 7D–F. (F) The total distance traveled by MIA PaCa-2 and AsPC-1 cells during perfusion at different flow speeds is displayed for each condition. (G) The FMI along the flow direction for each condition is shown ($n = 2133$–15,014 tracks, 2–5 biological repeats, see "Methods"). (D, E, G, H) Results are presented as boxplots, with whiskers extending from the 10th to the 90th percentiles. The boxes capture the interquartile range, with the median indicated by a line within each box. Data points outside the whiskers are depicted as individual dots. The numerical data and images used for this figure, as well as statistical summaries including pairwise Cohen's *d* values and results from statistical tests, have been archived on Zenodo (https://doi.org/10.5281/zenodo.17232437).

