## [Peer Review File · The EMBO Journal]

Fast label-free live imaging with FlowVision reveals key principles of cancer cell arrest on endothelial monolayers

Gautier Follain, Sujan Ghimire, Joanna Pylvänäinen, Monika Vaitkevičiūtė, Ivan Hidalgo-Cenalmor, Diana Wurzinger, Camilo Guzmán, James Conway, Michal Dibus, Jouni Härkönen, Sanna Oikari, Kirsi Rilla, Marko Salmi, Johanna Ivaska, and Guillaume Jacquemet

Corresponding authors: Guillaume Jacquemet (guillaume.jacquemet@utu.fi) , Johanna Ivaska (johanna.ivaska@utu.fi)

Review Timeline:

Submission Date:	19th May 25
Editorial Decision:	22nd Jul 25
Revision Received:	30th Sep 25
Editorial Decision:	20th Nov 25
Revision Received:	28th Nov 25
Accepted:	10th Dec 25

Editor: Ieva Gailite

Transaction Report:

This manuscript was previously reviewed at another journal. As EMBO Press has a transfer agreement (including the identities of the referees) with that journal, revision was invited based on the reports from that previous external review. (Note: With the exception of the correction of typographical or spelling errors that could be a source of ambiguity, letters and reports are not edited. Depending on transfer agreements, referee reports obtained elsewhere may or may not be included in this compilation. Referee reports are anonymous unless the Referee chooses to sign their reports.)

Structured summary of the key revisions we are proposing

Effect and timing of IL-1 β stimulation

To address concerns regarding endothelial activation protocols:

- We will perform new perfusion experiments using PBMCs and neutrophils on HUVECs stimulated with IL-1 β for 2 hours and 16 hours, to determine the effect of prolonged inflammatory activation on adhesion dynamics.
- We will assess CD44 expression and localization in endothelial cells after both stimulation periods using immunofluorescence and western blotting.
- IL-1 β treatment conditions will be clearly annotated throughout the manuscript and figure legends.

CD44 and attachment of circulating immune cells

To further clarify the molecular contribution of CD44, we will perform blocking antibody experiments on primary neutrophils and PBMCs to determine whether CD44 contributes to immune cell adhesion under flow, complementing our data on cancer cells.

Adhesion molecules and attachment of PDAC cells

- We will perform immunostaining for VCAM-1 and E-selectin on endothelial cells and assess the possible correlation with their expression and the presence of cancer cells.
- We will profile VCAM-1-binding integrins in our PDAC cell lines to explore whether differences in receptor expression correlate with adhesion heterogeneity.

Heterogeneity of the Endothelial Monolayer

To investigate endothelial molecular diversity and its potential role in cell adhesion:

- We will perform co-immunostaining for CD44 and ICAM-1 on endothelial monolayers and quantify co-expression at the single-cell level.
- We will perform immunostaining for VCAM-1 and E-selectin after IL-1 β stimulation (2h and 16h) and quantify expression heterogeneity across the monolayer.

Single cell vs cluster attachment

To clarify the impact of cell clustering in flow on adhesion behavior:

- We will reanalyze our dataset to distinguish single cells from small clusters (2–4 cells) and quantify differences in arrest frequency and stability.
- The manuscript will clarify that large aggregates were excluded from segmentation and that our clustering analysis reflects spatial deviation from randomness, not cluster counts.

Textual and figure improvements:

We will implement all text changes listed in the point-by-point answer to the reviewer's comments. Key changes include:

- Figure labels will be updated to specify which PDAC cell line is used.
- Shear stress values corresponding to flow rates will also be reported in dyn/cm².

- The Discussion will include a "Limitations of the Study" paragraph that addresses the simplified geometry, substrate stiffness, absence of extravasation data, and use of HUVECs.
- We will acknowledge key prior studies on junctional attachment, ECM exposure (e.g., Wang et al., JCB 2004), and heterogeneity of cancer cell adhesion.

We would like to respectfully clarify our position regarding two areas of emphasis raised by the reviewers:

Reviewer 1 strongly advocates for the inclusion of in vivo experiments. While we agree that in vivo validation is an important step toward translational relevance, we believe such experiments fall outside the scope of the current study. Our primary aim is to present a robust, high-resolution, and accessible imaging and analytical platform to study circulating cell-endothelial cell adhesion under physiological flow conditions. As detailed in our response to the reviewer, this work is methodological in nature and focuses on generating reusable datasets and tools for the broader community.

Reviewer 2 raises several points related to IL-1 β stimulation. We would like to clarify that IL-1 β was explicitly used to mimic the inflammatory activation of the endothelium and facilitate comparisons between immune and PDAC cell interactions. While we agree with the relevance of testing different time points and have committed to expanding our experiments accordingly, we believe that additional requests, such as AFM analysis of endothelial stiffness, go beyond the intended scope of our study. These requests would not substantially strengthen the central conclusions, which focus on the dynamic behavior of circulating cells and the utility of our imaging pipeline.

Point-by-point answers to the reviewer's comments and associated revision plan:

We have highlighted our answers in blue, along with the proposed plan and editorial discussion points for your consideration.

Reviewer #1 (Remarks to the Author):

The mechanisms of transendothelial migration of cancer cells under physiological flow conditions as a key step of metastatic organ colonization remain incompletely defined. In addition, albeit general agreement assumes that cancer cells and leukocytes utilize similar steps and mechanisms of extravasation, conclusive experimental evidence supporting such homology is lacking.

Follain and coworkers here used a perfused HUVEC monolayer system with label-free microscopy and AI-based tracking to reliably quantify cell arrest, migration along and, in part, transmigration through the endothelial wall in various cancer cell (PDAC) and leukocyte models (PBMCs, neutrophils). Using a range of flow rates, PDAC models diverge in adhesion strength but consistently engage endothelial CD44 via cell-surface hyaluronan on the cancer cell to arrest at endothelial

junctions, while transmigration was not observed. Interfering with CD44/HA interaction significantly reduced adhesion. Similar arrest patterns were obtained for leukocytes, followed by effective transmigration.

We thank the reviewer for taking the time to read and provide feedback on our work. We are delighted to learn that the focus and quality of our work are appreciated.

The imaging pipeline developed by the authors provides a useful tool for acquiring real-time insights into cancer cell-endothelial cell interactions, however the approach is within the scope of already existing tracking approaches and, hence, does not break new ground. The descriptive results are potentially interesting on mostly similarities between cancer cells and leukocytes. The molecular findings on HA/CD44 interaction to mediate cancer cell arrest indicate that this approach may be suitable to identify receptor-ligand cascades mediating adhesion in the vascular bed.

We thank the reviewer for their thoughtful comments and appreciate the opportunity to clarify the innovative aspects of our analysis pipeline. While our approach integrates existing tools, it does so in a novel and comprehensive manner that breaks significant new ground in studying circulating cell interactions under flow.

Our automated imaging framework is fully label-free, enabling high-speed (25 Hz), long-term imaging under physiologically relevant flow conditions without the phototoxicity or perturbation associated with fluorescence-based approaches. This capability allowed us to capture dynamic arrest behaviors in real time, something not achievable in earlier studies, including our own, which relied on fixed endpoint imaging.

We systematically analyzed thousands of individual cell trajectories across hundreds of movies by combining deep learning-based segmentation (StarDist) with robust tracking (TrackMate). This approach enabled quantitative assessment of adhesion dynamics and post-arrest migration at single-cell resolution and on a large scale. Furthermore, this high-throughput analysis revealed spatial organization in cell arrest, including the unexpected discovery of clustering behavior on the endothelial monolayer.

Another noteworthy novel component is the use of artificial labeling via pix2pix deep learning models to infer endothelial landmarks, such as nuclei and junctions, from brightfield images. This approach allowed us to localize cell landing and arrest with subcellular precision, without fluorescent staining. To our knowledge, this represents one of the rare instances where artificial labeling has been successfully applied to study a biological phenomenon rather than merely for method development.

Crucially, our pipeline is modular, built using well-maintained and widely available open-source tools. This is a deliberate design choice. By avoiding highly customized solutions, we ensure that researchers studying related processes, such as immune cell trafficking, stem cell homing, or CTC behavior in different vascular contexts, can easily adapt and extend our workflow to meet their needs. We believe this flexibility and accessibility will significantly increase the impact and adoption of our approach within the broader cell biology and vascular biology communities.

Lastly, we emphasize that all code, trained models, and tracking datasets are shared openly via GitHub and Zenodo, supporting full reproducibility and broad dissemination.

In summary, we respectfully disagree with the assessment that our pipeline lacks novelty. By uniting label-free imaging, deep learning-based segmentation, artificial labeling, and high-throughput tracking

in a reproducible, modular framework, our approach offers new capabilities for investigating the mechanisms of cell arrest and adhesion under flow.

As main shortcomings, the data fall short to deliver truly novel insights and lack validation in vivo. The in vitro model lacks the geometries of vessel lumens with circumferential walls which enable embolic trapping of tumor cells to a greater extent than of leukocytes, and little evidence is provided that the spatial heterogeneity and intravascular arrest mechanisms hold up on a mammalian metastasis model with a mature vascular system. The comparative data between PDAC cells and leukocytes indicate similarities as well as disparate behaviors, however no compelling information on overlapping and divergent adhesion mechanisms are provided. Lastly, the role of CD44 in tumor cell clustering and metastasis was recognized before, hence the conceptual advance on addressing CD44/HA linkage here requires maturation to resolve potentially novel mechanisms of heterogeneity of seeding behaviors and their relevance for metastasis in vivo.

We thank the reviewer for their constructive critique and welcome the opportunity to clarify our study's conceptual focus and scope. We agree that in vivo validation remains essential for translating mechanistic insights into physiological and clinical relevance. However, the primary goal of our work is to deliver a methodologically robust and accessible platform to study the active adhesion mechanisms used by circulating cells, particularly cancer cells, during their interaction with the vascular endothelium under flow.

In vivo, cancer cells can arrest in blood vessels through mechanical occlusion and active adhesion involving specific molecular interactions with the endothelium. Our study is designed to dissect this latter process. To that end, we employed a simplified yet idealized in vitro system: microfluidic channels lined with endothelial monolayers that intentionally exclude geometric constraints such as circumferential vessel walls. This setup enables us to isolate and investigate adhesion-mediated arrest events without confounding effects of passive trapping.

While the model lacks the full architectural complexity of intact vasculature, it reproduces key biomechanical features, including flow-driven cell-endothelium interactions. Microfluidic platforms like ours have been widely adopted in vascular biology and immunology to study rolling, firm adhesion, and transmigration events. They are broadly recognized as powerful tools for mechanistic studies.

Regarding the biology, while CD44 has been previously implicated in metastasis, we reveal new aspects of its function in dynamic arrest of pancreatic cancer cells. Our data demonstrate that PDAC cells preferentially arrest at CD44-high endothelial sites via a hyaluronan-dependent mechanism, and that this interaction drives spatial clustering on the endothelium under flow. This clustering is further shaped by local flow disturbances introduced by previously adhered cells, a novel observation that links molecular adhesion with spatial self-organization.

Furthermore, although we do not claim to fully resolve all overlapping and divergent adhesion mechanisms between cancer cells and leukocytes, our comparative analysis provides a valuable foundation by offering side-by-side dynamic data on their arrest kinetics, spatial behavior, and response to inflammatory stimuli.

In summary, while in vivo studies will be essential to test the generalizability of these findings in complex tissue contexts, we believe that our work fills a critical gap by providing a scalable, high-resolution platform to study active adhesion mechanisms. This platform complements in vivo models and offers new directions for future research on circulating cell vascular interactions.

Revision plan for the editor: We will revise the manuscript text to better clarify the study's aim, the choice of the model system, and its limitations.

1. The flat system configuration does not replicate the tube-like concentric geometry of blood vessels. The geometry of the used IBIDI chambers (dimensions in 3D) should be detailed. Moreover, in vivo data suggest that active adhesion of tumor cells to endothelial cells may not be as significant as proposed, given that tumor cells typically undergo embolic arrest in capillaries rather than relying on active adhesion (doi.org/10.1038/nm.2072; [doi: 10.1007/978-3-319-95294-9_11](https://doi.org/10.1007/978-3-319-95294-9_11)). For embolic arrest, active adhesion may even be dispensable. Thus, the physiological relevance of the proposed mechanism of intravascular arrest remains to be demonstrated (see also point 5).

We thank the reviewer for raising these critical considerations and welcome the opportunity to clarify our rationale and interpretation. We fully acknowledge that our microfluidic model employs a simplified geometry that does not recapitulate the complete 3D cylindrical architecture of native blood vessels. The IBIDI μ -Slide VI channels used in our experiments have a rectangular cross-section measuring 3.8 mm in width and 0.4 mm in height. While simplified, such flat perfusion systems are widely accepted in the field and have proven to be extremely valuable for dissecting dynamic cell-endothelium interactions under controlled flow conditions.

We recognize the interplay between vascular architecture, flow forces, and adhesion that govern intravascular arrest. However, we respectfully disagree with the notion that embolic arrest is the dominant or sole mechanism responsible for cancer cell arrest in vivo. Most in vivo studies capture the extravasation phase, often hours after the initial arrest, by which point the local vasculature has typically remodeled in response to the presence of tumor cells. This vascular response can lead to local vessel constriction and an apparent embolic profile, potentially overstating the role of mechanical occlusion while underrepresenting the contribution of active adhesion mechanisms.

Numerous studies, including our own, have demonstrated that cancer cells can arrest in vessels that are not geometrically occluded. For example, in Follain et al. (Dev Cell 2018), we used intravital microscopy and nanometric flow tracers to show that blood flow continues upstream and downstream of arrested tumor cells in capillary beds, indicating that occlusion is not occurring (see figure below). Moreover, we observed that non-adhesive beads of comparable size arrest at significantly lower frequencies, approximately four times less than cancer cells, indicating that passive size restriction alone cannot explain arrest behavior. These findings strongly support the role of active adhesion in mediating early arrest.

Furthermore, the necessity of adhesive interactions between endothelial and tumor cells for successful vascular arrest and subsequent extravasation has been demonstrated by several groups (Gassmann et al., IJCD 2009; Reymond et al., JCB 2012; Au et al., PNAS 2016; Offeddu et al., Comm. Biol. 2021). Using diverse tumor models, these studies have shown that disrupting specific adhesion molecules, including CD44, VCAM-1, ICAM-1, and integrins, significantly impairs cancer cell retention and transendothelial migration. Our current study builds upon this work by analyzing these interactions with high temporal and spatial resolution under defined flow conditions.

While we agree that our model does not encompass all anatomical features of the vasculature, it is ideally suited for isolating and analyzing active adhesion processes without confounding physical trapping. As such, our findings provide valuable mechanistic insight into early adhesion events that

would be difficult to resolve in vivo and likely complement rather than contradict the observations from capillary-rich tissues.

Overall, we believe that reduced flow and constricted vasculature similarly affect the arrest and stable adhesion: both slow down circulating cancer cells and prolong the contact time between these cells and the endothelium. This provides more time for active adhesion to occur, allowing the cells to engage their receptors and stop stably. Decoupling flow from physical constraints is nearly impossible in vivo. This is why studying the arrest in a non-constricting “vessel” necessitates the use of simple 2D systems.

We hope this clarifies our position and the physiological relevance of our model and findings. We agree that integrating in vivo validation will be an important future direction. However, we believe our current work offers meaningful mechanistic insight into adhesive arrest that is directly applicable to metastatic progression.

Brief figure legend

Extracted panels from Follain et al. *Dev. Cell* 2018, figure 5. (L) Experimental workflow and idealized representation (left). Embryos are injected with CTCs and 100-nm beads and imaged at high speed, before single-particle tracking analysis (middle). Corresponding quantification of tracks' straightness and velocity in three scenarios (open, partially closed, and closed) and in silico representation of laminar flow around arrested tumor cell (right). Values are mean ± SD. *p < 0.05. ns, not significant.

Extracted panels from Follain et al. *Dev. Cell* 2018, supplementary figure S2. (A) Experimental set up, representative image and quantifications of the 10-μm beads injection. [live recordings post-injection] Values are mean ± SD. *p < 0.05. ns, not significant.

Revision plan for the editor:

- We will revise the manuscript to clarify the study's aim, the choice of the model system, and its limitations more effectively.

- If necessary, we can also provide illustrative examples showing that pancreatic cancer cells can become arrested in the lung vasculature within vessels in the absence of occlusion, further demonstrating the physiological relevance of our study.

2. It is unclear how the experimental setup was constructed. Was shear flow applied to the endothelial cells before the introduction of cancer cells, or was it applied only briefly before? The strength of fluid flow influences molecular networks of HUVEC monolayers, thus the metrics used should be quantitative and reproducible. The actual shear forces should be revealed for the “high” and “low” flow speeds. How do these levels to arterial and venous in vivo vascular beds? Thus, the used conditions should be linked to realistic scenarios cancer cells would be confronted with in vivo.

We thank the reviewer for this valuable comment. We fully agree that flow dynamics significantly influence endothelial cell architecture and signaling, and we have ensured that our conditions are standardized and reported clearly and reproducibly.

In our experimental design, HUVEC monolayers were cultured under static conditions for 4 days until visual confluency was reached, and maintained for an additional 24 hours to ensure junctional maturation. The channels were then connected to the microfluidic perfusion system just 5 minutes before the introduction of circulating cells. This brief pre-flow period was sufficient to fill and equilibrate the channels without substantially altering the organization of the monolayer. We were particularly mindful that prolonged exposure to flow can reprogram endothelial cell morphology and gene expression, as demonstrated in our previous work (Follain et al., Sci Rep 2021). To maintain consistency across experiments, we ensured that all perfusions adhered to an identical protocol and duration.

Our system was calibrated to produce flow velocities ranging from 100 to 400 $\mu\text{m}/\text{sec}$, which are physiologically relevant and fall within the range observed in human capillaries and post-capillary venules (Follain et al., Nat Rev Cancer 2020). Importantly, due to the relatively large height of our microfluidic channels (400 μm), these flow speeds generate modest wall shear stresses, ranging from approximately 0.1 to 0.3 dyn/cm^2 at the endothelial surface. These values are lower than the typical shear stress in capillaries but are comparable to other microfluidic models of vascular adhesion (e.g., Cerutti et al., Cell Rep 2024) and are already sufficient to limit the adhesion of circulating cells in our 2D configuration, as shown in Figure 1 of the manuscript.

Additionally, we deliberately avoided high arterial shear stress conditions (e.g., $>10 \text{ dyn}/\text{cm}^2$), which are known to induce endothelial alignment and morphological polarization in the direction of flow. Since we focus on dissecting the adhesive behavior of circulating cells under capillary-like conditions, we aimed to model flow environments where cell arrest is most physiologically relevant.

In conclusion, we carefully chose our flow conditions to reflect the permissive vascular environments encountered during early metastatic dissemination, particularly in capillary and venular beds where extravasation typically occurs. All flow rates and shear values were standardized across experiments to ensure reproducibility and comparability of results.

Revision plan for the editor:

- We will revise the manuscript to more clearly highlight the model system's strengths and limitations.

3. The vessel beds used for extravasation in vivo differ for tumor cells (arterioles, capillaries) and leukocytes (postcapillary venules), and barely overlap. HUVECs cultured for several days are selected as a generic but immature endothelium. Likewise, the staining in zebrafish embryos represents developing, immature vessels. Thus, it is unclear to which extent CD44 expression and heterogeneity thereof represents an in vivo-relevant phenomenon in mature mammalian vessels. Quiescent endothelial cells in arteries, veins, and organ capillaries might exhibit varying expression of tight junctions and differ in glycocalyx composition and structure compared to HUVEC. For instance, venous endothelial cells are less permeable and more resistant to extravasation than the activated endothelium often modeled by HUVEC. Thus, the key findings need to be verified in mature vascular models with in vivo-like geometries. In addition, depending on the predicted site of extravasation, in line with the used shear forces, organ-specific or microvascular endothelial cells that more closely replicate the specialized characteristics of the endothelium should be included.

We fully agree that the endothelial phenotype varies significantly across the vascular tree and among different tissue beds, and that these distinctions play an important role in regulating the extravasation of both tumor and immune cells in vivo.

That said, the focus of the present study is not to recapitulate the full physiological diversity of the mammalian vasculature, but rather to introduce a novel, high-resolution, label-free live imaging and analysis platform that allows systematic quantification of adhesion dynamics under controlled flow conditions.

While HUVECs do not fully replicate the structure or function of mature arterial, venous, or tissue-specific microvascular endothelium, they remain one of the most commonly used and widely accepted models for initial studies of cell-endothelium interactions. Their use in our study allows direct comparison to a substantial body of published literature. It provides a standardized background against which perturbations, such as cytokine stimulation or receptor blocking, can be systematically assessed. Furthermore, the reproducibility and accessibility of HUVEC cultures make them an ideal substrate for a methodological framework that is designed to be shared and extended by the wider community.

Regarding the zebrafish data, we acknowledge that the vasculature in embryos represents a developing network. However, we chose this model because it offers unmatched in vivo optical accessibility and is well established for real-time analysis of circulating tumor cell behavior under flow.

We fully agree with the reviewer that future work should incorporate tissue-specific or organotypic endothelial models, and potentially incorporate advanced 3D microfluidic systems that better replicate in vivo geometries and biomechanical constraints. Indeed, one of the long-term goals of our study is to provide a modular pipeline that can be easily implemented across such advanced systems to dissect how vascular specialization influences cell behavior.

In summary, while our current model is intentionally simplified, we believe it is ideally suited for revealing core mechanistic principles of cell adhesion under flow. We view our work as a first and necessary step toward integrating this methodology into more physiologically complex systems, and

we hope it will serve as a valuable resource for researchers exploring vascular heterogeneity and tissue-specific metastasis in the future.

Revision plan for the editor:

- We will revise the manuscript to clarify the study's aim, the choice of model system, and its limitations more effectively.

4. The authors show (in Videos 1 and 2) that Mia PaCa-2 arrest on endothelial cells to a greater extent than AsPC-1. They further show that these two PDAC models express similar levels of CD44 and that CD44/HA silencing/blocking/depletion equally impairs the ability of cancer cells to attach to endothelial cells. Given these findings, they claim that CD44 is the sole mediator of arrest on the endothelium, which may ignore the difference between these cells.

We thank the reviewer for this comment and for raising critical points. There appears to have been a misinterpretation of our data, and we apologize for not being sufficiently clear. As shown in **Fig. S7A** of the current manuscript, the expression level of CD44 is not the same in the two cell lines (see below for convenience). It is significantly lower in AsPC-1 compared to MIA PaCa-2, fully correlating with the differences in their adhesion capability.

Fig. S7A: CD44 expression levels in AsPC-1, MIA PaCa-2 and Panc 10.05

Revision plan for the editor:

- We will revise the manuscript to better highlight the difference in CD44 expression between our cell lines.

Observations from the videos further suggest that Mia PaCa-2 cells tend to form clusters and flow in clusters, and that cluster kinetics differ from single cells. The authors should further explore clusters versus single-cell arrest to better clarify if the differences between the PDAC models are linked to other factors beyond CD44/HA.

We appreciate the reviewer for highlighting this interesting point. Indeed, Mia PaCa-2 cells tend to aggregate within the microfluidic tubing and often enter the flow chamber as clusters. We agree that this behavior may contribute to their distinct arrest dynamics and deserves further investigation.

We intentionally excluded large cell aggregates from our current analyses, as these clusters, often comprising tens to hundreds of cells that form sizable aggregates, cannot establish adhesive contacts with the endothelial monolayer. Instead, they tend to displace previously arrested cells due to their

bulk and inertia, a phenomenon we repeatedly observed during live imaging. To ensure the integrity of our dataset, we trained our deep learning-based segmentation algorithm to omit these large aggregates.

That said, smaller clusters, typically comprising 2–4 cells, are more frequently observed engaging with the endothelium and successfully arresting. Our current analysis did not explicitly distinguish these small clusters from single cells; both were included under the general category of "arrested cells." We fully agree with the reviewer that separating these two populations could provide valuable insight into factors affecting arrest efficiency.

Therefore, we will reanalyze our dataset focusing on this comparison. Since our imaging and tracking pipeline already extracts cell-level features such as area and shape, we can easily stratify events based on cell clustering status and quantify differences in arrest frequency, location, and duration between single cells and small clusters. This additional analysis will enhance the manuscript.

Revision plan for the editor:

- We will reanalyze our data to examine the difference in arrest potential between single cells and small clusters.
- The text will be modified to explain the differences in behavior between single cells and clusters.

5. The in vivo relevance and translatability of the key findings in a mammalian model remain unclear. The study observes a high degree of variability in endothelial cell adhesion between the seven PDAC cell lines, with some lines, such as AsPC-1 and MIA PaCa-2, exhibiting high adhesion to endothelial cells, while others, like SW1990 and PANC1, show reduced adhesion under flow conditions. However, it is unclear how these differences in adhesion are determined by CD44, other adhesion molecules downstream of CD44, and the cell clustering state in vitro. Likewise, the relevance of these findings for the metastatic potential of these cell lines in vivo remains unclear. In addition, in vivo, only a very small fraction of cancer cells but most if not all leukocytes extravasate. Thus, the model may inaccurately overrepresent adhesion (and transmigration?). The results using zebrafish embryos neither test seeding in a mature vascular system, nor provide molecular data to confirm the importance of CD44/HA interactions. At 48 hours post-fertilization, zebrafish embryos possess rather immature blood vessels, thus the relevance of this model is unclear. In addition, neither the relevance of this adhesion mechanism for transmigration and metastasis formation was explored. Thus, key results including interventions need to be shown for adhesion, transmigration, and outgrowth in a mammalian system.

We thank the reviewer for these critical points. As detailed in our responses above, the scope of this study is to investigate the adhesion machinery used by cancer cells on endothelial monolayers by introducing a label-free imaging and analysis platform for dissecting the dynamics of cancer and immune cell adhesion under flow. While we acknowledge that in vivo validation in a mammalian model is essential to fully establish the relevance of CD44/HA interactions for metastasis and extravasation, this lies beyond the scope of the current work. Our goal here is to provide the community with a high-resolution, reproducible method to study early adhesive events. The platform is broadly adaptable and will be instrumental in future efforts to explore molecular specificity, cell line heterogeneity, and in vivo relevance in more complex and organotypic models.

For the editorial board of the EMBO Journal: We believe these concerns regarding in vivo metastasis events are interesting and important but are well beyond the scope of our manuscript.

Revision plan for the editor:

- We will revise the manuscript to better clarify the study's aim, the choice of the model system, and its limitations.

6. Despite their claims in the introduction on similarities and differences of endothelial cell binding by cancer cells and leukocytes, the experiments do not explore shared and divergent mechanisms between these principal cell types, both at cellular and molecular level.

We respectfully disagree with the reviewer's assessment. Our manuscript includes direct comparisons between cancer cells and immune cells at both the cellular and behavioral levels, as shown in Figures 2, 3, 4, and 6. These analyses reveal significant similarities in arrest dynamics, such as preferential junctional attachment, as well as key differences, particularly in responsiveness to inflammatory cues and the capacity to transmigrate.

In response to Reviewer 2 and to further strengthen the manuscript, we plan to extend our analysis to include comparisons of the molecular machinery. Specifically, we will conduct CD44 blocking experiments in primary neutrophils and PBMCs to assess the role of CD44 in immune cell adhesion to endothelial cells. We believe these additions will provide a more comprehensive comparison of cancer and immune cell interactions with the endothelium.

Revision plan for the editor:

- To further strengthen our manuscript and address the reviewer's comments, we will perform additional experiments to assess the role of CD44 in immune cell adhesion. Specifically, we will conduct CD44 blocking treatments on primary neutrophils and PBMCs before perfusing them over endothelial monolayers. These experiments will help clarify the contribution of CD44-mediated adhesion in immune cells and broaden the comparative analysis of cancer and immune cell interactions with the endothelium.

Minor points:

7. The limitations of the chosen system design should be discussed in detail. Which context can be drawn to in vivo reality of cancer cell embolic arrest? Which additional work will be required to link this approach to in-vivo relevance?

We thank the reviewer for this comment. We will add a limitations paragraph to the manuscript to address those points and clarify our text. Our understanding of the importance of emboli in vivo was addressed previously.

Revision plan for the editor:

- We will revise the manuscript text to clarify the study's aim, the choice of model system, and its limitations.

8. In page 7 authors write “Using Ripley’s L function and Monte Carlo simulations, we revealed a pronounced tendency for arrested cells to cluster especially at higher flow speed (Fig. 6B and Fig. S6A-B). Notably, PDAC cells showed more clustering than neutrophils. AsPC-1 cells clustered the most, and the MIA PaCa-2 cells clustered similarly to PBMCs (Fig. 6B and Fig. S6B). This finding suggests a non-random pattern of arrest on endothelial monolayers, with cells preferentially arresting at specific hotspots.”. However, this claim does not seem to align with the observations in the videos (see e.g., video 1). The authors should clarify the discrepancy between their claim from simulated data and the visual data presented.

We thank the reviewer for highlighting this potential confusion. To clarify, the data presented in Figure 6B and Supplementary Figures S6A-B do not come from simulated datasets; instead, they are derived from quantitative spatial analyses performed directly on the experimental video data. We employed Ripley’s L function and Monte Carlo spatial point pattern analysis to assess the tendency of arrested cells to cluster in comparison to a random spatial distribution.

Importantly, this analysis does not quantify the number of clusters; rather, it measures whether the observed spatial arrangement of cells deviates significantly from randomness. Our data indicates a non-random arrest pattern or “hotspot” behavior across the monolayer. We will revise the manuscript text to clarify this distinction and to prevent any misinterpretation that our clustering data were based solely on simulation.

Revision plan for the editor:

- We will revise the manuscript to clarify these analyses.

9. In page 10 the authors write “Interestingly, CD44 is known to form homophilic interactions (Liu et al., 2019), and MIA PaCa-2 and AsPC-1 cells exhibited high CD44 surface expression (Fig. 6C).” but Fig. 6C displays immunofluorescence staining of PDAC cells on the endothelium. The correct figure panel to cite may be Figure S8C or Figure 8A.

We thank the reviewer for identifying this point and apologize for the lack of clarity. In Figure 6C, we used immunofluorescence to demonstrate that both MIA PaCa-2 and AsPC-1 cells exhibit strong CD44 staining during their interaction with the endothelium, with signal intensity significantly higher than that observed on the endothelial cells themselves.

To support and quantify this observation, we included the corresponding western blot (Figure S7A), which confirms elevated CD44 protein levels in both PDAC cell lines. We will revise the text to ensure accurate cross-referencing between figures and to clarify the distinction between imaging and protein quantification.

Revision plan for the editor:

- We will revise the manuscript to ensure accurate cross-referencing between figures and to clarify the distinction between imaging and protein quantification.

10. The number of tracks, cells, fields of view, and biological replicates is clearly stated in the figure legends, ensuring transparency. However, the substantial variation in these numbers across

experiments—ranging from 30 to over 15,000 for analyzed cells or tracks or fields of view, and from 2 to 12 biological replicates—raises concerns about the homogeneity of the experimental design. The authors should further clarify the rationale for these variations.

We thank the reviewer for this important observation. We acknowledge that the number of tracks, cells, and biological replicates varies across experiments and agree that this warrants clarification. We will revise the manuscript to include more explicit information about our statistical strategy, as suggested by Reviewer 2, and will expand the Methods section to clarify the rationale behind this variability.

Specifically, for the condition “MIA PaCa-2 + IL-1 β ,” we conducted two biological replicates, as described in the Methods section (“Cell tracking and quantitative analysis of tracking data”). Given the strong adhesive behavior of MIA PaCa-2 cells, these two replicates yielded a very high number of tracks ($n = 7045$), which we considered sufficient to obtain statistically meaningful and representative results. That said, we are open to including a third biological replicate for this condition if the editorial board deems it necessary, and we are happy to incorporate this into our revision plan.

Regarding Figure 7L, which shows only 35 tracks for the CD44 blocking condition, this reflects the expected biological outcome: CD44 inhibition markedly reduces cell adhesion, as seen in Figure 7D–E. After applying our filtering criteria to ensure that only stable adhesion events are analyzed, very few tracks remain. This accurately represents the biology of the system rather than indicating an inconsistency in the experimental design.

Broadly, the variability in track counts across conditions reflects the biological heterogeneity of adhesion among different cell types and treatments, ranging from highly adherent PDAC lines to weakly interacting or inhibited cells. Our statistical analysis strategy addresses these differences by utilizing robust non-parametric randomization tests and reporting effect sizes via Cohen’s d , which facilitates reliable comparisons even when sample sizes differ. These methods are particularly well-suited for datasets with a large number of observations (as is the case in most of our analyses) and ensure our conclusions are supported by rigorous quantitative analysis.

If the robustness of our data analysis remains a concern, we are fully prepared to conduct additional analyses, including resampling strategies such as sub-sampling, to further confirm the stability and reproducibility of our findings across conditions with varying sample sizes. However, we would also like to emphasize that, in the spirit of transparency and reproducibility, we have made all our numerical data, tracking datasets, and statistical code openly available. This allows for full reanalysis by the community and reflects our commitment to scientific rigor. We believe this openness further strengthens the robustness of our conclusions and invites collaborative scrutiny, which we welcome.

Revision plan for the editor:

- We will revise the manuscript to include more detailed information about our statistical strategy.
- If necessary, we will provide additional statistical analyses using sub-sampling.

Reviewer #2 (Remarks to the Author):

In this methods manuscript, the authors describe an automated pipeline to analyze attachment of

cancer cells and leukocytes to endothelial cells under flow conditions in vitro, without using fluorescent dyes to label cells. Particularly useful if their ability to outline endothelial cell junctions and nuclei without fixing and labeling, and to determine where cells attach with respect to endothelial expression of a selection of three adhesion molecules. The methodology advances the field and will be useful for those using similar approaches to investigate how leukocytes/cancer cells attach to endothelial cells (provided it is easy to access for non-machine learning experts), although not so far for those studying these processes in vivo.

Overall, the manuscript is well written although the statistical analysis is difficult to understand for a non-expert in the way it is presented. Some ANOVA results would help here.

We thank the reviewer for their positive feedback. We are delighted to learn that our efforts to make deep learning approaches accessible to biologists are well received.

Limitations of method: The authors need to describe the limitations of their approach, and that it has not been tested under physiological conditions or in vivo. First, endothelial cells are grown and analyzed on a hard inflexible surface of glass/plastic, very unlike the softer more flexible environment of capillaries and small blood vessels in vivo, where cancer cells arrest. Second, it is also a large tube that is far wider than a capillary in vivo. Third, the authors use HUVECs, which are not microvascular endothelial cells and have different properties. Fourth, they are not quantifying extravasation of cancer cells but just adhesion. Indeed, in some cases, cancer cells are initially stay in blood vessels and proliferate there, rather than extravasate e.g. Almhedi et al., Nature Med 2000. The authors should consider this in their Introduction/Discussion sections.

We thank the reviewer for this thoughtful and constructive suggestion. We fully agree that acknowledging the limitations of our in vitro system and its relevance to physiological conditions is crucial. In response, we will add a dedicated “Limitations of the Study” paragraph to the Discussion section of the revised manuscript.

Revision plan for the editor:

- We will add a “Limitations of the Study” paragraph to the Discussion section of the revised manuscript.

Main points for revision:

1. CD44 and cancer cell attachment: this is very well characterized from previous studies, which should be referenced in the Introduction i.e. that this is already known.

We thank the reviewer for this helpful suggestion. We agree that previous studies, particularly in breast cancer models, have established a role for CD44 in mediating cancer cell adhesion to the endothelium. We will revise the Introduction to acknowledge and properly reference this foundational work.

Our study, however, offers novel mechanistic insights within the context of pancreatic cancer, a biologically and clinically distinct setting. While CD44 has been broadly implicated in adhesion processes, the precise nature of its interaction, whether through CD44–CD44 homophilic binding, CD44–hyaluronan (HA) heterophilic binding, or a combination of both, remains unresolved. Our data provides new evidence in support of a “sandwich” model, where HA functions as a bridging molecule between CD44 molecules on neighboring cells, enabling both homophilic and heterophilic interactions. This mechanistic inference is bolstered by our high-resolution imaging and the statistical power provided by our label-free, high-throughput tracking platform. We will ensure the revised manuscript clearly articulates how our findings build upon and advance the current understanding of CD44-mediated adhesion in the broader context of cancer biology.

Revision plan for the editor:

- We revise our manuscript to clearly articulate how our findings build upon and advance the current understanding of CD44-mediated adhesion in the broader context of cancer biology.

2. Extracellular matrix patches in vivo: a very relevant paper, much earlier than those referenced in the manuscript, is Wang H et al. JCB 2004 (exposed laminin patches in living mouse lung), which also described the importance of integrins in cancer cell attachment to endothelial cells and arrest in vivo.

We thank the reviewer for highlighting this important and relevant earlier study. We will include a citation to Wang et al., JCB 2004 in the main text to acknowledge their contribution to the concept of available extracellular matrix (ECM) patches for arrested cancer cell arrest.

While we recognize the significance of their findings, we note that the evidence presented in that study, particularly regarding ECM exposure, is primarily based on a single transmission electron microscopy (TEM) image (Figure 4). In contrast, our study provides a more detailed and temporally resolved analysis of ECM (specifically fibronectin) patch exposure on endothelial monolayers. Through high-content imaging and quantitative analysis, we offer a broader and more robust view of the distribution and dynamics of these ECM features under flow conditions. We believe this adds substantial new insight to the concept first proposed by Wang et al. and helps to extend and validate their pioneering observations.

Revision plan for the editor:

- We will revise our manuscript to clearly acknowledge earlier foundational studies, including Wang et al. (JCB, 2004), that introduced the concept of extracellular matrix exposure.

3. Videos: it is clear from the videos that MIA PaCa2 cells frequently flow through the chambers as large clumps containing many cells. This could well impact on their ability to attach – it is unlikely that a whole cell clump will attach effectively to endothelial cells. What did the authors do to minimize the formation of cell clumps for this cell line? They should analyze and describe in the text how these clumps affect attachment.

We thank the reviewer for this important observation. Indeed, MIA PaCa-2 cell clustering in suspension has posed a challenge throughout our study. As noted, these cells frequently form large aggregates that persist during perfusion and can be clearly seen in the videos.

We explored several approaches to minimize clustering prior to perfusion, including EDTA washes, extended trypsinization, and alternative cell dissociation buffers. While some of these treatments reduced clustering immediately after detachment, we observed that clustering often re-emerged during perfusion, likely within the reservoir or tubing system, resulting in little to no net improvement in the experimental setting. Importantly, since divalent cations (such as Ca^{2+} and Mg^{2+}) are also required for the function of many adhesion molecules, more aggressive anti-clumping treatments risk impairing the physiological adhesion behavior of MIA PaCa-2 cells. For these reasons, and to ensure consistency across conditions, we standardized our preparation protocol across all cell types (see Methods).

As discussed in response to Reviewer 1, large MIA PaCa-2 aggregates (often involving dozens to hundreds of cells) do not adhere to the endothelial monolayer. On the contrary, due to their mass and momentum, they tend to sweep away previously arrested cells. To avoid including these artifacts in our quantitative analysis, we trained our deep learning-based segmentation algorithm to exclude large clusters specifically.

However, smaller clusters, typically 2–4 cells, are observed to interact productively with the endothelium and form stable arrests. In our current analysis, these were not separated from single cells. We will therefore reanalyze our dataset to distinguish single cells from small clusters based on size and morphology.

Revision plan for the editor:

- We will reanalyze our data to examine the difference in arrest potential between single cells and small clusters.
- The text will be modified to explain the differences in behavior between single cells and clusters.

In addition, from the quantification and images it appears that adherent and arrested MIA PaCa2 cells form more clusters than other cell types, yet the authors state on p. 7 that they clustered similarly to leukocytes. This is surprising and indeed the impression from the images is that not all the MIA PaCa2 cells in clusters are being recognized by the algorithm. The authors should comment on this potential drawback to the quantification.

We thank the reviewer for raising this point and for the opportunity to clarify our analysis. As discussed in our response to Reviewer 1, our clustering analysis does not aim to quantify the number of clusters or the number of cells per cluster. Instead, it assesses whether the spatial distribution of arrested cells deviates significantly from a random pattern, using Ripley's L function and Monte Carlo simulations. This allows us to identify non-random "hotspot" behavior, where cells preferentially arrest in proximity to one another more often than would be expected by chance. We agree that the manuscript text on page 7 could be more precise in distinguishing between apparent visual clustering and the statistical clustering tendency assessed in our analysis. We will revise the text to clearly explain this distinction

and to prevent any misinterpretation regarding the scope and limitations of the clustering quantification.

Revision plan for the editor:

- We will revise the manuscript to clarify these analyses.

4. It is surprising that the authors only stimulate the endothelial cells with IL-1 β for 2 hours, which is very short compared to standard stimuli designed to increase, for example, PBMC adhesion and transmigration. Why did they choose this short time point? They should also test a longer time (e.g. 16 hours) when all adhesion molecules for PMBCs will be strongly upregulated e.g. E-selectin, VCAM-1, ICAM-1, and test differences between unstimulated and stimulated HUVECs at that timepoint. Does IL-1 β treatment alter CD44 expression or % of HUVECs expressing CD44?

We thank the reviewer for this important point. In our current study, we used a 2-hour IL-1 β stimulation protocol that consistently induces a rapid endothelial activation response characterized by increased cellular protrusiveness and altered morphology, features we specifically aimed to capture as part of the early activation phase. This effect tends to diminish after 3–4 hours post-treatment. Our choice was guided by previous literature showing time-dependent, transient surface expression patterns of key endothelial adhesion molecules in response to IL-1 β (Smith et al., JCI 1988; Pober et al., J Immunol 1986; Bhogal et al., J Immunol 1986; Bevilacqua et al., Science 1989; Haraldsen et al., J Immunol 1996; Chen et al., J Virol 1997), suggesting that the endothelial response is dynamic and not necessarily stronger or more uniform at longer stimulation times.

That said, we fully agree that comparing responses at later time points is important, especially to better characterize the upregulation of adhesion molecules relevant for PBMC and neutrophil arrest and transmigration. In our revised manuscript, we will investigate the impact of stimulation with IL-1 β for 16 hours.

Revision plan for the editor:

- We will conduct new live perfusion recordings using PMBCs and neutrophils on HUVEC monolayers stimulated with IL-1 β for 2 and 16 hours. If substantial differences are observed, we will expand these experiments to include PDAC cells.
- We will assess CD44 expression and localization in HUVECs after IL-1 β treatment for 2 and 16 hours. This will be accomplished through immunofluorescence and western blot analysis to determine whether IL-1 β modulates CD44 surface levels or distribution in endothelial cells.

5. Given the importance of endothelial selectins for leukocyte rolling and VCAM-1 for cancer cell and leukocyte attachment, the authors should also stain endothelial cells for these molecules after IL1 β stimulation. VCAM-1 has been implicated in the endothelial attachment of a number of cancer cell lines.

We thank the reviewer for this excellent suggestion. We agree that investigating VCAM-1 and E-selectin expression following IL-1 β stimulation in our model is highly relevant and will add valuable context to our findings on cancer and immune cell adhesion. We will, therefore, perform

immunostaining of HUVEC monolayers for both VCAM-1 and E-selectin after IL-1 β treatment. We will quantify the spatial distribution and heterogeneity of VCAM-1 expression across the endothelial surface, using an analytical approach similar to that shown in Figure 6C for CD44. Additionally, we will assess the expression of VCAM-1-binding integrins (such as $\alpha 4\beta 1$) in our panel of PDAC cell lines to explore whether differences in integrin expression correlate with variations in adhesion capacity.

Revision plan for the editor:

- Immunostaining of HUVEC monolayers for VCAM-1 and E-selectin following IL-1 β stimulation, along with quantification of expression patterns and heterogeneity.
- We will perform immunostaining for VCAM-1 and E-selectin on endothelial cells and assess the possible correlation with their expression and the presence of cancer cells
- Expression profiling of VCAM-1-binding integrins in our PDAC cell lines to assess their potential contribution to endothelial attachment.

6. It is not stated in some figure legends whether or not endothelial cells were stimulated with IL1b prior to analysis. If not, then the comparison +/- IL-1 β is important in places. For example, in Figure 5 it is clear from the images that the endothelial cells have abundant stress fibers. If the cells were quiescent and growth factors reduced (did the authors do this in their experiments?) then they would not have stress fibers unless stimulated for several hours with a pro-inflammatory cytokine such as IL1 β . It is important to test whether the topology and stiffness of the endothelial cells (AFM) is altered under their conditions by IL-1 β stimulation.

We thank the reviewer for this observation and apologize for any confusion caused by the incomplete labeling of IL-1 β treatment status. We will carefully review all figure legends and will ensure that IL-1 β stimulation is clearly and consistently indicated throughout the revised manuscript.

Regarding Figure 5, we confirm that no IL-1 β stimulation was used. The endothelial cells were maintained under standard conditions with complete endothelial growth medium containing 2% FBS and growth supplements, as detailed in the Methods section. We did not use growth factor-reduced media. The presence of stress fibers under these conditions is consistent with prior studies reporting that even in the absence of pro-inflammatory cytokines, such media can sustain moderate levels of cytoskeletal organization and tension in HUVEC monolayers.

We will revise the manuscript and relevant figure legends to clarify the specific culture conditions and treatment status in all experiments.

Regarding the suggestion to perform AFM measurements to assess changes in endothelial stiffness following IL-1 β stimulation, we appreciate the scientific merit of this idea. However, we respectfully note that IL-1 β activation is not a primary focus of our study. Given the technical demands and time required to perform AFM analysis, and considering the limited direct impact such data would have on the main conclusions, we believe this lies outside the scope of our current work.

Revision plan for the editor:

- We will revise the manuscript to ensure that IL-1 β stimulation status is clearly and consistently indicated in all figure legends and relevant sections of the text.

- We will include a statement in the Discussion acknowledging that it would be of interest in future studies to investigate whether IL-1 β stimulation modulates endothelial monolayer stiffness and topology, for example, through approaches such as atomic force microscopy.

7. When tracking cells that are 'in focus', are most of them traveling at a lower speed than the flow rate, simply because they are close to the wall of the microfluidic chamber and the flow rate is lower here? Presumably the cells moving at this rate are not attached and hence not migrating on the endothelial surface. This is not completely clear from the text description.

We thank the reviewer for this insightful question. Indeed, when tracking cells that are 'in focus,' they are situated near the bottom of the microfluidic channel, within approximately 10–25 μm of the glass substrate, where the local flow speed is lower than the maximal flow speed available in the channel due to the gaussian (Poiseuille) flow profile (See Figure S1, and below).

In addition, the theoretical maximum flow speed in our channel geometry, assuming laminar flow, occurs at the centerline of the channel and is higher than what is observed near the wall. However, our speed calibration was performed directly within the imaging window using non-adherent cells flowing through BSA-coated channels. Under these conditions, we measured a maximum flow speed of approximately 400 $\mu\text{m}/\text{sec}$ in the near-wall region, which establishes the relevant baseline for the cells we track.

Cells in this region are not necessarily adherent; instead, they are suspended in flow and located where physical contact with the endothelial monolayer is possible. Our tracking approach is designed to capture cells at this interface, enabling us to observe the complete sequence of behaviors, including free flow, deceleration, landing, arrest, and potential detachment. Cells that maintain high velocity throughout their trajectory are categorized as non-adherent, while those that decelerate and remain below defined speed thresholds are classified as arrested.

We will revise the manuscript to clarify the relationship between flow profile, imaging depth, and our calibration strategy to ensure this point is clearly understood.

Figure S1B: Theoretical flow profile in our channels.

Revision plan for the editor:

- We will revise the manuscript to clarify the relationship between the flow profile, imaging depth, and our calibration strategy.

8. For neutrophil tracking, did the authors include neutrophils migrating under the endothelial cells (transmigrated) as well as on the surface?

We thank the reviewer for this question. In our study, we focused specifically on the arrest phase of circulating cells, as cancer cells, unlike neutrophils, do not transmigrate within the 10-minute window of our live imaging experiments. To ensure consistency across conditions and cell types, we designed our deep learning segmentation models to detect only those cells that remained rounded and in contact with the endothelial surface.

Neutrophils that underwent transmigration rapidly changed shape, spreading beneath the endothelial monolayer. These morphological changes rendered them undetectable to the segmentation model designed for surface-adherent, rounded cells. Consequently, transmigrated neutrophils were intentionally excluded from the analysis. This enabled us to directly compare the arrest behavior of cancer cells and immune cells on a consistent basis, within the same imaging timeframe and under similar segmentation criteria.

We will revise the Methods section to clarify this in the manuscript.

Revision plan for the editor:

- We will revise the manuscript to clarify that only the arrest phase is quantified.

9. A table delineating the mutations in each of the PDAC cell lines used should be included for reference. The authors should also state whether there is any connection between mutational status and adhesion to HUVECs/ CD44 expression.

We thank the reviewer for this valuable suggestion. We will use the Cancer Cell Line Encyclopedia (CCLE) database (DOI: 10.1038/nature11003) to compile and present the mutational profiles of the PDAC cell lines included in our study. This information will be summarized in a new figure panel in the revised manuscript for ease of reference (see initial figure draft below).

Based on the available data, we do not observe a definitive correlation between specific oncogenic mutations and the adhesion behavior of these cell lines under flow conditions.

Regarding CD44 expression levels (or isoforms), to our knowledge, there is no established link between CD44 expression and the specific mutational burden of PDAC cell lines. However, this remains an intriguing question, and future studies could explore whether particular mutations influence CD44 regulation or function in the context of vascular adhesion.

A tentative figure displaying the mutational burden found in the PDAC lines used in this study.

Revision plan for the editor:

- We will add a figure showing the mutational burden found in the PDAC lines used in our study.

10. P. 6: the authors state that the basal ECM should presumably be inaccessible to their antibodies, but it should be accessible where it is exposed and indeed this is shown in their images (and see Wang et al. JCB 2004).

We thank the reviewer for this clarification and fully agree. We apologize if our original wording was unclear. As shown in our images and consistent with previous work (e.g., Wang et al., JCB 2004), the basal extracellular matrix is indeed accessible where it is exposed, and we successfully detect fibronectin (FN) in these regions.

In the manuscript, our intention was to convey that we did not observe apical fibronectin patches on the endothelial surface accessible to circulating cells. We will revise the text to clearly distinguish between apical and exposed basal FN, and to prevent any misunderstanding regarding antibody accessibility or ECM localization.

Revision plan for the editor:

- We will update the manuscript to clearly indicate that basal fibronectin (FN) patches are accessible when they are exposed, as demonstrated in our imaging data. We will revise the wording to eliminate any confusion and ensure it is clear that our results specifically refer to the absence of detectable apical FN, not basal FN.

11. Others have previously reported heterogeneity of cancer cell lines in their ability to attach to endothelial cells in vitro, which should be referenced in the Introduction.

We thank the reviewer for this important point. We agree that previous studies have reported heterogeneity among cancer cell lines in their capacity to adhere to endothelial cells in vitro. We will revise the Introduction to acknowledge this prior work and provide appropriate references after conducting a careful literature review.

Revision plan for the editor:

- We will incorporate additional references indicating that others have previously reported heterogeneity in the capacity of cancer cells to interact with endothelial monolayers.

12. In a few panels, the authors do not state which PDAC cell line was used e.g. Figure 1B.

We thank the reviewer for pointing this out and apologize for the oversight. We have carefully reviewed the relevant figure panels and updated them to specify the PDAC cell line used in each case. This will ensure clarity and consistency across the manuscript. The following modifications have been made:

- **Figure 1B:** “PDAC” will be labeled as MIA PaCa-2
- **Figure S2A, B, D:** “PDAC” or “cancer cells” will be labeled as MIA PaCa-2
- **Figure S3A:** “PDAC” will be labeled as MIA PaCa-2
- **Figure 4C, 4F:** “PDAC” will be labeled as MIA PaCa-2
- **Figure 5H, 5J:** “PDAC” will be labeled as MIA PaCa-2

These changes will be incorporated in the revised figures and legends to ensure transparency and accurate interpretation.

Revision plan for the editor:

- The figures will be updated accordingly

13. The authors should convert their chamber flow speeds to approximate dynes/cm² which are commonly used by other groups.

In the revised manuscript, we will explicitly equate our flow speed with the resulting shear force at the surface of the endothelial monolayer.

Revision plan for the editor:

- The manuscript will be updated accordingly

14. Other groups have reported that leukocytes/cancer cells mostly arrest close to endothelial cell junctions and the extension of filopodia by arrested cancer cells/leukocytes; their work should be acknowledged on p. 6 and p. 7.

We thank the reviewer for this important point. We will revise the Introduction to acknowledge this prior work and provide appropriate references after conducting a thorough literature review.

Revision plan for the editor:

- The manuscript will be updated accordingly

15. Figure 4, zebrafish study. How many cells were analyzed in how many different zebrafish? Is this just one image? Given that the authors have not tested their automated tracking algorithms in vivo, it is not clear why this single image is relevant unless quantitative analysis is added.

We thank the reviewer for this helpful comment. The zebrafish panel in Figure 4 originates from a single embryo and serves as a representative example to visually illustrate a key spatial constraint observed in vivo: the close proximity between arrested cancer cells and endothelial cell–cell junctions.

Although this panel lacks support from in vivo automated tracking or quantification, we included it as a qualitative illustration to complement our in vitro observations. We agree that further biological replicates and quantitative analysis could be undertaken if necessary, and based on our previous experience with similar experiments, we would expect the results to consistently demonstrate 100% of arrested cancer cells in contact with endothelial junctions.

Specifically for the editorial board of EMBO Journal: We are happy to discuss the inclusion of this figure. While this panel is illustrative, we are confident that additional biological repeats would yield the same conclusion: that in vivo, all arrested cells make contact with endothelial junctions due to vascular geometry. We are also willing to remove this figure if deemed necessary, but we believe it provides useful contextual support for the in vitro findings.

16. Figure 5J: why did the authors allow the MIA PaCa2 cells to interact with HUVECs for 30 minutes here, much longer than their other assays? This should be explained in the text.

We thank the reviewer for this helpful comment. The extended time point used in Figure 5J reflects the specific requirements of the structured illumination microscopy (SIM) protocol, designed to capture the stable adhesion phase of MIA PaCa-2 cells interacting with the endothelial monolayer.

To achieve this, we perfused MIA PaCa-2 cells for 10 minutes, followed by a 20-minute wash with cell-free media at the same flow rate. This ensured that only firmly adherent cells remained on the endothelium by the time of fixation and imaging, enabling us to focus on cells that had established stable adhesive contacts. Based on prior data from our lab examining cancer cell transmigration, we know that MIA PaCa-2 cells typically do not initiate transmigration until 1–2 hours post-attachment. Therefore, at the 30-minute mark, the cells remain surface-attached and have not yet begun extravasation.

We will update the manuscript to clarify the experimental design and rationale.

Revision plan for the editor:

- The manuscript will be updated accordingly

17. Figure 6: what % of ICAM-1-expressing endothelial cells also express CD44?

We thank the reviewer for raising this interesting point. To address it, we plan to perform co-immunostaining of endothelial monolayers for both ICAM-1 and CD44. This will enable us to assess the degree of co-expression at the single-cell level and determine whether these two adhesion molecules are enriched in the same endothelial subpopulations. We will include representative images and quantification of co-expression percentages in the revised manuscript.

Revision plan for the editor:

- We will perform co-immunostainings of endothelial monolayers for ICAM-1 and CD44 to evaluate their co-expression.

In panel B, why were the data for control and IL-1 β stimulation pooled? Can they also be shown separately?

We thank the reviewer for this question. The data in panel B were originally pooled to increase statistical power, as each field of view provides a single measurement, and separating the conditions initially led to limited data points per group. Additionally, our preliminary analysis showed no significant difference between IL-1 β -treated and control conditions, which supported pooling for the initial representation.

However, we fully agree that disaggregating the control and IL-1 β -stimulated datasets could provide greater clarity. We will reanalyze the data and present the two conditions separately in the revised figure to ensure transparency and allow readers to directly assess any subtle differences.

Revision plan for the editor:

- We will reanalyse the data by disaggregating the control and IL-1 β -stimulated datasets.

18. Did the CD44 blocking antibody reduce adhesion of PBMCs or neutrophils as well as PANC cells? This is an important control to include, because CD44 should be specific to cancer cell attachment.

We thank the reviewer for highlighting this exciting point. To directly investigate the potential role of CD44 in immune cell adhesion, we will experimentally address this in our revision. Specifically, we will perform CD44 antibody blocking treatments on primary neutrophils and PBMCs before perfusing over endothelial monolayers. This will enable us to assess whether CD44 contributes to immune cell arrest under flow and compare its role in immune versus cancer cell adhesion within the same assay framework.

Revision plan for the editor:

- We will conduct live perfusion experiments to evaluate the role of the anti-CD44 blocking antibody on immune cell adhesion.

Dear Guillaume and Johanna,

Thank you for transferring your manuscript together with reviewer comments from the previous assessment round and your revision plan to the EMBO Journal. As discussed, I have now received input from two additional reviewers with expertise in microfluidics-based models of vasculature and their imaging, which are included below for your information.

As you will see, the reviewers appreciate the relevance of the label-free imaging approach to the research field, but also indicate that further details on the imaging pipeline would need to be provided, and its broader applicability to other imaging contexts would need to be more clearly tested or spelled out. They also have provided some comments on the input from the previous two reviewers highlighting the aspects they find particularly important to address.

Taken together, I find that there is sufficient support from the reviewers' side to invite you to revise the manuscript according to your revision plan, while also integrating these additional comments. Please let me know if you would like to discuss the revision in more detail via email or phone/videoconferencing.

We generally allow three months as standard revision time, which can be extended to six months in the case of major revisions. Should you foresee a problem in meeting this deadline, please let us know in advance to discuss an extension.

As a matter of policy, competing manuscripts published during this period will not negatively impact on our assessment of the conceptual advance presented by your study. However, please contact me as soon as possible upon publication of any related work to discuss the appropriate course of action.

When preparing your letter of response to the referees' comments, please bear in mind that this will form part of the Review Process File and will therefore be available online to the community. For more details on our Transparent Editorial Process, please visit our website: <https://www.embopress.org/page/journal/14602075/authorguide#transparentprocess>. Please also see the attached instructions for further guidelines on preparation of the revised manuscript.

Please feel free to contact me if have any further questions regarding the revision. Thank you for the opportunity to consider your work for publication, and I look forward to your revision.

With best wishes,

Ieva

- a point-by-point response to the referees' comments, with a detailed description of the changes made (as a word file).
- a word file of the manuscript text.
- individual production quality figure files (one file per figure)

- a complete author checklist, which you can download from our author guidelines (<https://www.embopress.org/page/journal/14602075/authorguide>).

- Expanded View files (replacing Supplementary Information)

We realize that it is difficult to revise to a specific deadline. In the interest of protecting the conceptual advance provided by the work, we recommend a revision within 3 months (20th Oct 2025). Please discuss the revision progress ahead of this time with the editor if you require more time to complete the revisions.

Referee #1:

The manuscript from Follain and colleagues has developed an imaging analysis pipeline based on bright field images combined with deep-learning approaches to characterize and quantify cell adhesion to endothelial cells under flow. The methodology could be very useful for all labs studying blood cells interactions in vitro, not only for cancer, but also for immunologists, vascular biologists and infections biologists. The authors use the developed pipeline to study pancreatic ductal adenocarcinoma (PDCA) adhesion to HUVEC cells seeded in a microfluidic chamber through CD44 and HA. The manuscript is complete, highly quantitative and with extensive methodological description. I support its publication with some important points for revision.

1. The manuscript is intended to be a resource article for scientists using the pipeline. However, all the introduction and most of the discussion focuses on the research question that the authors are trying to address. The title on the EMBO portal "Flow dynamics and CD44-HA interaction modulate cancer cell arrest on endothelial monolayers" also does not refer to the resource, and the title in the main manuscript "Fast label-free manuscript reveals key roles..." better reflects, but would make the resource not findable for other researches not working on cancer. I would suggest that the authors rewrite these key sections of the manuscript, including the abstract to make it more appealing and exciting for other researchers not working on cancer that might be interested in this tool.
2. Similar to the comment above, I would suggest the authors to better describe in the results the experimental pipelines used. Including track mate, pix2pic, Stardist, cell tracks collab. If the paper is intended to be a resource for biologists not so familiar with these software / packages, the technical overview of the study now in supplemental should have higher relevance, as well as its description. Right now, this is just limited to no more than 10 lines on the results.
3. I would specify in the results of Fig1 which of the cancer cell lines used in the study correspond to the mutation.
4. The authors use very nice schematic figures to describe the methodology, except for key parameters that are used throughout the manuscript. Track speed, total distance traveled and especially forward migration index. Again, this would be useful for biologists.
5. I would recommend the authors to state in the text physiological capillary flow velocities, how these compare to their flow conditions as well as introduced wall shear stress values, as suggested by both reviewers. Although the difficulty of modelling simultaneously relevant WSS and flow velocities in most commercial and non-commercial microfluidic chips is a big limitation in the field, this should be acknowledged in the discussion.
6. The pipeline is not useful to study transmigration, and this process is barely studied in the manuscript. I would recommend to remove references to it in the results and probably include it in the discussion.
7. I agree with reviewer 2 that the IL1-b treatment is short and could account for the lack of differences, both on PDCA and PBMEC/neutrophils. As suggested, I would suggest to repeat the experiment after a longer incubation. Alternatively, I would use TNF-a which is much widely used, and presents a stronger and faster response on the endothelium.
8. The authors should specifically state that the experiment in zebrafish is a proof of concept and that future iterations if the pipeline could include quantification in vivo.
9. The HA findings are interesting but confusing. The imaging seems to indicate that HA is localized on the PDCA and not on the endothelial cells. How would the staining look on an endothelial cell monolayer without PDCA? If CD44 is mediating the interaction on the PDCA side (siRNA experiments and blocking antibody are quite conclusive) how does the CD44 and HA interaction occur. How do the authors explain that the digestion on HA in EC would inhibit the interaction given the low expression in HUVEC. How can both inhibition in HA and CD44 give such strong results in PDCA. Are they cooperative?

10. I agree with both reviewers that the authors should state the limitations of the model clearly in the discussion, including the limitations on the use of HUVEC, low WSS (as mentioned in comment 5), lack of vessel-like architecture/geometry and quite important, the lack of soft ECM. This is especially important in the context of their claim that PDCA and blood cells bind to junctional regions because of the high stiffness in these regions. Given the thinness of endothelial cells junctions (most of the time lower than 500nm), how do the authors ensure the AFM measurements are not on the substrate. In my opinion, given the scientific claims regarding on PDCA interactions with HUVEC, the substrate stiffness is an important one.

11. I agree with reviewer 2 that the quantification of VCAM, and E-selectin would improve the manuscript.

12. I suggest that the anti-CD44 blocking experiment on leukocyte (PBMC, neutrophils) suggested by reviewer 1 should be done in combination and without and anti-ICAM-1 antibody, given that it is one of the major receptors for this interaction.

13. Similar to point 1 and 2, Montecarlo simulations are difficult to understand and a more detailed description would be useful.

Referee #2:

Monolayers of endothelial cells (HUVEC) are exposed to cancer cells (PDAC) and leukocytes under various flow regimes. The Authors combine multiple deep-learning resources (Cellpose, StarDist, Trackmate) in order to establish a pipeline for investigating circulating cells arrest in relation to subcellular endothelial arrangements.

The results appear robust and clear, with the Authors demonstrating HA-CD44 adhesion mechanisms between PDAC cells and endothelium and both PDAC and leukocyte adhesion occurring near cell junctions.

Label-free analysis is important in the context of immune cells and other precious patient samples - wherein labelling could alter cell behavior and/or viability. As it stands, there are some interesting adhesion mechanisms revealed with subcellular localization to endothelial junctions, confirming earlier findings. It is unclear how the current imaging pipeline is advantageous/improves robustness of quantification as each new pair of circulating cell and underlying monolayer would require new manual annotations/training.

Major concerns

The first couple of sentences of the abstract are a bit misleading. The article, as it stands focuses quite strongly on the adhesion/arrest of circulating cells on the endothelium, not extravasation. Moreover, if the purpose in the article is to highlight the image analysis method and its utility, then perhaps re-focusing the story is needed. As it stands Fig S1 is difficult to follow and it is in the supplement. Fig 4A is perhaps the most promising novel feature of the pipeline.

It is unclear for example why AFM was performed and then the image not correlated to either fibronectin patches or ICAM1 and or CD44. This would significantly improve the novelty of the paper.

Minor concerns

If ECs are not pre-conditioned with flow prior to perfusion with circulating cells, then it is possible they are just activated by flow alone. It would be worthwhile to check ICAM expression as a ratio of total cell number in the field of view. It will also be important to rule out other adhesion mechanisms - vWF nets etc.

If fibronectin coating was performed in all the flow chambers, it is unclear why it does not show up uniformly in the images - is this expected since permeabilization was not performed?

The pipeline, as far as I can tell, would need re-training for any new cell type. It would be useful to show the robustness of the current analysis pipeline(s) for various immune cells or for mixed populations (Similar to <https://doi.org/10.1038/s41597-023-02482-8>) in time-lapse data - this would be a significant advancement and could be quite useful for the community.

Point-by-point answers to the reviewer's comments:**Reviewer's comments from EMBO Journal**

Referee #1:

The manuscript from Follain and colleagues has developed an imaging analysis pipeline based on bright field images combined with deep-learning approaches to characterize and quantify cell adhesion to endothelial cells under flow. The methodology could be very useful for all labs studying blood cells interactions in vitro, not only for cancer, but also for immunologists, vascular biologists and infections biologists. The authors use the developed pipeline to study pancreatic ductal adenocarcinoma (PDCA) adhesion to HUVEC cells seeded in a microfluidic chamber through CD44 and HA. The manuscript is complete, highly quantitative and with extensive methodological description. I support its publication with some important points for revision.

We thank the reviewer for taking the time to evaluate our work and for their constructive feedback.

1. The manuscript is intended to be a resource article for scientists using the pipeline. However, all the introduction and most of the discussion focuses on the research question that the authors are trying to address. The title on the EMBO portal "Flow dynamics and CD44-HA interaction modulate cancer cell arrest on endothelial monolayers" also does not refer to the resource, and the title in the main manuscript "Fast label-free manuscript reveals key roles..." better reflects, but would make the resource not findable for other researches not working on cancer. I would suggest that the authors rewrite these key sections of the manuscript, including the abstract to make it more appealing and exciting for other researchers not working on cancer that might be interested in this tool.

We thank the reviewer for this helpful suggestion. In the current revision, we have further emphasized the resource aspect of our work. The manuscript now:

1. Highlights the method in the title.
 2. Introduces FlowVision in the Abstract and details its capabilities.
 3. Begins the Discussion by stressing the general usefulness and limitations of the pipeline.
- We also highlight the open access to code, trained models, and datasets (GitHub/Zenodo) to

improve discoverability and facilitate reuse by researchers outside the field of cancer cell biology.

We hope these updates address the reviewer's concerns about positioning and findability.

2. Similar to the comment above, I would suggest the authors to better describe in the results the experimental pipelines used. Including track mate, pix2pic, Stardist, cell tracks collab. If the paper is intended to be a resource for biologists not so familiar with these software / packages, the technical overview of the study now in supplemental should have higher relevance, as well as its description. Right now, this is just limited to no more than 10 lines on the results.

We thank the reviewer for this helpful suggestion. We have expanded the Results to provide a clear, user-focused overview of the experimental and analysis process and to specify the software components used (StarDist, TrackMate, pix2pix artificial labeling, and CellTracksColab). The revised text reads:

“To obtain unbiased, quantitative measurements of cell landing, arrest, and migration under flow conditions, we developed FlowVision, a label-free analysis pipeline that processes brightfield time-lapse movies captured in microfluidic channels (Fig. 1F; Fig. EV1; Appendix Fig. 2A). From each movie, cells are segmented using StarDist, a deep-learning instance segmentation method ideally suited for detecting rounded cells, which we custom-trained on our annotated frames (Schmidt et al., 2018). Importantly, out-of-focus cells and very large cell clusters (>10 cells) were not detected at this stage. Segmented cells are then linked into trajectories with the Fiji plugin TrackMate (Ershov et al., 2022). The resulting tracks are then aggregated and analyzed using a modified version of CellTracksColab (Gómez-de-Mariscal et al., 2024b). This notebook-based environment enables the batch processing of millions of tracks, calculation of track metrics, event classification (such as landing frequency, arrest onset/duration, and endothelial migration), and downstream statistical analysis (Fig. EV1; see Methods for details). Importantly, all deep learning models, their training datasets, and the code used are available on a dedicated Zenodo community.”

and (for pix2pix):

“Having characterized the adhesion and movement dynamics of cancer and immune cells, we aimed to map their spatial locations within the endothelial layer with sub-cellular resolution. We enhanced FlowVision by adding an artificial-labeling module that predicts pseudo-fluorescent markers for endothelial cell junctions and nuclei directly from a brightfield video. Specifically, a pix2pix conditional generative adversarial network (Isola et al., 2018), trained on paired brightfield/fluorescence images (PECAM-1 for junctions and DAPI for

nuclei), was applied frame by frame to generate junction and nuclear probability maps. These maps were then segmented with Cellpose to identify individual endothelial nuclei and junctions. The resulting masks were registered with the circulating cell single-cell track data, allowing per-frame measurements such as the distance from a circulating cell to the nearest endothelial junction or nucleus (Fig. 4A, Fig. EV1, Appendix Fig. 4, and Video 5; see Methods for details) (Elmalm et al., 2024; von Chamier et al., 2021). This approach enabled the high-precision inference of endothelial junction and nuclear locations without the use of fluorescent markers (Appendix Fig. 4).”

3. I would specify in the results of Fig1 which of the cancer cell lines used in the study correspond to the mutation.

We thank the reviewer for this suggestion. We have now added a new figure panel (**AppendixFig. 1C**) that highlights the mutations of the cancer cell lines used in the study.

AppendixFig. 1C: Mutational burden of the PDAC lines included in this study. Source data: Cancer Cell Line Encyclopedia, queried for mutations in genes: KRAS, BRAF, CDKN2A, PIK3CA, SMAD4, ARID1A, TGFBR2, TP53, FN1, MUC16, and BRCA2.

Method section:

The Cancer Cell Line Encyclopedia (CCLE) mutation annotation format (MAF) data was downloaded from The Cancer Dependency Map Portal (DepMap). The data was queried for mutations in genes such as KRAS, BRAF, CDKN2A, PIK3CA, SMAD4, ARID1A, TGFBR2, TP53, FN1, MUC16, and BRCA2 in the cell lines involved in this study. Tumor mutational burden (TMB) was defined as the number of mutations per megabase pair (Mbp), which approximates the size of the human exome at around 30 Mbp. The data was visualized using the ComplexHeatmap R package with R version 4.3.3.

4. The authors use very nice schematic figures to describe the methodology, except for key parameters that are used throughout the manuscript. Track speed, total distance traveled and especially forward migration index. Again, this would be useful for biologists.

We thank the reviewer for this helpful suggestion. We have added clear, user-focused definitions of the key trajectory metrics at their first mention in the Results to enhance clarity. The new text reads:

“To further analyze adhesion dynamics at the single-cell level, we measured each cell's slowest, fastest, and average speeds based on frame-to-frame movement (Fig. 1J and Appendix Fig. 2B). We also calculated the total distance traveled, which reflects how far each cell moved along its path during the analysis period (Fig. 1K). Directional bias was evaluated using the forward migration index (FMI), indicating how much a trajectory aligns with or against the flow (positive = downstream, 0 = no bias, negative = upstream; Fig. 1L).”

5. I would recommend the authors to state in the text physiological capillary flow velocities, how these compare to their flow conditions as well as introduced wall shear stress values, as suggested by both reviewers. Although the difficulty of modelling simultaneously relevant WSS and flow velocities in most commercial and non-commercial microfluidic chips is a big limitation in the field, this should be acknowledged in the discussion.

We thank the reviewer for the suggestion. We have now included the wall shear stress values in the manuscript, as well as noted that we use a flat monolayer in our microfluidic system as part of the study's limitations.

“The resulting wall shear stresses applied to the circulating cells in our system at the surface of the endothelial cells range from 0.4 to 0.05 dyn/cm².”

“Limitations of the study:

We used HUVECs to model the interactions between PDAC cells and endothelium. Although HUVECs are venous in origin and may not fully capture the features of liver- or lung-derived endothelial cells, which are common metastatic sites in PDAC (diSibio and French, 2008), they are widely adopted in microfluidic platforms due to their robustness and suitability for in vitro studies. Past work demonstrates that HUVEC-based observations can be translated in vivo (Santio et al., 2024; Follain et al., 2018a; Osmani et al., 2019).

Our microfluidic system does not replicate the full complexity of in vivo vascular geometry, such as circumferential vessel walls, where cancer cells may arrest through passive occlusion. However, eliminating potential physical occlusion in our model is a necessary trade-off to focus on studying the mechanisms of active adhesion between cancer and endothelial cells. Additionally, while the flow speed values used in our study are comparable

to those found in capillaries, the shear stresses experienced by the cells are lower due to the flat geometry of our monolayer. Nonetheless, our system offers precise flow control and scalability for quantitative analysis of the effects of molecular interventions. It also enables high-resolution imaging, making it ideal for investigating the active mechanisms governing cancer cell adhesion to endothelial cells under flow.

Adapting FlowVision segmentation models to new imaging settings or cell/monolayer combinations might require a quick re-annotation and fine-tuning step (usually involving about 40–60 images and taking 6–8 hours from start to finish). We see this as a practical and manageable limitation, which is balanced by the benefits in scale and reproducibility.”

6. The pipeline is not useful to study transmigration, and this process is barely studied in the manuscript. I would recommend to remove references to it in the results and probably include it in the discussion.

We thank the reviewer for this helpful clarification. Our pipeline is indeed designed to quantify landing, arrest, and migration along the endothelium; it does not directly measure transendothelial migration (TEM). We clearly indicated this in the Results, providing a brief descriptive note that TEM events were observed for immune cells but not for cancer cells within our imaging window. We explicitly state in the Discussion that TEM is beyond the current scope of FlowVision but will be addressed in a future version.

7. I agree with reviewer 2 that the IL1- β treatment is short and could account for the lack of differences, both on PDCA and PBMEC/neutrophils. As suggested, I would suggest to repeat the experiment after a longer incubation. Alternatively, I would use TNF- α which is much widely used, and presents a stronger and faster response on the endothelium.

We thank the reviewer for this thoughtful suggestion. In response, we expanded our endothelial activation experiments to include both short (2 h) and longer (16 h) IL-1 β treatments. Prolonged exposure to 10 ng/mL IL-1 β was not well tolerated and resulted in significant intercellular gaps; therefore, for the 16 h assays, we reduced the dose to 5 ng/mL to maintain monolayer integrity.

At the molecular level, CD44 expression was unchanged in endothelial cells by IL-1 β (WB and IF, below). By contrast, endothelial adhesion molecules exhibited expected activation kinetics: ICAM-1 and VCAM-1 increased after 2 hours and continued to accumulate by 16 hours, whereas E-selectin was higher at 2 hours and declined by 16 hours (**Figure EV2D-E and below**).

Functionally, both neutrophils and PBMCs exhibited increased attachment to IL-1 β -stimulated monolayers at 2 hours and 16 hours compared to untreated controls; however, the 16-hour

condition resulted in only a modest additional increase over the 2-hour condition (**Figure EV2F-G and below**). Together, these data support the 2 h stimulation used in the primary assays as a physiologically relevant and practical time point that enhances adhesion while minimizing barrier disruption.

We appreciate the suggestion to test TNF- α . While our pipeline readily enables such comparisons, incorporating a full TNF- α dataset is beyond the scope of the current revision. We have noted this as a valuable avenue for future work aimed at systematically benchmarking cytokine-specific activation regimes.

Figure for the reviewer: CD44 expression levels in HUVECs after IL-1 β stimulation. Endothelial monolayers, either untreated or treated with IL-1 β (10 ng/ml for 2 hours and 5 ng/ml for 16 hours), were lysed, and CD44 protein levels were measured using western blots. A representative western blot is shown, along with the quantifications (6 biological repeats).

Figure EV2: Tracking and analysis of PDAC cells and immune cells' behavior under various conditions

(D-E) Endothelial monolayers, either untreated or treated with IL-1 β (10 ng/ml for 2 hours and 5 ng/ml for 16 hours), were fixed and stained to visualize E-selectin, VCAM1, ICAM1, and CD44. Stainings were performed without permeabilization to specifically label surface-accessible adhesion molecules. Images were captured using a spinning disk confocal microscope. (D) Representative fields of view are shown. Scale bar: 100 μ m. (E) Quantification of the marker per field of view is presented. Intensities were normalized to the number of nuclei per field of view, as well as the average intensity measured in the control in each repeat ($n = 45$ fields of view, 3 biological repeats).

(F-G) The number of arrested neutrophils (F) or PBMCs (G) over time, in the presence or absence of IL-1 β stimulation (10 ng/ml for 2 hours and 5 ng/ml for 16 hours). Bold lines indicate the average, and shaded areas represent the SD (4-7 biological repeats, see methods).

8. The authors should specifically state that the experiment in zebrafish is a proof of concept and that future iterations if the pipeline could include quantification in vivo.

We thank the reviewer and agree. We now explicitly state in the Results that the zebrafish experiment serves as a proof of concept rather than a quantitative in vivo analysis. The revised text reads:

“As proof of concept, we extended these observations to an in vivo context by injecting cancer cells into zebrafish embryos with labeled endothelial cell junctions (VE-Cadherin-GFP).”

We also clarify this point in the Discussion and note the planned direction:

“The current framework focuses on the arrest phase of the interaction with endothelial monolayers, and future work will expand this framework to analyze the later stages of the interaction, including extravasation in vitro and in vivo.”

9. The HA findings are interesting but confusing. The imaging seems to indicate that HA is localized on the PDCA and not on the endothelial cells. How would the staining look on an endothelial cell monolayer without PDCA? If CD44 is mediating the interaction on the PDCA side (siRNA experiments and blocking antibody are quite conclusive) how does the CD44 and HA interaction occur. How do the authors explain that the digestion on HA in EC would inhibit the interaction given the low expression in HUVEC. How can both inhibition in HA and CD44 give such strong results in PDCA. Are they cooperative?

We thank the reviewer for these helpful questions. Hyaluronan (HA) is present on the endothelial monolayer, as shown by HABP staining of endothelial cells in the absence of PDAC cells; this signal is decreased by hyaluronidase treatment, confirming its specificity (**see figure below**). Therefore, although HA is more abundant on PDAC cells, it is also present on the endothelium.

Functionally, depleting or blocking CD44 in PDAC cells significantly reduces adhesion, and the enzymatic removal of HA on either endothelial cells or PDAC cells also decreases adhesion. We therefore conclude that both perturbations impair adhesion, likely through the CD44–HA axis. We agree that the exact mechanism remains unresolved, including the relative roles of endothelial versus tumor-cell HA at the interface. This is clearly stated in the Discussion, and we note that defining this mechanism is beyond the scope of the present study; we plan to investigate it in future work.

Figure for the reviewer: Validation of hyaluronidase (HA digestion) treatment in endothelial cells. HUVEC monolayers were treated with hyaluronidase, fixed, stained, and imaged using a spinning disk confocal microscope. (A) Representative SUM projections are displayed. Note that the Hyaluronic signal in the digested sample is inside the cell. The HA signal per field of view was quantified from the SUM projections.

10. I agree with both reviewers that the authors should state the limitations of the model clearly in the discussion, including the limitations on the use of HUVEC, low WSS (as mentioned in comment 5), lack of vessel-like architecture/geometry and quite important, the lack of soft ECM. This is especially important in the context of their claim that PDCA and blood cells bind to junctional regions because of the high stiffness in these regions. Given the thinness of endothelial cells junctions (most of the time lower than 500nm), how do the authors ensure the AFM measurements are not on the substrate. In my opinion, given the scientific claims regarding on PDCA interactions with HUVEC, the substrate stiffness is an important one.

We thank the reviewer for this critical point. As noted in the Discussion, we now explicitly state the limitations of our model, including the use of HUVEC, low WSS, and the lack of vessel-like architecture/geometry.

Regarding AFM: During stiffness mapping, we observed out-of-range values (**white squares in Fig. 5D and below**) corresponding to the glass/substrate, confirming that the setup can detect the substrate. Although the presence of glass may influence absolute values at thin junctions, we consider measurements with a Young's modulus below approximately 3 kPa to reflect the cell surface properties. Importantly, these measurements align with the conditions of our perfusion assay, in which adhering cells sense local stiffness variations; we do not claim that in vivo stiffness follows the same trend.

Fig. 5D: Representative image showing AFM results acquired in contact mode. Scale bar: 10 μm.

Finally, in a separate dataset focused on extravasation, we compared endothelial monolayers on glass and 10 kPa acrylamide gels. After perfusing PDAC cells, the number of adhered cells at 15 minutes was similar. At 8 hours, softer substrates exhibited a delayed effect, characterized by increased subendothelial spreading and evidence of monolayer permeability (anti-fibronectin staining without permeabilization, indicating gaps). These observations suggest that substrate compliance mainly influences later steps, while initial arrest remains comparable under our conditions. A full analysis of ECM compliance is beyond the scope of this manuscript, but will be pursued in future work.

Confidential figure for reviewers removed

11. I agree with reviewer 2 that the quantification of VCAM, and E-selectin would improve the manuscript.

We now provide these additional quantifications in our revised manuscript (See updated Fig. 6C and D and below).

Figure 6C and D: Spatial relationship between arrested PDAC cells (MIA PaCa-2 and AsPC-1) and endothelial adhesion molecules. Labeled PDAC cells were perfused over an endothelial monolayer for 10 minutes, fixed, and stained to visualize E-selectin (E-select), VCAM1, ICAM1, ICAM2, CD44, and fibronectin (FN). Images were captured using a spinning disk confocal microscope. (C) Representative fields of view are displayed. Scale bar: 100 μ m. (D) PDAC cells were automatically segmented, and the positivity of the endothelial cells in contact with the arrested PDAC cells for each specific marker was manually scored. The percentage of PDAC cells in contact with marker-positive areas is shown (observed). The expected rate of cells in contact with each marker was calculated using Monte Carlo simulations, which took into account the cell diameter and the area of the field of view covered by each marker (see Methods for details) (n = 71-143 fields of view, three biological replicates).

12. I suggest that the anti-CD44 blocking experiment on leukocyte (PBMC, neutrophils) suggested by reviewer 1 should be done in combination and without and anti-ICAM-1 antibody, given that it is one of the major receptors for this interaction.

We thank the reviewer for this suggestion. As outlined in our revision plan, we tested anti-CD44 blockade on leukocytes (PBMCs and neutrophils) so these data could be directly compared to the anti-CD44 experiments on PDAC cells (see figure below). Under the baseline (non-IL-1 β -stimulated) endothelial condition used, leukocyte attachment is low. However, our data indicate that targeting CD44 appears to reduce neutrophil adhesion, while targeting CD44 seems to slightly increase PBMC attachment. The data are included here for the reviewer only. Since baseline adhesion is already limited, combined blocking with anti-ICAM-1 would have limited interpretability in this setting and would require obtaining a new dataset under inflammatory activation (e.g., \pm IL-1 β , multiple time points/doses, and antibody combinations across cell types). Therefore, we did not include anti-ICAM-1 in this revision, as we believe it falls outside the scope of the current manuscript.

Figure for the reviewer: The number of arrested neutrophils (Neutro.) or PBMCs over time, in the presence or absence of pre-treatment with a CD44 blocking function antibody. Bold lines indicate the average, and shaded areas represent the SD (4 biological repeats).

13. Similar to point 1 and 2, Montecarlo simulations are difficult to understand and a more detailed description would be useful.

We thank the reviewer for this point and have expanded the description to make the analysis more straightforward:

“To determine whether the frequency of arrests in CD44-positive areas exceeds the arrest frequency that could be expected to occur by chance, we used a Monte Carlo test: for each image, we kept the number of arrests fixed and randomly reassigned those points within the endothelial area thousands of times, recording how often they landed on CD44-positive regions to generate a “by-chance” baseline. The actual overlap was higher than this baseline

(Fig. 6D), suggesting a possible correlation between the presence of CD44 on the endothelial surface and the specific locations where cancer cells arrest.”

Referee #2:

Monolayers of endothelial cells (HUVEC) are exposed to cancer cells (PDAC) and leukocytes under various flow regimes. The Authors combine multiple deep-learning resources (Cellpose, StarDist, Trackmate) in order to establish a pipeline for investigating circulating cells arrest in relation to subcellular endothelial arrangements.

The results appear robust and clear, with the Authors demonstrating HA-CD44 adhesion mechanisms between PDAC cells and endothelium and both PDAC and leukocyte adhesion occurring near cell junctions.

We thank the reviewer for taking the time to carefully review our manuscript.

Label-free analysis is important in the context of immune cells and other precious patient samples - wherein labelling could alter cell behavior and/or viability. As it stands, there are some interesting adhesion mechanisms revealed with subcellular localization to endothelial junctions, confirming earlier findings. It is unclear how the current imaging pipeline is advantageous/improves robustness of quantification as each new pair of circulating cell and underlying monolayer would require new manual annotations/training.

We thank the reviewer for emphasizing the importance of label-free analysis for circulating cell-endothelium interactions. Brightfield imaging avoids disturbing cell behavior and viability, capturing rare, transient events under flow. It also allows studies on patient-derived cells, which we are actively exploring.

Regarding quantification, FlowVision enables single-cell analyses across millions of tracks with unified criteria, batch processing, and built-in QC overlays, features that are not feasible or reproducible with manual scoring.

Some annotation or training may be required when switching microscopes, adjusting magnifications, or changing cell/monolayer pairs. We added a sentence in the study limitations to acknowledge this.

In practice, this is manageable: starting from our released models and code, a new model typically takes about 6–8 hours (annotation takes 2-4 hours on 20-40 images, training takes 2-4 hours). Models can often be used directly when imaging conditions are similar, or they can be fine-tuned with a small, curated set. Finally, FlowVision is modular and compatible with various instance segmentation models; as foundation models improve, the need for retraining is expected to decrease further.

Major concerns

The first couple of sentences of the abstract are a bit misleading. The article, as it stands focuses quite strongly on the adhesion/arrest of circulating cells on the endothelium, not extravasation. Moreover, if the purpose in the article is to highlight the image analysis method and its utility, then perhaps re-focusing the story is needed.

We thank the reviewer for this helpful suggestion. We have removed “extravasation” from the abstract. In addition, in the current revision, we have further emphasized the resource aspect of our work. The manuscript now:

1. Highlights the method in the title.
2. Introduces FlowVision in the Abstract and details its capabilities.
3. Begins the Discussion by stressing the general usefulness and limitations of the pipeline. We also highlight the open access to code, trained models, and datasets (GitHub/Zenodo) to improve discoverability and facilitate reuse by researchers outside the field of cancer cell biology.

We hope that these changes address the reviewer’s concerns.

As it stands Fig S1 is difficult to follow and it is in the supplement. Fig 4A is perhaps the most promising novel feature of the pipeline.

In response to Reviewer 1, we have provided more details about the steps in the image analysis pipeline in the main text. These updates will help readers better understand the analyses that have been conducted.

It is unclear for example why AFM was performed and then the image not correlated to either fibronectin patches or ICAM1 and or CD44. This would significantly improve the novelty of the paper.

We thank the reviewer for this helpful suggestion. We performed AFM to test two simple hypotheses based on our observation of arrests at endothelial junctions: (i) whether topography (nuclear “hills” and junction “valleys”) could passively trap cells, and (ii) whether junctional stiffness is higher and could promote stronger adhesion through reinforcement. Our measurements show that the monolayer is surprisingly flat, indicating that a topographic trapping mechanism is unlikely under our conditions. Junctions are only slightly stiffer than neighboring cell bodies, suggesting stiffness may play a role, but probably isn’t the only factor causing arrest. We agree that directly correlating AFM maps with CD44, ICAM-1, or fibronectin on the same field would strengthen the study, but it is unfortunately not within our capabilities at the moment. This requires a correlative AFM-fluorescence setup with precise co-registration (shared optics/fiducials and live-compatible labeling), which we do not

currently have access to. Our AFM and immunostaining were performed on separate samples, and without specialized hardware, pixel-level alignment would be unreliable.

Minor concerns

If ECs are not pre-conditioned with flow prior to perfusion with circulating cells, then it is possible they are just activated by flow alone.

We thank the reviewer and agree that acute flow activates endothelial cells. In our study, ECs were not preconditioned with flow; instead, we used a standardized protocol to ensure consistent flow-induced activation across all conditions. Specifically, all assays were performed with the same flow onset and identical culture timelines, as well as parallel controls, ensuring that comparisons (e.g., treatments, cell types) occur within the same activation context.

It would be worthwhile to check ICAM expression as a ratio of total cell number in the field of view.

We thank the reviewer for this excellent suggestion. We now report the expression of CD44, ICAM-1, VCAM-1, and E-selectin in control and IL-1 β -stimulated endothelium (2 h and 16 h), as a ratio of the total number of cells per field of view. These results are presented in **Figure EV2 D-E** (and below for convenience).

Figure EV2D-E: Tracking and analysis of PDAC cells and immune cells' behavior under various conditions

(D-E) Endothelial monolayers, either untreated or treated with IL-1 β (10 ng/ml for 2 hours and 5 ng/ml for 16 hours), were fixed and stained to visualize E-selectin, VCAM1, ICAM1, and CD44. Stainings were performed without permeabilization to specifically label surface-accessible adhesion molecules. Images were captured using a spinning disk confocal microscope. (D) Representative fields of view are shown. Scale bar: 100 μ m. (E) Quantification of the marker per field of view is presented. Intensities were normalized to the number of nuclei per field of view, as well as the average intensity measured in the control in each repeat (n = 45 field of view, 3 biological repeats).

It will also be important to rule out other adhesion mechanisms - vWF nets etc.

We have now broadened our analysis of adhesion molecules to include VCAM-1 and E-selectin, as recommended by other reviewers (see Figure 6 below for reference). All our data consistently show that CD44 is a key adhesion molecule involved in mediating the attachment of PDAC cells to endothelial monolayers. However, we also acknowledge that other adhesion molecules, like integrins, are likely to play a role in this process.

Figure 6C and D: Spatial relationship between arrested PDAC cells (MIA PaCa-2 and AsPC-1) and endothelial adhesion molecules. Labeled PDAC cells were perfused over an endothelial monolayer for 10 minutes, fixed, and stained to visualize E-selectin (E-select), VCAM1, ICAM1, ICAM2, CD44, and fibronectin (FN). Images were captured using a spinning disk confocal microscope. (C) Representative fields of view are displayed. Scale bar:

100 μm . (D) PDAC cells were automatically segmented, and the positivity of the endothelial cells in contact with the arrested PDAC cells for each specific marker was manually scored. The percentage of PDAC cells in contact with marker-positive areas is shown (observed). The expected rate of cells in contact with each marker was calculated using Monte Carlo simulations, which took into account the cell diameter and the area of the field of view covered by each marker (see Methods for details) ($n = 71\text{-}143$ fields of view, three biological replicates).

If fibronectin coating was performed in all the flow chambers, it is unclear why it does not show up uniformly in the images - is this expected since permeabilization was not performed?

We thank the reviewer for this clarification. All flow chambers were coated uniformly with fibronectin. Additionally, endothelial cells produce and assemble their own fibronectin network. The observed non-uniform fibronectin signal in our images is expected because staining was performed without permeabilization: the antibody labels only surface-accessible fibronectin, which could include apical patches that circulating cells could engage with and matrix exposed through junctional gaps. We also use this as a readout of monolayer leakiness. This intentional approach follows our previously published strategy (Ball et al., 2024, JCB). We have clarified this rationale in the Results section and the figure legend.

From Ball et al. 2024: Figure S3B: HUVECs were allowed to form a monolayer. Cells were then fixed and stained for DAPI, F-actin, and fibronectin (with or without permeabilization) before being imaged on a spinning disk confocal microscope. Representative maximum intensity projections are displayed. Scale bar: 250 μm .

The pipeline, as far as I can tell, would need re-training for any new cell type. It would be useful to show the robustness of the current analysis pipeline(s) for various immune cells or for mixed populations (Similar to <https://doi.org/10.1038/s41597-023-02482-8>) in time-lapse data - this would be a significant advancement and could be quite useful for the community.

We thank the reviewer for this suggestion. As previously discussed, some annotation or training may be necessary when changing microscopes, magnifications, or cell/monolayer pairs. We have added a sentence in the study limitations to acknowledge this. In practice, this process is manageable: starting with our released models and code, creating a new model usually takes about 6–8 hours (annotation requires 2-4 hours on 20-40 images, and training takes 2-4 hours). Models can often be used directly if imaging conditions are similar, or they can be fine-tuned with a small, curated set. Lastly, FlowVision is modular and compatible with various instance segmentation models; as foundation models improve, the need for retraining is expected to decrease further.

We thank the reviewer and agree that analyzing mixed populations would be very valuable. A thorough study would need (i) additional dedicated datasets with specified cell compositions and conditions, and (ii) new analysis workflows capable of combined semantic and instance segmentation and co-tracking to identify and track multiple cell types within the same video. Developing and validating these methods is a significant effort that would warrant a separate manuscript. We are actively pursuing this direction; however, a comprehensive benchmark across various immune subsets and mixed samples is beyond the scope of this current revision.

Reviewer's comments transferred from another journal

Reviewer #1 (Remarks to the Author):

The mechanisms of transendothelial migration of cancer cells under physiological flow conditions as a key step of metastatic organ colonization remain incompletely defined. In addition, albeit general agreement assumes that cancer cells and leukocytes utilize similar steps and mechanisms of extravasation, conclusive experimental evidence supporting such homology is lacking.

Follain and coworkers here used a perfused HUVEC monolayer system with label-free microscopy and AI-based tracking to reliably quantify cell arrest, migration along and, in part, transmigration through the endothelial wall in various cancer cell (PDAC) and leukocyte models (PBMCs, neutrophils). Using a range of flow rates, PDAC models diverge in adhesion strength but consistently engage endothelial CD44 via cell-surface hyaluronan on the cancer cell to arrest at endothelial junctions, while transmigration was not observed. Interfering with CD44/HA interaction significantly reduced adhesion. Similar arrest patterns were obtained for leukocytes, followed by effective transmigration.

We thank the reviewer for taking the time to read and provide feedback on our work. We are delighted to learn that the focus and quality of our work are appreciated.

The imaging pipeline developed by the authors provides a useful tool for acquiring real-time insights into cancer cell-endothelial cell interactions, however the approach is within the scope of already existing tracking approaches and, hence, does not break new ground. The descriptive results are potentially interesting on mostly similarities between cancer cells and leukocytes. The molecular findings on HA/CD44 interaction to mediate cancer cell arrest indicate that this approach may be suitable to identify receptor-ligand cascades mediating adhesion in the vascular bed.

We thank the reviewer for their insightful comments and appreciate the chance to clarify the innovative features of our analysis pipeline. Although our approach combines existing tools, it does so in a novel and comprehensive way that makes significant new advances in understanding how circulating cells interact under flow.

Our automated imaging framework is completely label-free, enabling high-speed (25 Hz) and long-term imaging under physiologically relevant flow conditions without the phototoxicity or disturbance associated with fluorescence-based methods. This capability allowed us to capture dynamic arrest behaviors in real-time, a feat not previously possible in earlier studies, including our own, which depended on fixed endpoint imaging.

We systematically analyzed thousands of individual cell trajectories across hundreds of movies by combining deep learning-based segmentation (StarDist) with robust tracking (TrackMate). This method enabled quantitative assessment of adhesion dynamics and post-arrest migration at single-cell resolution and on a large scale. Furthermore, this high-throughput analysis revealed spatial organization in cell arrest, including the unexpected discovery of clustering behavior on the endothelial monolayer.

Another notable novel component is the use of artificial labeling through pix2pix deep learning models to identify endothelial landmarks, such as nuclei and junctions, from brightfield images. This method allowed us to pinpoint cell landing and arrest with subcellular accuracy, without relying on fluorescent staining. To our knowledge, this is one of the few cases where artificial labeling has been successfully utilized to study a biological phenomenon, rather than solely for method development.

Our pipeline is modular, constructed with well-maintained, widely available open-source tools. This is an intentional design decision. By avoiding highly customized solutions, we make it easier for researchers studying related processes, such as immune cell trafficking, stem cell homing, or CTC behavior in different vascular contexts, to adapt and extend our workflow to suit their needs. We believe this flexibility and accessibility will greatly enhance the impact and adoption of our approach within the broader cell biology and vascular biology communities.

Lastly, we highlight that all code, trained models, and tracking datasets are openly shared via GitHub and Zenodo, ensuring full reproducibility and wide dissemination.

In summary, we respectfully disagree with the assessment that our pipeline lacks innovation. By incorporating label-free imaging, deep learning-based segmentation, artificial labeling, and high-throughput tracking into a reproducible, modular framework, our approach offers new capabilities for exploring the mechanisms of cell arrest and adhesion under flow.

As main shortcomings, the data fall short to deliver truly novel insights and lack validation in vivo. The in vitro model lacks the geometries of vessel lumens with circumferential walls which enable embolic trapping of tumor cells to a greater extent than of leukocytes, and little evidence is provided that the spatial heterogeneity and intravascular arrest mechanisms hold up on a mammalian metastasis model with a mature vascular system. The comparative data between PDAC cells and leukocytes indicate similarities as well as disparate behaviors, however no compelling information on overlapping and divergent adhesion mechanisms are provided. Lastly, the role of CD44 in tumor cell clustering and metastasis was recognized before, hence the conceptual advance on addressing CD44/HA linkage here requires maturation to resolve potentially novel mechanisms of heterogeneity of seeding behaviors and their relevance for metastasis in vivo.

We appreciate the reviewer's constructive feedback and welcome the opportunity to clarify the primary focus and scope of our study. We agree that in vivo validation is crucial for translating mechanistic insights into physiological and clinical applications. However, the primary objective of our work is to provide a methodologically sound and user-friendly platform for investigating the active adhesion mechanisms employed by circulating cells, particularly cancer cells, as they interact with the vascular endothelium under flow conditions.

In vivo, cancer cells can become arrested in blood vessels through mechanical occlusion and active adhesion, involving specific molecular interactions with the endothelium. Our study aims to dissect this latter process. To achieve this, we utilized a simplified yet idealized in vitro system: microfluidic channels lined with endothelial monolayers that deliberately exclude geometric constraints, such as circumferential vessel walls. This setup allows us to isolate and examine adhesion-mediated arrest events without the confounding effects of passive trapping.

Although the model lacks the full architectural complexity of intact vasculature, it replicates key biomechanical features, including flow-driven cell-endothelium interactions. Microfluidic platforms like ours are widely used in vascular biology and immunology to study rolling, firm adhesion, and transmigration events. They are broadly recognized as effective tools for mechanistic research.

Regarding the biological findings, while CD44 has been previously linked to metastasis, we reveal new aspects of its role in the dynamic arrest of pancreatic cancer cells. Our data show that PDAC cells preferentially arrest at CD44-high endothelial sites through a hyaluronan-dependent mechanism, and that this interaction promotes the spatial clustering of

PDAC cells on the endothelium under flow conditions. This clustering is further influenced by local flow disturbances caused by previously adhered cells, a novel observation that connects molecular adhesion with spatial self-organization.

Furthermore, although we do not claim to fully resolve all overlapping and divergent adhesion mechanisms between cancer cells and leukocytes, our comparative analysis offers a valuable foundation by providing side-by-side dynamic data on their arrest kinetics, spatial behavior, and response to inflammatory stimuli.

In summary, although *in vivo* studies will be essential to test the applicability of these findings in complex tissue environments, our work bridges a significant gap by providing a scalable, high-resolution platform for studying active adhesion mechanisms. This platform complements *in vivo* models and opens new pathways for future research on circulating cell vascular interactions.

1. The flat system configuration does not replicate the tube-like concentric geometry of blood vessels. The geometry of the used IBIDI chambers (dimensions in 3D) should be detailed. Moreover, *in vivo* data suggest that active adhesion of tumor cells to endothelial cells may not be as significant as proposed, given that tumor cells typically undergo embolic arrest in capillaries rather than relying on active adhesion (doi.org/10.1038/nm.2072; [doi: 10.1007/978-3-319-95294-9_11](https://doi.org/10.1007/978-3-319-95294-9_11)). For embolic arrest, active adhesion may even be dispensable. Thus, the physiological relevance of the proposed mechanism of intravascular arrest remains to be demonstrated (see also point 5).

We thank the reviewer for raising these critical points and welcome the opportunity to clarify our reasoning and interpretation. We fully acknowledge that our microfluidic model uses a simplified geometry that does not replicate the complete 3D cylindrical structure of native blood vessels. The IBIDI μ -Slide VI channels used in our experiments have a rectangular cross-section measuring 3.8 mm in width and 0.4 mm in height. Although simplified, such flat perfusion systems are widely accepted in the field and have proven highly valuable for studying dynamic cell-endothelial interactions under controlled flow conditions.

We recognize the interplay between vascular architecture, flow forces, and adhesion that govern intravascular arrest. However, we respectfully disagree with the notion that embolic arrest is the primary or sole mechanism responsible for arresting cancer cells *in vivo*. Most *in vivo* studies focus on the extravasation phase, which typically occurs hours after the initial arrest. By this time, the local vasculature has usually remodeled in response to the presence of tumor cells. This vascular response can cause local vessel constriction and create an apparent embolic profile, which may overstate the importance of mechanical blockage while underestimating the role of active adhesion mechanisms.

Numerous studies, including our own, have demonstrated that cancer cells can become lodged in vessels that are not blocked by geometry. For example, in Follain et al. (*Dev Cell* 2018), we utilized intravital microscopy and nanometric flow tracers to demonstrate that blood

flow persists both upstream and downstream of arrested tumor cells in capillary beds, indicating that occlusion does not occur (**see the figure below**). Additionally, we found that non-adhesive beads of similar size arrest at a much lower rate, approximately four times less than cancer cells, suggesting that passive size restriction alone cannot fully explain arrest behavior. These findings strongly emphasize the importance of active adhesion in mediating early arrest.

Furthermore, the significance of adhesive interactions between endothelial and tumor cells for effective vascular arrest and subsequent extravasation has been confirmed by multiple studies (Gassmann et al., IJCD 2009; Reymond et al., JCB 2012; Au et al., PNAS 2016; Offeddu et al., Comm. Biol. 2021). Using various tumor models, these studies have demonstrated that blocking specific adhesion molecules, including CD44, VCAM-1, ICAM-1, and integrins, significantly reduces cancer cell retention and transendothelial migration. Our ongoing research builds on this work by analyzing these interactions with high temporal and spatial resolution under controlled flow conditions.

While we recognize that our model does not encompass all anatomical features of the vasculature, it is well-suited for isolating and analyzing active adhesion processes without interference from physical trapping. Therefore, our findings provide valuable mechanistic insights into early adhesion events that are difficult to observe in vivo and are likely to complement rather than contradict the observations from capillary-rich tissues.

Overall, we believe that reduced flow and constricted vasculature similarly affect arrest and stable adhesion: both slow down circulating cancer cells and increase the contact time between these cells and the endothelium. This provides more opportunity for active adhesion to occur, allowing the cells to engage their receptors and attach stably. Decoupling flow from physical constraints is nearly impossible in vivo. That's why studying arrest in a non-constricting "vessel" requires the use of simple 2D systems.

We hope this clarifies our position and the biological significance of our model and findings. We agree that adding in vivo validation will be an important future step. However, we believe our current work offers valuable mechanistic insights into adhesive arrest that are directly relevant to metastatic progression.

In the revised manuscript, we clarified the main aims of our study throughout the text and added a paragraph describing the study's limitations, which clearly highlights these restrictions.

Extracted panels from Follain et al. Dev. Cell 2018, figure 5 and figS 2. (L) Experimental workflow and idealized representation (left). Embryos are injected with CTCs and 100-nm beads, and then imaged at high speed, before undergoing single-particle tracking analysis (middle). Corresponding quantification of tracks' straightness and velocity in three scenarios (open, partially closed, and closed) and in silico representation of laminar flow around an arrested tumor cell (right). Values are mean \pm SD. * $p < 0.05$. ns, not significant. Extracted panels from Follain et al. Dev. Cell 2018, supplementary figure S2. (A) Experimental setup, representative image, and quantifications of the 10- μ m bead injection. [live recordings post-injection] Values are mean \pm SD. * $p < 0.05$. ns, not significant.

2. It is unclear how the experimental setup was constructed. Was shear flow applied to the endothelial cells before the introduction of cancer cells, or was it applied only briefly before? The strength of fluid flow influences molecular networks of HUVEC monolayers, thus the metrics used should be quantitative and reproducible. The actual shear forces should be revealed for the "high" and "low" flow speeds. How do these levels to arterial and venous in vivo vascular beds? Thus, the used conditions should be linked to realistic scenarios cancer cells would be confronted with in vivo.

We thank the reviewer for this valuable comment. We fully agree that flow dynamics significantly influence endothelial cell architecture and signaling, and we have ensured that our conditions are standardized and reported clearly and reproducibly.

In our experimental design, HUVEC monolayers were cultured under static conditions for 4 days until visual confluency was reached, and maintained for an additional 24 hours to ensure junctional maturation. The channels were then connected to the microfluidic perfusion system just 5 minutes before the introduction of circulating cells. This brief pre-flow period was sufficient to fill and equilibrate the channels without substantially altering the monolayer's

organization. We were particularly mindful that prolonged exposure to flow can reprogram endothelial cell morphology and gene expression, as demonstrated in our previous work (Follain et al., Sci Rep 2021). To maintain consistency across experiments, we ensured that all perfusions adhered to an identical protocol and duration.

Our system was calibrated to produce flow velocities ranging from 100 to 400 $\mu\text{m}/\text{sec}$, which are physiologically relevant and fall within the range observed in human capillaries and post-capillary venules (Follain et al., Nat Rev Cancer 2020). Importantly, due to the relatively large height of our microfluidic channels (400 μm), these flow speeds generate modest wall shear stresses, ranging from approximately 0.4 to 0.05 dyn/cm^2 at the endothelial surface. These values are lower than the typical shear stress in capillaries but are comparable to those in other microfluidic models of vascular adhesion (e.g., Cerutti et al., Cell Rep, 2024). They are already sufficient to limit the adhesion of circulating cells in our 2D configuration, as shown in Figure 1 of the manuscript.

Additionally, we deliberately avoided high arterial shear stress conditions (e.g., $>10 \text{ dyn}/\text{cm}^2$), which are known to induce endothelial alignment and morphological polarization in the direction of flow. Since we focus on dissecting the adhesive behavior of circulating cells under capillary-like conditions, we aimed to model flow environments that are most physiologically relevant for cell arrest.

In conclusion, we carefully chose our flow conditions to reflect the permissive vascular environments encountered during early metastatic dissemination, particularly in capillary and venular beds where extravasation typically occurs. All flow rates and shear values were standardized across experiments to ensure reproducibility and comparability of results.

In the revised manuscript, we clearly describe our technical system as well as its limitations.

3. The vessel beds used for extravasation *in vivo* differ for tumor cells (arterioles, capillaries) and leukocytes (postcapillary venules), and barely overlap. HUVECs cultured for several days are selected as a generic but immature endothelium. Likewise, the staining in zebrafish embryos represents developing, immature vessels. Thus, it is unclear to which extent CD44 expression and heterogeneity thereof represents an *in vivo*-relevant phenomenon in mature mammalian vessels. Quiescent endothelial cells in arteries, veins, and organ capillaries might exhibit varying expression of tight junctions and differ in glycocalyx composition and structure compared to HUVEC. For instance, venous endothelial cells are less permeable and more resistant to extravasation than the activated endothelium often modeled by HUVEC. Thus, the key findings need to be verified in mature vascular models with *in vivo*-like geometries. In addition, depending on the predicted site of extravasation, in line with the used shear forces,

organ-specific or microvascular endothelial cells that more closely replicate the specialized characteristics of the endothelium should be included.

We fully agree that the endothelial phenotype varies significantly across the vascular tree and among different tissue beds, and that these distinctions play an essential role in regulating the extravasation of both tumor and immune cells in vivo.

That said, the focus of the present study is not to recapitulate the full physiological diversity of the mammalian vasculature, but rather to introduce a novel, high-resolution, label-free live imaging and analysis platform that allows systematic quantification of adhesion dynamics under controlled flow conditions.

While HUVECs do not fully replicate the structure or function of mature arterial, venous, or tissue-specific microvascular endothelium, they remain some of the most commonly used and widely accepted models for initial studies of cell-endothelium interactions. Their use in our study allows for direct comparison with a substantial body of published literature. It provides a standardized background against which perturbations, such as cytokine stimulation or receptor blocking, can be systematically evaluated. Additionally, the reproducibility and accessibility of HUVEC cultures make them an ideal platform for a methodological framework designed to be shared and expanded by the broader community. Regarding the zebrafish data, we acknowledge that the vasculature in embryos represents a developing network. However, we chose this model because it offers exceptional in vivo optical accessibility and is well-established for the real-time analysis of circulating tumor cell behavior under flow. We fully agree with the reviewer that future research should incorporate tissue-specific or organotypic endothelial models and utilize advanced 3D microfluidic systems to better mimic in vivo geometries and biomechanical constraints. One of the long-term goals of our study is to develop a modular pipeline that can be implemented across such advanced systems to investigate how vascular specialization influences cell behavior.

In summary, while our current model is intentionally simplified, it effectively reveals core mechanistic principles of cell adhesion under flow. We see our work as a foundational and essential step toward integrating this methodology into more physiologically complex systems, and we hope it will serve as a valuable resource for researchers studying vascular heterogeneity and tissue-specific metastasis in the future. As mentioned earlier, in the revised manuscript, we clarify the aims, objectives, and limitations of our in vitro models more explicitly.

4. The authors show (in Videos 1 and 2) that Mia PaCa-2 arrest on endothelial cells to a greater extent than AsPC-1. They further show that these two PDAC models express similar levels of CD44 and that CD44/HA silencing/blocking/depletion equally impairs the ability of cancer cells to attach to endothelial cells. Given these findings, they claim that CD44 is the sole mediator of arrest on the endothelium, which may ignore the difference between these cells.

We thank the reviewer for this comment and for raising important points. There appears to be a misinterpretation of our data, and we apologize for any confusion. As shown in **AppendixFig. 6A** of the current manuscript, the expression level of CD44 differs between the two cell lines (see below for convenience). It is significantly lower in AsPC-1 compared to MIA PaCa-2, which directly correlates with their differences in adhesion capability.

AppendixFig. 6A: CD44 expression levels in AsPC-1, MIA PaCa-2 and Panc 10.05

Observations from the videos further suggest that Mia PaCa-2 cells tend to form clusters and flow in clusters, and that cluster kinetics differ from single cells. The authors should further explore clusters versus single-cell arrest to better clarify if the differences between the PDAC models are linked to other factors beyond CD44/HA.

We thank the reviewer for bringing this point to our attention. MIA PaCa-2 cells tend to aggregate within the tubing and can enter the chamber as clusters. We agree that this behavior may affect their arrest dynamics and warrants further investigation. In our study, we excluded large aggregates (tens to hundreds of cells) from analysis because they do not form adhesive contacts with the endothelium and instead displace previously arrested cells due to bulk or inertia, an effect we repeatedly observed during live imaging. To maintain dataset quality, our deep learning-based segmentation was trained to omit these large aggregates.

Smaller clusters (2–8 cells) are more frequently observed engaging the endothelium and are capable of successfully arresting. In the revised manuscript, we quantify arrests as single cells versus small clusters (2–8 cells). Even for MIA PaCa-2, over 85% of arrests involve single cells; for comparison, approximately 96% of AsPC-1 arrests are single-cell events. Among immune cells, around 10% of neutrophil arrests and about 4% of monocyte arrests involve small clusters. These findings are now described in the text, with the related analysis shown in **Figures 1I and 2E**.

Figure 1I. Percentage of AsPC-1 and MIA PaCa-2 cells arresting as single cells per movie. Here, a cluster is defined as at least two cells that arrest together (at the same time) within a cell diameter of each other (see Methods for details). Results from the various flow speeds are pooled ($n > 24$ videos, 6-7 biological repeats; see Methods).

Figure 2E: Percentage of neutrophils and PBMCs cells arresting as single cells per movie in the presence or absence of IL-1 β stimulation (10 ng/ml for 2 hours). Here, a cluster is defined as at least two cells that arrest together (at the same time) within a cell diameter of each other (see Methods for details). Results from the various flow speeds are pooled ($n > 16$ videos, 4-8 biological repeats; see Methods).

5. The in vivo relevance and translatability of the key findings in a mammalian model remain unclear. The study observes a high degree of variability in endothelial cell adhesion between the seven PDAC cell lines, with some lines, such as AsPC-1 and MIA PaCa-2, exhibiting high adhesion to endothelial cells, while others, like SW1990 and PANC1, show reduced adhesion under flow conditions. However, it is unclear how these differences in adhesion are determined by CD44, other adhesion molecules downstream of CD44, and the cell clustering

state in vitro. Likewise, the relevance of these findings for the metastatic potential of these cell lines in vivo remains unclear. In addition, in vivo, only a very small fraction of cancer cells but most if not all leukocytes extravasate. Thus, the model may inaccurately overrepresent adhesion (and transmigration?). The results using zebrafish embryos neither test seeding in a mature vascular system, nor provide molecular data to confirm the importance of CD44/HA interactions. At 48 hours post-fertilization, zebrafish embryos possess rather immature blood vessels, thus the relevance of this model is unclear. In addition, neither the relevance of this adhesion mechanism for transmigration and metastasis formation was explored. Thus, key results including interventions need to be shown for adhesion, transmigration, and outgrowth in a mammalian system.

We thank the reviewer for these critical points. As detailed in our responses above, the goal of this study is to examine the adhesion mechanisms used by cancer cells on endothelial monolayers by developing a label-free imaging and analysis platform to investigate the dynamics of cancer and immune cell adhesion under flow conditions. While we recognize that in vivo validation in a mammalian model is essential to fully understand the significance of CD44/HA interactions in metastasis and extravasation, this is beyond the scope of the current study. Our goal is to provide the community with a high-resolution, reproducible method for investigating early adhesive events. The platform is highly adaptable and will be valuable for future research on molecular specificity, cell line heterogeneity, and in vivo relevance in more complex and organotypic models.

As mentioned earlier, in the revised manuscript, we provide a more precise explanation of the aims, objectives, and limitations of our in vitro models.

6. Despite their claims in the introduction on similarities and differences of endothelial cell binding by cancer cells and leukocytes, the experiments do not explore shared and divergent mechanisms between these principal cell types, both at cellular and molecular level.

We respectfully disagree with the reviewer's assessment. Our manuscript includes direct comparisons between cancer cells and immune cells at both the cellular and behavioral levels, as shown in **Figures 2, 3, 4, and 6**. These analyses reveal significant similarities in arrest dynamics, such as preferential junctional attachment, as well as key differences, particularly in responsiveness to inflammatory cues and the ability to transmigrate.

In response to Reviewers 1 and 2, we have conducted experiments blocking CD44 in primary neutrophils and PBMCs to assess CD44's role in immune cell adhesion to endothelial cells. We found that blocking CD44 reduces neutrophil adhesion under non-stimulated conditions, while it slightly increases the attachment of PBMCs. The data are included here solely for the reviewer (**see below**).

Figure for the reviewer: The number of arrested neutrophils (Neutro.) or PBMCs over time, in the presence or absence of pre-treatment with a CD44 blocking function antibody. Bold lines indicate the average, and shaded areas represent the SD (4 biological repeats).

Minor points:

7. The limitations of the chosen system design should be discussed in detail. Which context can be drawn to in vivo reality of cancer cell embolic arrest? Which additional work will be required to link this approach to in-vivo relevance?

We thank the reviewer for this comment. We added a limitations paragraph to the manuscript to address those points and clarify our text. The importance of emboli in vivo has been discussed previously.

The limitations paragraph reads as follows:

“We used HUVECs to model the interactions between PDAC cells and endothelium. Although HUVECs are venous in origin and may not fully capture the features of liver- or lung-derived endothelial cells, which are common metastatic sites in PDAC (diSibio and French, 2008), they are widely adopted in microfluidic platforms due to their robustness and suitability for in vitro studies. Past work demonstrates that HUVEC-based observations can be translated in vivo (Santio et al., 2024; Follain et al., 2018a; Osmani et al., 2019).

Our microfluidic system does not replicate the full complexity of in vivo vascular geometry, such as circumferential vessel walls, where cancer cells may arrest through passive occlusion. However, eliminating potential physical occlusion in our model is a necessary trade-off to focus on studying the mechanisms of active adhesion between cancer and endothelial cells. Additionally, while the flow speed values used in our study are comparable to those found in capillaries, the shear stresses experienced by the cells are lower due to the flat geometry of our monolayer. Nonetheless, our system offers precise flow control and

scalability for quantitative analysis of the effects of molecular interventions. It also enables high-resolution imaging, making it ideal for investigating the active mechanisms governing cancer cell adhesion to endothelial cells under flow.

Adapting FlowVision segmentation models to new imaging settings or cell/monolayer combinations might require a quick re-annotation and fine-tuning step (usually involving about 40–60 images and taking 6–8 hours from start to finish). We see this as a practical and manageable limitation, which is balanced by the benefits in scale and reproducibility.“

8. In page 7 authors write “Using Ripley’s L function and Monte Carlo simulations, we revealed a pronounced tendency for arrested cells to cluster especially at higher flow speed (Fig. 6B and Fig. S6A-B). Notably, PDAC cells showed more clustering than neutrophils. AsPC-1 cells clustered the most, and the MIA PaCa-2 cells clustered similarly to PBMCs (Fig. 6B and Fig. S6B). This finding suggests a non-random pattern of arrest on endothelial monolayers, with cells preferentially arresting at specific hotspots.”. However, this claim does not seem to align with the observations in the videos (see e.g., video 1). The authors should clarify the discrepancy between their claim from simulated data and the visual data presented.

We thank the reviewer for pointing out this potential confusion. To clarify, the data shown in **Figure 6B and in this version of the manuscript Figures EV3A and EV3B** do not originate from simulated datasets; instead, they are based on quantitative spatial analyses conducted directly on the experimental video data. We used Ripley’s L function and Monte Carlo spatial point pattern analysis to evaluate whether arrested cells tend to cluster more than expected by chance. Importantly, this analysis does not determine the number of clusters; it assesses whether the spatial arrangement of cells differs significantly from randomness. Our data reveal a non-random arrest pattern or “hotspot” behavior across the monolayer.

9. In page 10 the authors write “Interestingly, CD44 is known to form homophilic interactions (Liu et al., 2019), and MIA PaCa-2 and AsPC-1 cells exhibited high CD44 surface expression (Fig. 6C).” but Fig. 6C displays immunofluorescence staining of PDAC cells on the endothelium. The correct figure panel to cite may be Figure S8C or Figure 8A.

We appreciate the reviewer for pointing this out and apologize for the lack of clarity. In **Figure 6C**, we used immunofluorescence to demonstrate that both MIA PaCa-2 and AsPC-1 cells exhibit strong CD44 staining during their interaction with the endothelium, with signal intensity significantly higher than that observed on the endothelial cells themselves. To support and quantify this finding, we included the corresponding western blot (**AppendixFig. 6A; EV4A**), which confirms higher CD44 protein levels in both PDAC cell lines. We have revised the text to improve cross-referencing between figures and to clarify the difference between imaging and protein quantification.

10. The number of tracks, cells, fields of view, and biological replicates is clearly stated in the figure legends, ensuring transparency. However, the substantial variation in these numbers across experiments—ranging from 30 to over 15,000 for analyzed cells or tracks or fields of view, and from 2 to 12 biological replicates—raises concerns about the homogeneity of the experimental design. The authors should further clarify the rationale for these variations.

We thank the reviewer for this important observation. We acknowledge that the number of tracks, cells, and biological replicates varies across experiments and agree that this requires clarification. Specifically, for the condition “MIA PaCa-2 + IL-1 β ,” we conducted two biological replicates, as described in the Methods section (“Cell tracking and quantitative analysis of tracking data”). Due to the strong adhesive behavior of MIA PaCa-2 cells, these two replicates produced a very high number of tracks ($n = 7045$), which we considered sufficient to obtain statistically meaningful and representative results. Regarding **Figure 7L**, which displays only 35 tracks for the CD44 blocking condition, this reflects the expected biological outcome: CD44 inhibition significantly reduces cell adhesion, as shown in **Figures 7D and 7E**. After applying our filtering criteria to ensure that only stable adhesion events are analyzed, very few tracks remain. This accurately represents the biology of the system rather than indicating an inconsistency in the experimental design. Overall, the variation in track counts across conditions reflects the biological diversity of adhesion among different cell types and treatments, ranging from highly adherent PDAC lines to weakly interacting or inhibited cells. Our statistical analysis approach addresses these differences by employing robust non-parametric randomization tests and reporting effect sizes using Cohen’s d , thereby enabling reliable comparisons even with varying sample sizes. These methods are especially suitable for datasets with many observations (as in most of our analyses) and ensure that our conclusions are supported by rigorous quantitative analysis.

Reviewer #2 (Remarks to the Author):

In this methods manuscript, the authors describe an automated pipeline to analyze attachment of cancer cells and leukocytes to endothelial cells under flow conditions in vitro, without using fluorescent dyes to label cells. Particularly useful is their ability to outline endothelial cell junctions and nuclei without fixing and labeling, and to determine where cells attach with respect to endothelial expression of a selection of three adhesion molecules. The methodology advances the field and will be useful for those using similar approaches to investigate how leukocytes/cancer cells attach to endothelial cells (provided it is easy to access for non-machine learning experts), although not so far for those studying these processes in vivo.

Overall, the manuscript is well written although the statistical analysis is difficult to understand for a non-expert in the way it is presented. Some ANOVA results would help here.

We thank the reviewer for their positive feedback. We are delighted to learn that our efforts to make deep learning approaches accessible to biologists are well-received.

Limitations of method: The authors need to describe the limitations of their approach, and that it has not been tested under physiological conditions or in vivo. First, endothelial cells are grown and analyzed on a hard inflexible surface of glass/plastic, very unlike the softer more flexible environment of capillaries and small blood vessels in vivo, where cancer cells arrest. Second, it is also a large tube that is far wider than a capillary in vivo. Third, the authors use HUVECs, which are not microvascular endothelial cells and have different properties. Fourth, they are not quantifying extravasation of cancer cells but just adhesion. Indeed, in some cases, cancer cells are initially stay in blood vessels and proliferate there, rather than extravasate e.g. Almehdi et al., Nature Med 2000. The authors should consider this in their Introduction/Discussion sections.

We thank the reviewer for this thoughtful and constructive suggestion. We fully agree that acknowledging the limitations of our in vitro system and its relevance to physiological conditions is crucial. In response, we have added a dedicated “Limitations of the Study” paragraph to the Discussion section of the revised manuscript.

The limitations paragraph reads as follows:

“We used HUVECs to model the interactions between PDAC cells and endothelium. Although HUVECs are venous in origin and may not fully capture the features of liver- or lung-derived endothelial cells, which are common metastatic sites in PDAC (diSibio and French, 2008), they are widely adopted in microfluidic platforms due to their robustness and suitability for in vitro studies. Past work demonstrates that HUVEC-based observations can be translated in vivo (Santio et al., 2024; Follain et al., 2018a; Osmani et al., 2019).

Our microfluidic system does not replicate the full complexity of in vivo vascular geometry, such as circumferential vessel walls, where cancer cells may arrest through passive occlusion. However, eliminating potential physical occlusion in our model is a necessary trade-off to focus on studying the mechanisms of active adhesion between cancer and endothelial cells. Additionally, while the flow speed values used in our study are comparable to those found in capillaries, the shear stresses experienced by the cells are lower due to the flat geometry of our monolayer. Nonetheless, our system offers precise flow control and scalability for quantitative analysis of the effects of molecular interventions. It also enables high-resolution imaging, making it ideal for investigating the active mechanisms governing cancer cell adhesion to endothelial cells under flow.

Adapting FlowVision segmentation models to new imaging settings or cell/monolayer combinations might require a quick re-annotation and fine-tuning step (usually involving about

40–60 images and taking 6–8 hours from start to finish). We see this as a practical and manageable limitation, which is balanced by the benefits in scale and reproducibility.”

Main points for revision:

1. CD44 and cancer cell attachment: this is very well characterized from previous studies, which should be referenced in the Introduction i.e. that this is already known.

We thank the reviewer for this helpful suggestion. We agree that previous studies, particularly in breast cancer models, have established a role for CD44 in mediating the adhesion of cancer cells to the endothelium. We have revised our manuscript to ensure we acknowledge and properly reference this foundational work.

Our study, however, offers novel mechanistic insights within the context of pancreatic cancer, a biologically and clinically distinct setting. While CD44 has been broadly implicated in adhesion processes, the precise nature of its interaction, whether through CD44–CD44 homophilic binding, CD44–hyaluronan (HA) heterophilic binding, or a combination of both, remains unresolved. Our data provide new evidence in support of a “sandwich” model, where HA functions as a bridging molecule between CD44 molecules on neighboring cells, enabling both homophilic and heterophilic interactions. This mechanistic inference is bolstered by our high-resolution imaging and the statistical power provided by our label-free, high-throughput tracking platform. We will ensure the revised manuscript clearly articulates how our findings build upon and advance the current understanding of CD44-mediated adhesion in the broader context of cancer biology.

2. Extracellular matrix patches in vivo: a very relevant paper, much earlier than those referenced in the manuscript, is Wang H et al. JCB 2004 (exposed laminin patches in living mouse lung), which also described the importance of integrins in cancer cell attachment to endothelial cells and arrest in vivo.

We thank the reviewer for highlighting this important and relevant earlier study. We have included a citation to Wang et al., JCB 2004, in the main text to acknowledge their contribution to the concept of available extracellular matrix (ECM) patches for arrested cancer cell arrest.

While we recognize the significance of their findings, we note that the evidence presented in that study, particularly regarding ECM exposure, is primarily based on a single transmission electron microscopy (TEM) image (Figure 4). In contrast, our study provides a more detailed and temporally resolved analysis of ECM (specifically fibronectin) patch exposure on endothelial monolayers. Through high-content imaging and quantitative analysis, we provide a more comprehensive and robust view of the distribution and dynamics of these ECM

features under flow conditions. We believe this offers substantial new insight into the concept first proposed by Wang et al. and helps extend and validate their pioneering observations.

3. Videos: it is clear from the videos that MIA PaCa2 cells frequently flow through the chambers as large clumps containing many cells. This could well impact on their ability to attach – it is unlikely that a whole cell clump will attach effectively to endothelial cells. What did the authors do to minimize the formation of cell clumps for this cell line? They should analyze and describe in the text how these clumps affect attachment.

We thank the reviewer for bringing this point to our attention. As already indicated to Reviewer 1. MIA PaCa-2 cells tend to aggregate within the tubing and can enter the chamber as clusters. We agree that this behavior may affect their arrest dynamics and warrants further investigation. In our study, we excluded large aggregates (tens to hundreds of cells) from analysis because they do not form adhesive contacts with the endothelium and instead displace previously arrested cells due to bulk or inertia, an effect we repeatedly observed during live imaging. To maintain dataset quality, our deep learning–based segmentation was trained to omit these large aggregates.

Smaller clusters (2–8 cells) are more frequently observed engaging the endothelium and are capable of successfully arresting. In the revised manuscript, we quantify arrests as single cells versus small clusters (2–8 cells). Even for MIA PaCa-2, over 85% of arrests involve single cells; for comparison, approximately 96% of AsPC-1 arrests are single-cell events. Among immune cells, around 10% of neutrophil arrests and about 4% of monocyte arrests involve small clusters. These findings are now described in the text, with the related analysis presented in **Figures 1I and 2E and below for convenience.**

Figure 1I. Percentage of AsPC-1 and MIA PaCa-2 cells arresting as single cells per movie. Here, a cluster is defined as at least two cells that arrest together (at the same time) within a cell diameter of each other (see

Methods for details). Results from the various flow speeds are pooled ($n > 24$ videos, 6-7 biological repeats; see Methods).

Figure 2E: Percentage of neutrophils and PBMCs cells arresting as single cells per movie in the presence or absence of IL-1 β stimulation (10 ng/ml for 2 hours). Here, a cluster is defined as at least two cells that arrest together (at the same time) within a cell diameter of each other (see Methods for details). Results from the various flow speeds are pooled ($n > 16$ videos, 4-8 biological repeats; see Methods).

In addition, from the quantification and images it appears that adherent and arrested MIA PaCa2 cells form more clusters than other cell types, yet the authors state on p. 7 that they clustered similarly to leukocytes. This is surprising and indeed the impression from the images is that not all the MIA PaCa2 cells in clusters are being recognized by the algorithm. The authors should comment on this potential drawback to the quantification.

We thank the reviewer for raising this point and for the opportunity to clarify our analysis. As discussed in our response to Reviewer 1, our clustering analysis does not aim to quantify the number of clusters or the number of cells per cluster. Instead, it assesses whether the spatial distribution of arrested cells deviates significantly from a random pattern, using Ripley's L function and Monte Carlo simulations. This allows us to identify non-random "hotspot" behavior, where cells preferentially arrest in proximity to one another more often than would be expected by chance. We agree that the manuscript text could be more precise in distinguishing between apparent visual clustering and the statistical clustering tendency assessed in our analysis. We have revised the text to explain this distinction and to prevent any misinterpretation regarding the scope and limitations of the clustering quantification.

4. It is surprising that the authors only stimulate the endothelial cells with IL-1 β for 2 hours, which is very short compared to standard stimuli designed to increase, for example, PBMC adhesion and transmigration. Why did they choose this short time point? They should also

test a longer time (e.g. 16 hours) when all adhesion molecules for PMBCs will be strongly upregulated e.g. E-selectin, VCAM-1, ICAM-1, and test differences between unstimulated and stimulated HUVECs at that timepoint. Does IL-1 β treatment alter CD44 expression or % of HUVECs expressing CD44?

We thank the reviewer for this critical point. In our current study, we used a 2-hour IL-1 β (10ng/ml) stimulation protocol that consistently induces a rapid endothelial activation response characterized by increased cellular protrusiveness and altered morphology, features we specifically aimed to capture as part of the early activation phase. This effect tends to diminish after 3–4 hours post-treatment (visual assessment). Our choice was guided by previous literature showing time-dependent, transient surface expression patterns of key endothelial adhesion molecules in response to IL-1 β (Smith et al., JCI 1988; Bevilacqua et al., Science 1989; Shen et al., J Virol 1997), suggesting that the endothelial response is dynamic and not necessarily stronger or more uniform at longer stimulation times.

In response, we expanded our endothelial activation experiments to include both short (2 h) and longer (16 h) IL-1 β treatments (**EV2D-G**). Prolonged exposure to 10 ng/mL IL-1 β was not well tolerated and resulted in significant intercellular gaps; therefore, for the 16 h assays, we reduced the dose to 5 ng/mL to maintain monolayer integrity.

At the molecular level, CD44 expression was mostly unchanged in endothelial cells by IL-1 β (WB and IF, below). By contrast, endothelial adhesion molecules exhibited expected activation kinetics: ICAM-1 and VCAM-1 increased after 2 hours and continued to accumulate by 16 hours, whereas E-selectin was higher at 2 hours and declined by 16 hours.

Functionally, both neutrophils and PBMCs exhibited increased attachment to IL-1 β -stimulated monolayers at 2 hours and 16 hours compared to untreated controls; however, the 16-hour condition resulted in only a modest additional increase over the 2-hour condition. Together, these data support the 2 h stimulation used in the primary assays as a physiologically relevant and practical time point that enhances adhesion while minimizing barrier disruption.

Figure EV2D-G: Tracking and analysis of PDAC cells and immune cells' behavior under various conditions

(D-E) Endothelial monolayers, either untreated or treated with IL-1 β (10 ng/ml for 2 hours and 5 ng/ml for 16 hours), were fixed and stained to visualize E-selectin, VCAM1, ICAM1, and CD44. Stainings were performed without permeabilization to specifically label surface-accessible adhesion molecules. Images were captured using a spinning disk confocal microscope. (D) Representative fields of view are shown. Scale bar: 100 μ m. (E) Quantification of the marker per field of view is presented. Intensities were normalized to the number of nuclei per field of view, as well as the average intensity measured in the control in each repeat (n = 45 field of view, 3 biological repeats).

(F-G) The number of arrested neutrophils (F) or PBMCs (G) over time, in the presence or absence of IL-1 β stimulation (10 ng/ml for 2 hours and 5 ng/ml for 16 hours). Bold lines indicate the average, and shaded areas represent the SD (4-7 biological repeats, see methods).

Figure for the reviewer: CD44 expression levels in HUVECs after IL-1 β stimulation. Endothelial monolayers, either untreated or treated with IL-1 β (10 ng/ml for 2 hours and 5 ng/ml for 16 hours), were lysed, and CD44 protein levels were measured using western blots. A representative western blot is shown, along with the quantifications (6 biological repeats).

5. Given the importance of endothelial selectins for leukocyte rolling and VCAM-1 for cancer cell and leukocyte attachment, the authors should also stain endothelial cells for these molecules after IL1 β stimulation. VCAM-1 has been implicated in the endothelial attachment of a number of cancer cell lines.

We have expanded our analysis to include VCAM-1 and E-selectin (**Fig. 6C-D**). Under our conditions, PDAC attachment to the endothelial monolayer does not correlate with VCAM-1 surface levels: although IL-1 β significantly increases VCAM-1 expression, attachment does not match this increase, suggesting VCAM-1 is unlikely to be a major factor here. To put this into context, FACS profiling of AsPC-1 and MIA PaCa-2 shows low surface $\alpha 4$ integrin (the primary VCAM-1 ligand, as $\alpha 4\beta 1$) compared to other integrin heterodimers. Overall, our data consistently indicate that CD44 is a key adhesion molecule mediating PDAC attachment to endothelial monolayers in this assay.

Figure 6C and D: Spatial relationship between arrested PDAC cells (MIA PaCa-2 and AsPC-1) and endothelial adhesion molecules. Labeled PDAC cells were perfused over an endothelial monolayer for 10 minutes, fixed, and stained to visualize E-selectin (E-select), VCAM1, ICAM1, ICAM2, CD44, and fibronectin (FN). Images were captured using a spinning disk confocal microscope. **(C)** Representative fields of view are displayed. Scale bar: 100 μ m. **(D)** PDAC cells were automatically segmented, and the positivity of the endothelial cells in contact with the arrested PDAC cells for each specific marker was manually scored. The percentage of PDAC cells in contact with marker-positive areas is shown (observed). The expected rate of cells in contact with each marker was calculated using Monte Carlo simulations, which took into account the cell diameter and the area of the field of view covered by each marker (see Methods for details) ($n = 71$ -143 fields of view, three biological replicates).

Confidential figure for reviewers removed

6. It is not stated in some figure legends whether or not endothelial cells were stimulated with IL1b prior to analysis. If not, then the comparison +/- IL-1 β is important in places. For example, in Figure 5 it is clear from the images that the endothelial cells have abundant stress fibers. If the cells were quiescent and growth factors reduced (did the authors do this in their experiments?) then they would not have stress fibers unless stimulated for several hours with a pro-inflammatory cytokine such as IL1 β . It is important to test whether the topology and stiffness of the endothelial cells (AFM) is altered under their conditions by IL-1 β stimulation.

We thank the reviewer for this observation and apologize for any confusion caused by the incomplete labeling of IL-1 β treatment status. We have carefully reviewed all figure legends and ensured that IL-1 β stimulation is clearly and consistently indicated throughout the revised manuscript.

Regarding Figure 5, we confirm that no IL-1 β stimulation was used. The endothelial cells were maintained under standard conditions with complete endothelial growth media containing 2% FBS and growth supplements, as detailed in the Methods section. We did not use growth factor–reduced media. The presence of stress fibers under these conditions is consistent with prior studies, which report that even in the absence of pro-inflammatory cytokines, such media can sustain moderate levels of cytoskeletal organization and tension in HUVEC monolayers.

Regarding the suggestion to perform AFM measurements to assess changes in endothelial stiffness following IL-1 β stimulation, we appreciate the scientific merit of this idea. However, we respectfully note that IL-1 β activation is not a primary focus of our study. Given the technical demands and the time required to perform AFM analysis, and considering the limited direct impact such data would have on the main conclusions, we believe this lies outside the scope of our current work.

7. When tracking cells that are ‘in focus’, are most of them traveling at a lower speed than the flow rate, simply because they are close to the wall of the microfluidic chamber and the flow rate is lower here? Presumably the cells moving at this rate are not attached and hence not migrating on the endothelial surface. This is not completely clear from the text description.

We thank the reviewer for this insightful question. Indeed, when tracking cells that are ‘in focus,’ they are situated near the bottom of the microfluidic channel, within approximately 10–25 μm of the glass substrate, where the local flow speed is lower than the maximal flow speed available in the channel due to the Gaussian (Poiseuille) flow profile (**represented below** from our data, similar to commercially available documentation (Ibidi)).

Additionally, the theoretical maximum flow speed in our channel geometry, assuming laminar flow, occurs at the centerline of the channel and is higher than the speed observed near the wall. However, our speed calibration was performed directly within the imaging window using non-adherent cells flowing through BSA-coated channels. Under these conditions, we measured a maximum flow speed of approximately 400 $\mu\text{m}/\text{sec}$ in the near-wall region, which establishes the relevant baseline for the cells we track (**AppendixFig. 1A-B**).

Cells in this region are not necessarily adherent; instead, they are suspended in flow and located where physical contact with the endothelial monolayer is possible. Our tracking approach is designed to capture cells at this interface, enabling us to observe the complete sequence of behaviors, including free flow, deceleration, landing, arrest, and potential detachment. Cells that maintain high velocity throughout their trajectory are categorized as non-adherent, while those that decelerate and remain below defined speed thresholds are classified as arrested.

Figure for the reviewer: Theoretical flow profile in our channels.

AppendixFig. 1A-B, Flow profile: Validation of flow profiles in microfluidic channels. PDAC cells were perfused through BSA-coated channels and imaged using brightfield microscopy just above the glass bottom of the channel. Cell detection was performed using a custom-trained StarDist model, and cell tracking was accomplished with TrackMate. **(A)** Validation of the StarDist model used to detect the perfused cancer cells and **(B)** quantification of mean cell velocity in the microfluidic channel. The Input, the StarDist prediction, and the ground truth (GT) image are displayed alongside the F1 score. **(A)** Scale bar = 100 µm.

8. For neutrophil tracking, did the authors include neutrophils migrating under the endothelial cells (transmigrated) as well as on the surface?

We thank the reviewer for this question. In our study, we focused specifically on the arrest phase of circulating cells, as cancer cells, unlike neutrophils, do not transmigrate within the 10-minute window of our live imaging experiments. To ensure consistency across conditions and cell types, we designed our deep learning segmentation models to detect only those cells that remained rounded and in contact with the endothelial surface. Neutrophils that underwent transmigration rapidly changed shape, spreading beneath the endothelial monolayer. These morphological changes made them undetectable to the segmentation model, which was designed for surface-adherent, rounded cells. As a result, transmigrated neutrophils were intentionally excluded from the analysis. This allowed us to consistently compare the arrest behavior of cancer cells and immune cells, directly within the same imaging timeframe and under similar segmentation criteria.

9. A table delineating the mutations in each of the PDAC cell lines used should be included for reference. The authors should also state whether there is any connection between mutational status and adhesion to HUVECs/ CD44 expression.

We thank the reviewer for this valuable suggestion. We used the Cancer Cell Line Encyclopedia (CCLE) database (DOI: 10.1038/nature11003) to compile and present the mutational profiles of the PDAC cell lines included in our study. This information is summarized in a new figure panel in the revised manuscript for easy reference (see **Appendix Fig. 1C and below for convenience**).

Based on the available data, we do not see a clear correlation between specific oncogenic mutations and the adhesion behavior of these cell lines under flow conditions.

Regarding CD44 expression levels (or isoforms), to our knowledge, there is no established link between CD44 expression and the mutational burden of PDAC cell lines. However, this remains an intriguing question, and future studies could explore whether specific mutations influence CD44 regulation or function in vascular adhesion.

Appendix Fig. 1C: Tumor mutational burden of PDAC lines included in this study. Source data: Cancer Cell Line Encyclopedia, queried for mutations in genes: KRAS, BRAF, CDKN2A, PIK3CA, SMAD4, ARID1A, TGFBR2, TP53, FN1, MUC16, and BRCA2.

10. P. 6: the authors state that the basal ECM should presumably be inaccessible to their antibodies, but it should be accessible where it is exposed and indeed this is shown in their images (and see Wang et al. JCB 2004).

We thank the reviewer for this clarification and fully agree. We apologize if our original wording was unclear. As shown in our images and consistent with previous work (e.g., Wang et al., JCB 2004), the basal extracellular matrix is accessible where it is exposed, and we successfully detect fibronectin (FN) in these regions. In the manuscript, our goal was to convey that we did not observe apical fibronectin patches on the endothelial surface

accessible to circulating cells. We have revised the text to clearly differentiate between apical and exposed basal FN and to avoid any confusion regarding antibody access or ECM localization.

11. Others have previously reported heterogeneity of cancer cell lines in their ability to attach to endothelial cells in vitro, which should be referenced in the Introduction.

We thank the reviewer for this important point. We agree that previous studies have reported heterogeneity among cancer cell lines in their capacity to adhere to endothelial cells in vitro. We have revised the Introduction to acknowledge this prior work and provide appropriate references after conducting a careful literature review.

12. In a few panels, the authors do not state which PDAC cell line was used e.g. Figure 1B.

We thank the reviewer for bringing this to our attention and apologize for the oversight. We have carefully reviewed the relevant figure panels and updated them to specify the PDAC cell line used in each case. This will ensure clarity and consistency across the manuscript. The following modifications have been made:

- **Figure 1B:** “PDAC” will be labeled as AsPC-1
- **AppendixFig. 1A:** “PDAC” or “cancer cells” will be labeled as MIA PaCa-2
- **AppendixFig. 1D:** “PDAC” or “cancer cells” will be labeled as AsPC-1
- **AppendixFig. 2:** “PDAC” will be labeled as MIA PaCa-2
- **Figure 4C:** “PDAC” will be labeled as Panc 10.05
- **Figure 4F:** “PDAC” will be labeled as MIA PaCa-2
- **Figure 5H, 5J:** “PDAC” will be labeled as MIA PaCa-2

13. The authors should convert their chamber flow speeds to approximate dynes/cm² which are commonly used by other groups.

In the revised manuscript, we will explicitly equate our flow speed with the resulting shear force at the surface of the endothelial monolayer.

“The shear stresses applied to the circulating cells in our system at the surface of the endothelial cells range from 0.4 to 0.05 dyn/cm²”

14. Other groups have reported that leukocytes/cancer cells mostly arrest close to endothelial cell junctions and the extension of filopodia by arrested cancer cells/leukocytes; their work should be acknowledged on p. 6 and p. 7.

We thank the reviewer for this important point. We have now updated our manuscript accordingly.

15. Figure 4, zebrafish study. How many cells were analyzed in how many different zebrafish? Is this just one image? Given that the authors have not tested their automated tracking algorithms in vivo, it is not clear why this single image is relevant unless quantitative analysis is added.

We thank the reviewer for this helpful comment. The zebrafish panel in **Figure 4** originates from a single embryo. It serves as a representative example to visually illustrate a key spatial constraint observed in vivo: the proximity between arrested cancer cells and endothelial cell-cell junctions.

Although this panel lacks support from in vivo automated tracking or quantification, we included it as a qualitative illustration to complement our in vitro observations. In the revised manuscript, we clearly indicate that this experiment is a proof of concept.

“As proof of concept, we extended these observations to an in vivo context by injecting cancer cells into zebrafish embryos with labeled endothelial cell junctions (VE-Cadherin-GFP).”

16. Figure 5J: why did the authors allow the MIA PaCa2 cells to interact with HUVECs for 30 minutes here, much longer than their other assays? This should be explained in the text.

We thank the reviewer for this helpful comment. The extended time point used in **Figure 5J** reflects the specific requirements of the structured illumination microscopy (SIM) protocol, designed to capture the stable adhesion phase of MIA PaCa-2 cells interacting with the endothelial monolayer.

To achieve this, we perfused MIA PaCa-2 cells for 10 minutes, followed by a 20-minute wash with cell-free media at the same flow rate. This ensured that only firmly adherent cells remained on the endothelium by the time of fixation and imaging, enabling us to focus on cells that had established stable adhesive contacts. Based on prior data from our lab examining cancer cell transmigration, we know that MIA PaCa-2 cells typically do not initiate transmigration until 1–2 hours post-attachment. Therefore, at the 30-minute mark, the cells remain surface-attached and have not yet begun extravasation.

17. Figure 6: what % of ICAM-1-expressing endothelial cells also express CD44?

To answer this question, we co-stained endothelial monolayers for ICAM-1 and CD44, performed single-cell segmentation, and measured the fluorescence intensity of each marker in individual cells. Instead of imposing a binary “positive/negative” classification based on an arbitrary cutoff, we initially analyzed the continuous signals (**see figure below**).

Importantly, across all cells, ICAM-1 and CD44 showed no significant co-variation (Spearman's $\rho = -0.102$, Pearson's $r = -0.086$), indicating a very weak negative relationship. When focusing on the highest expressers (top 25% of cells for each marker), the negative association became more evident (Spearman $\rho = -0.328$, Pearson $r = -0.244$). This suggests that cells with high ICAM-1 expression are less likely to have high CD44 expression than what would happen by chance under our conditions.

Using the top 25% of ICAM-1 intensity as the cutoff for “ICAM-1-high,” 36% of those cells were CD44-positive.

Figure for the reviewer: Co-expression of CD44 and ICAM-1 at single-cell resolution. Endothelial monolayers were co-stained for CD44 and ICAM-1, segmented, and per-cell fluorescence intensities were quantified from sum-projection images. Background levels per image (5th percentile) were subtracted, values were log_{1p}-transformed, and then scaled using min-max normalization across each replicate. A 2D hexagonal bin density plot displays the normalized CD44 versus ICAM-1 intensities, with color indicating the number of cells per bin. Dashed lines mark the median values for each marker after normalization.

In panel B, why were the data for control and IL-1 β stimulation pooled? Can they also be shown separately?

We thank the reviewer for this question. The data in **Figure 6B** were pooled to increase statistical power, as each field of view provides a single measurement, and separating the conditions initially led to limited data points per group. Additionally, our analysis showed no significant difference between IL-1 β -treated and control conditions, which supported pooling for the initial representation.

18. Did the CD44 blocking antibody reduce adhesion of PBMCs or neutrophils as well as PANC cells? This is an important control to include, because CD44 should be specific to cancer cell attachment.

We thank the reviewer for this suggestion. As outlined in our revision plan, we tested anti-CD44 blockade on leukocytes (PBMCs and neutrophils) so these data could be directly compared to the anti-CD44 experiments on PDAC cells. Under the baseline (non-IL-1 β -stimulated) endothelial conditions used, leukocyte attachment is low. Still, our data indicate that targeting CD44 appears to limit neutrophil adhesion, while targeting CD44 seems to increase PBMC attachment slightly. The data are included here for the reviewer's consideration only (**see figure below**). It is unclear why the reviewer suggests that CD44 should be specific to cancer cells, as others have shown that CD44 contributes to leukocyte trafficking (for instance, see the review by McDonald and Kubes, 2015).

Figure for the reviewer: The number of arrested neutrophils (Neutro.) or PBMCs over time, in the presence or absence of pre-treatment with a CD44 blocking function antibody. Bold lines indicate the average, and shaded areas represent the SD (4 biological repeats).

Dear Guillaume,

Thank you for submitting a revised version of your manuscript. We have now received input from both original reviewers, who are broadly satisfied with the performed revisions. There now remain only a few editorial points that need to be addressed before I can extend official acceptance of the manuscript:

1. In the author checklist, please select the appropriate responses in the column D.
2. Please check that the funding information is correct and identical both in the manuscript and our online system; currently, Research Council of Finland 338585 is missing in our system; also Academy of Finland (357910, 35791, and 337120), Solutions for Health strategic funding to Åbo Akademi University, the Finnish Cancer Institute, the Research Council of Finland Centre of Excellence program (346131 & 364182), the Marie Skłodowska-Curie grant agreement (841973 and 101108089), and the Turku Collegium for Science, Medicine, and Technologies. Please make sure to add them if these funders are relevant and should be included in the list.
3. CRediT has replaced the traditional author contributions section because it offers a systematic, machine-readable author contributions format that allows for more effective research assessment. Please remove the Authors Contributions from the manuscript and use the free text boxes beneath each contributing author's name in our online submission system to add specific details on the author's contribution. More information is available in our guide to authors.
4. Please merge "Code availability" with the "Data Availability" section.
5. For movies, please remove the legends from the manuscript text file and ZIP them with each movie file. Please correct the titles and the callouts in the manuscript text to "Movie EV1" - "Movie EV10". Further information is available here: <https://www.embopress.org/page/journal/14602075/authorguide#expandedview>
6. Please do upload the Reagents and Tools Table as a separate file choosing the file type "Reagent Table".
7. Please update references according to The EMBO Journal style - where there are more than 10 authors on a paper, the first 10 should be listed, followed by 'et al.' Please remove DOIs from the reference list. Further information can be found here: <https://www.embopress.org/page/journal/14602075/authorguide#referencesformat>
8. In the Appendix, please add a table of contents with page numbers on the first page and correct the nomenclature to "Appendix Figure S1" etc. Please remove the appendix legends from the main manuscript text.
9. Our data editors have flagged the following issues in figure legends that need correcting:
 - Please provide the exact p values in the legends of figures 5E, G; 7G-L; 8B, F-H; EV1 B, EV2 B, C, E; EV4 A, C, F, H.
 - Please define the box plots in terms of minima, maxima, centre, bounds of box and whiskers, and percentile in the legends of figures 1I, 2E.
 - Please provide information on the number and nature of replicates in the legends of figures 2D, F.
 - Please note that, according to the source data, only 2 biological repeats were performed for the AsPC1 condition in the right panel for figure 8E. Since statistical analysis cannot be reliably performed on such a small number of replicates, please do not show error bars for this condition.
 - Please define the error bars in the legend of figure 4B.
10. Papers published in The EMBO Journal are accompanied online by a 'Synopsis' to enhance discoverability of the manuscript. It consists of A) a short (1-2 sentences) summary of the findings and their significance, B) 3-4 bullet points highlighting key results (the highlights can be repurposed for this) and C) a synopsis image that is 550x300-600 pixels large (width x height, jpeg or png format). You can either show a model or key data in the synopsis image. Please note that the image size is rather small and that text needs to be readable at the final size.
11. As part of the EMBO Press transparent editorial process, The EMBO Journal will publish online a Peer Review File to accompany accepted manuscripts. This file will be published in conjunction with your paper and will include the anonymous referee reports, your point-by-point response and all pertinent correspondence relating to the manuscript, including decision letters. Please note that the Author Checklist will be published at the end of the Peer Review File. Please let us know if you want to remove or not any figures or data from the Peer Review File prior to publication. Please note that retaining unpublished data in the Peer Review File means that these count as published and that the Peer Review File would need to be referenced in future publications.

With best wishes,

leva

leva Gailite, PhD
Senior Scientific Editor

The EMBO Journal
Meyerhofstrasse 1
D-69117 Heidelberg
Tel: +4962218891309
i.gailite@embojournal.org

We realize that it is difficult to revise to a specific deadline. In the interest of protecting the conceptual advance provided by the work, we recommend a revision within 3 months (18th Feb 2026). Please discuss the revision progress ahead of this time with the editor if you require more time to complete the revisions.

Referee #1:

I would like to congratulate the authors for the development of the development of FlowVision and for the detailed characterization of PDAC binding to endothelium, as well as for the discovery of their interaction with CD44. In my opinion, the authors fully addressed my comments and other reviewers, and no further revisions are needed.

Referee #2:

The Authors have addressed all of the Reviewers comments. The only remaining concern is regarding the novelty of the segmentation/tracking pipeline (of which there are many) and biological findings which use PDAC cell lines and HUVEC to explore binding and adhesion dynamics. The main finding is that CD44 and HA play a significant role in adhesion, which has been shown previously. The data is however extensive, sound within the model limitations and well-characterized and still presents a valuable set of work for the community. This work is a stepping stone for modeling extravasation.

The authors addressed the remaining editorial issues.

Dear Guillaume,

Thank you for incorporating the final editorial requests in the revised version. I am now pleased to inform you that your manuscript has been accepted for publication.

Before we forward your manuscript to our publishers, we would like to propose some edits in the manuscript title, abstract and synopsis - please see in the attached file. We would like to suggest a lightly more general title version that highlights the imaging approach. We have also prepared a short blurb that will accompany the title of your manuscript in our online system. Please take a look and let me know if any corrections are needed.

You may qualify for financial assistance for your publication charges - either via a Springer Nature fully open access agreement or an EMBO initiative. Check your eligibility: <https://link.springer.com/journal/44318/how-to-publish-with-us>

If you have any questions, please do not hesitate to contact the Editorial Office. Thank you for this contribution to The EMBO Journal, and congratulations on a nice study!

With best wishes,

Ieva

Ieva Gailite, PhD
Senior Scientific Editor
The EMBO Journal
Meyerhofstrasse 1
D-69117 Heidelberg
Germany
i.gailite@embojournal.org

Please note that it is The EMBO Journal policy for the transcript of the editorial process (containing referee reports and your response letters) to be published as an online supplement to each paper. If you should prefer removal of any referee-only figures included in the point-by-point response(s), e.g. because they may still be used for future publication or because they have been reproduced from published work by others, please do let us know immediately via response email.

More information is available here: <https://link.springer.com/partners/embo-press/editorial-policies#Peer%20review>